# Unifying Attention Heads and Task Vectors via Hidden State Geometry in In-Context Learning

**Haolin Yang**[1]     **Hakaze Cho**[2]     **Yiqiao Zhong**[3]     **Naoya Inoue**[2,4]

[1]University of Chicago     [2]JAIST     [3]University of Wisconsin - Madison     [4]RIKEN

haolinyang2001@uchicago.edu, yfzhao@jaist.ac.jp
yiqiao.zhong@wisc.edu, naoya-i@jaist.ac.jp

## Abstract

The unusual properties of **in-context learning** (ICL) have prompted investigations into the internal mechanisms of large language models. Prior work typically focuses on either special attention heads or task vectors at specific layers, but lacks a unified framework linking these components to the evolution of hidden states across layers that ultimately produce the model's output. In this paper, we propose such a framework for ICL primarily in classification tasks by analyzing two geometric factors that govern performance: the *separability* and *alignment* of query hidden states. A fine-grained analysis of layer-wise dynamics reveals a striking two-stage mechanism—separability emerges in early layers, while alignment develops in later layers. Ablation studies further show that Previous Token Heads drive separability, while Induction Heads and task vectors enhance alignment. Our findings thus bridge the gap between attention heads and task vectors, offering a unified account of ICL's underlying mechanisms.[1]

## 1 Introduction

One of the most remarkable features of **L**arge **L**anguage **M**odels (LLMs) is their ability to respond to novel queries in user-desired mannerisms on-the-fly solely by learning from demonstrations provided in the input (as shown in Figure 1 (A))—without any additional training. This capability is known as **I**n-**c**ontext **L**earning (ICL) [Brown et al., 2020, Radford et al., 2019]. ICL has revolutionized natural language processing by reducing dependence on burdensome data collection and costly finetuning, enabling swift and seamless adaptation of LLMs to a variety of downstream tasks [Dong et al., 2024].

Due to its stark departure from traditional gradient-based learning paradigms, ICL has attracted significant academic interest. One line of research aims to develop a mechanistic understanding of ICL by analyzing the internal behavior of LLMs in ICL settings [Reddy, 2024]. Some studies highlight the importance of key structural components in LLMs, such as **I**nduction **H**eads (IH) [Elhage et al., 2021, Olsson et al., 2022, Cho et al., 2025a], which retrieve information from demonstrations and perform copy-like operations within Transformer-based architectures. Other studies adopt a hidden-state-centric view, interpreting ICL as a process in which LLMs construct vector representations of the task from demonstrations (so-called task vectors [Hendel et al., 2023, Liu et al., 2024]) using the hidden states at an intermediate layer, and then use the vectors to steer hidden states and produce better next-token predictions. Both lines of work support their claims primarily through ablation and intervention experiments that assess the impact on model outputs.

**However, treating model components and hidden states as separate entities in ICL analysis is both unnecessary and misleading.** The two are inherently intertwined—components like attention heads transform hidden states layer by layer, and these intermediate updates ultimately determine the

---

[1]Code implementation: https://github.com/HLYang2001/ICL_Hidden_Geometry.

39th Conference on Neural Information Processing Systems (NeurIPS 2025).

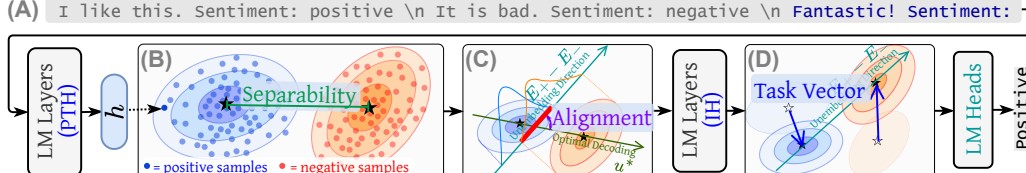

Figure 1: **(A)** An example for ICL input. **(B)** In early layers, LLMs promote **separability** among the last tokens' hidden state ($h$) clusters w.r.t. the ground-truth labels of the queries through **P**revious **T**oken **H**eads (PTHs). **(C)** In early layers or zero-shot scenarios, the direction where the hiddens are maximally separated is insufficiently **aligned** with the output direction (i.e., the difference vector of the label-token unembedding vectors), increasing cluster overlap after mapping and inducing higher classification errors, and **(D)** in later layers, **I**nduction **H**eads (IHs) align these clusters towards the output direction, with the same underlying mechanism of task vectors.

model's output. As such, analyzing only the influence of attention heads or task vectors on the final prediction reduces ICL to a black-box process. A true mechanistic understanding of ICL requires tracing how these components progressively shape hidden states throughout the model, not just their end effects. A unified framework that synthesizes both perspectives and assesses their importance in the layer-wise evolution of hidden states is thus highly essential.

Therefore, in this work, we propose a framework, shown in Figure 1, based on a key geometric insight into LLM classification: the success of LLMs in classification depends on whether the hidden states of queries with different ground-truth labels are **(1)** sufficiently separable and **(2)** well aligned with the unembedding vectors of labels. In other words, hidden states should be separated along a direction that aligns with the difference between the unembedding vectors of different labels. To investigate this, we collect hidden states in both ICL and zero-shot settings and analyze them using a range of geometric measures to assess how ICL improves hidden states in terms of separability and alignment. Our approach operates on three levels: **(1)** We examine the layer-wise trends in these measures and compare zero-shot with ICL (and across different ICL settings) to quantify the effect of ICL on the hidden state geometry. **(2)** We observe how changes in measure values reflect the evolution of semantic information represented in the hidden states that ultimately inform model outputs in ICL. **(3)** We probe specific layers to examine the role of different types of attention heads and how they influence both the values and trajectories of these geometric measures.

Our experiments find that ICL improves classification performance primarily by enhancing alignment between query hidden states and unembedding vectors, with less impact on separability. Results further suggest that ICL proceeds as a two-stage process: early layers mainly increase hidden-state separability, while in middle-to-late layers, separability plateaus and alignment improves. The refinement of alignment arises through layer updates that filter out task-irrelevant semantics and preserve label-relevant content, thereby semantically refining query hidden states. Such updates qualify ICL query hidden states as task vectors. We also test this two-stage characterization on ICL for a generation task and confirm its explanatory power across settings.

To build the bridge between the task vector and model components, we analyze the effect of attention heads on these geometric measures and find that (as demonstrated by Figure 1): **(1)** The early increase in separability is driven by a proliferation of **P**revious **T**oken **H**eads (PTHs)—attention heads that attend to the previous token at each position—in early layers. **(2)** In middle-to-late layers, **I**nduction **H**eads (IHs) enhance alignment by amplifying hidden state components along the unembedding directions of the labels. Thus, our findings not only explain why attention heads like PTHs and IHs are essential for ICL performance, but also clarify why induction head outputs empirically serve as effective task vectors [Todd et al., 2024, Hendel et al., 2023].

## 2   Related works

**IH and PTH**   Previous works on the mechanistic cause of ICL identify structural circuits in Transformers comprising a **P**revious **T**oken **H**ead (PTH) and an **I**nduction **H**ead (IH), where the PTH gathers information at demonstration label tokens, and the IH copies it to query positions for prediction [Olsson et al., 2022, Singh et al., 2024, Song et al., 2025]. Their importance in classification has been confirmed in large-scale models as well [Cho et al., 2025a]. However, existing analyses focus on end-to-end effects, leaving open how these heads shape query hidden states.

**Task vectors** An alternative theory proposes that LLMs summarize demonstrations into task vectors, which can be used to steer zero-shot hidden states to increase prediction accuracy. These vectors may be ICL query hidden states extracted at an intermediate layer [Hendel et al., 2023, Liu et al., 2024], attention head outputs [Todd et al., 2024, Li et al., 2024], or MLP outputs [Merullo et al., 2024]. Decoding task vectors through the unembedding matrix reveals that they encode task-related tokens and support semantic manipulations akin to "word2vec"-style algebra [Mikolov et al., 2013]. Despite intriguing empirical findings, explanations for their mechanism remain preliminary. Kahardipraja et al. [2025] show that heads expressing contextual information or model's parametric knowledge produce task vectors with distinct function. Jiang et al. [2025b] and Han et al. [2025] study how the geometry of model's task representation impacts the effects of injecting task vectors. Jiang et al. [2025a] reveal the correlation between model's catastrophic forgetting and the change in task vectors. **However, none of these studies answer the question of how task vectors influence the concrete internal computation process of models which directly and ultimately determine the outputs.**

**Geometry of hidden states** LLMs encode inputs in geometrically structured ways [Park et al., 2024]. For instance, they encode cities and countries by geographic relations [Gurnee and Tegmark, 2024] and numbers in rings under modular arithmetic [Liu et al., 2023]. For classification tasks, they can separate hidden states into clusters by attributes such as truth value [Marks and Tegmark, 2024] or toxicity [Lee et al., 2024]. However, the alignment of such clusters with the unembedding vectors of their labels—an equally crucial geometric factor—has received limited attention, with only a few studies noting suboptimal alignment at the output layer [Cho et al., 2025b, Kirsanov et al., 2025].

## 3 Theoretical framework

We begin our framework with an analysis of classification tasks, where an LLM is prompted with a query $x$ and its ground-truth label $y$, e.g., $x =$ "I like this movie. Sentiment:" and $y =$ "positive". The LLM with $L$ layers encodes $x$ into a vector $\boldsymbol{h}_x \in \mathbb{R}^d$, taken from the hidden state at the final token (":"). Given a vocabulary $\mathbb{V} = \{v_1, ..., v_{|\mathbb{V}|}\}$ and an output embedding matrix $\boldsymbol{E} \in \mathbb{R}^{|\mathbb{V}| \times d}$, the model predicts $p(v_j|x) = \exp(\boldsymbol{E}_{v_j}\boldsymbol{h}_x)/\sum_{j'=1}^{|\mathbb{V}|} \exp(\boldsymbol{E}_{v_{j'}}\boldsymbol{h}_x)$, where $\boldsymbol{E}_{v_j}$ is the unembedding vector of $v_j$. The model outputs the token $v_j$ with the highest probability as the predicted label for $x$.

In the ICL scenario, as shown in Figure 1 (A), $k$ demonstration-label pairs $(x_1, y_1), ..., (x_k, y_k)$ are prepended to form the prompt $[x_1, y_1, ..., x_k, y_k, x]$, which modifies the hidden state of $x$ and results in a different distribution $p(v_j|x_1, y_1, ..., x_k, y_k, x)$, potentially improving prediction accuracy.

Rather than a single query, we more often evaluate *classification accuracy over a dataset* of $n$ queries $\{x_1, ..., x_n\}$, with hidden states $\{\boldsymbol{h}_1, ..., \boldsymbol{h}_n\}$ and ground-truth labels $\{y_1, ..., y_n\}$. We will show that the classification accuracy depends on two geometric properties of the hidden state collection $\boldsymbol{H} = [\boldsymbol{h}_i]_{i=1}^n \in \mathbb{R}^{n \times d}$: **(1)** its separability, and **(2)** its alignment with the unembedding vectors.

As mentioned before, an LLM performs classification using the unembedding matrix $\boldsymbol{E}$, which is a multiclass classifier on the hidden states. We define its accuracy on the dataset as:

$$\text{Acc} = \frac{1}{n}\sum_{i=1}^n \mathbb{1}(y_i = \arg\max_{v_j} \frac{\exp(\boldsymbol{E}_{v_j}\boldsymbol{h}_i)}{\sum_{j'=1}^{|\mathbb{V}|} \exp(\boldsymbol{E}_{v_{j'}}\boldsymbol{h}_i)}) = \frac{1}{n}\sum_{i=1}^n \mathbb{1}(y_i = \arg\max_{v_j} \frac{\boldsymbol{E}_{v_j}\boldsymbol{h}_i}{\sum_{j'=1}^{|\mathbb{V}|} \boldsymbol{E}_{v_{j'}}\boldsymbol{h}_i}), \quad (1)$$

where the first equality states that classification accuracy equals the proportion of the labels being predicted as the most probable token, and the second equality uses the monotonicity of softmax.

We first consider the simple case where $\{x_1, ..., x_n\}$ only has 2 different labels, $y_A$ and $y_B$, i.e. $y_i \in \{y_A, y_B\}, \forall i$, and leave the generalization to the case of more labels to Appendix B. For a unit vector $\boldsymbol{u} \in \mathbb{R}^d$, we define the **separability of $\boldsymbol{H}$ along direction $\boldsymbol{u}$** with labels $y_A, y_B$ to be:

$$S(\boldsymbol{u}) = \frac{1}{n}(\sum_{i \in \mathbb{N}_A} \mathbb{1}(\boldsymbol{u}^\top \boldsymbol{h}_i \geq 0) + \sum_{i \in \mathbb{N}_B} \mathbb{1}(\boldsymbol{u}^\top \boldsymbol{h}_i < 0)), \quad (2)$$

where $\mathbb{N}_A = \{i : y_i = y_A\}$ and $\mathbb{N}_B = \{i : y_i = y_B\}$. $S(\boldsymbol{u})$ measures the fraction of hidden states that can be correctly classified by a linear decision boundary orthogonal to $\boldsymbol{u}$. By considering all possible directions, we obtain the **maximum separability of $\boldsymbol{H}$**, i.e. $S^* = \sup_{\boldsymbol{u} \in \mathbb{R}^d, \|\boldsymbol{u}\|_2 = 1} S(\boldsymbol{u})$. Though $S(\boldsymbol{u})$

is not a continuous function, $S^*$ as a supremum can be attained on $\mathbb{S}^{d-1} = \{\boldsymbol{u} : \boldsymbol{u} \in \mathbb{R}^d, \|\boldsymbol{u}\|_2 = 1\}$ (See Appendix A). Denote $\boldsymbol{u}^*$ as the $\boldsymbol{u}$ where $S$ attains the supremum, then:

**Theorem 1** $\text{Acc} \le S^*$. *The equality is achieved when* $\max_{v \in \mathbb{V}, v \notin \{y_A, y_B\}} \boldsymbol{E}_v \boldsymbol{h}_i < \max(\boldsymbol{E}_{y_A} \boldsymbol{h}_i, \boldsymbol{E}_{y_B} \boldsymbol{h}_i), \forall i$ *and* $\frac{\boldsymbol{E}_{y_A} - \boldsymbol{E}_{y_B}}{\|\boldsymbol{E}_{y_A} - \boldsymbol{E}_{y_B}\|_2} = c\boldsymbol{u}^*$ *for some positive constant c.*

This theorem shows that accuracy is upper-bounded by the **maximum separability** of $\boldsymbol{H}$ and that the bound is achieved when: **(1)** each $\boldsymbol{h}_i$ is maximally aligned with either $\boldsymbol{E}_{y_A}$ or $\boldsymbol{E}_{y_B}$—the unembedding vectors of $y_A$ and $y_B$—while excluding interference from other tokens, a condition we call **output alignment**; and **(2)** the direction of maximum separation $\boldsymbol{u}^*$ aligns with $\boldsymbol{E}_{y_A} - \boldsymbol{E}_{y_B}$, which we refer to as **directional alignment**. Thus, separability and alignment of hidden states critically determine accuracy. Since query hidden states in ICL differ from those in the zero-shot setting due to the presence of demonstrations, we conclude that ICL can improve classification accuracy by enhancing both their separability and alignment.

**Attention heads** influence query hidden states as well by their outputs to the residual stream. At layer $l$, the $h^{\text{th}}$ attention head adds $\boldsymbol{a}_{i,l}^h$ to the hidden state $\boldsymbol{h}_{i,l}$ of the $i^{\text{th}}$ query, affecting the transition to $\boldsymbol{h}_{i,l+1}$. Ablating a head (i.e., setting $\boldsymbol{a}_{i,l}^h = 0$) alters all downstream hidden states and can degrade accuracy if the final-layer hidden states $\boldsymbol{H}_L$ lose separability or alignment.

**Task-vector**-based experiments operate similarly by steering hidden states at intermediate layers. In those experiments, a task vector $\boldsymbol{t}_i$ is added to[2] $\boldsymbol{h}_{i,l}$ and the influence likewise propagates through subsequent layers. This intervention is effective if the final $\boldsymbol{H}_L$ has better separability and is better aligned with the labels of the task encoded in $\boldsymbol{t}_i$.

# 4   Method: Measuring separability and alignment of hidden states

Given the aforementioned influence of ICL, attention heads, and task vectors on the separability and alignment of hidden states across layers, we introduce several measures[3] to quantify their influence. The formal definitions and mathematical details of all measures are provided in Appendix C.

**A. Separability measure: Separability score**
Theorem 1 shows that the classification is affected by the maximum separability of hidden states. However, finding maximum separability requires evaluating separability along infinitely many directions in $\mathbb{S}^{d-1}$ and is intractable. As a practical proxy, we train a logistic classifier on a subset of $\boldsymbol{H}$ and evaluate its accuracy on held-out data. We call the accuracy **separability score** as it reflects the empirical maximum separability of hidden states achieved by the classifier.

**B. Alignment measures**
In Theorem 1 we show that both **output alignment** and **directional alignment** affect accuracy. Below, we introduce measures for both.

**B.1 Output alignment**
**Output alignment** is defined as the classification accuracy obtained by applying the unembedding matrix $\boldsymbol{E}$ directly to hidden states $\boldsymbol{H}$, skipping subsequent layers. It measures how well $\boldsymbol{H}$ aligns with the label unembedding vectors $\boldsymbol{E}_{y_A}$ and $\boldsymbol{E}_{y_B}$, and is also known as logit lens accuracy [nostalgebraist, 2020]. When $\boldsymbol{H}$ comes from the final layer, it equals standard classification accuracy.

**B.2–B.5 Directional alignment**
**Directional alignment** is the alignment between $\boldsymbol{u}^*$ and the label unembedding **difference** direction $\frac{\boldsymbol{E}_{y_A} - \boldsymbol{E}_{y_B}}{\|\boldsymbol{E}_{y_A} - \boldsymbol{E}_{y_B}\|_2}$. Since finding $\boldsymbol{u}^*$ is intractable, we introduce the following proxies based on approximating $\boldsymbol{u}^*$ or measuring separability along the label-difference direction.

---

[2] Activation patching which directly replaces the zero-shot hidden states with ICL hidden states can also be subsumed under this form, with $\boldsymbol{t}_i$ being the difference between the hidden states in the two settings.

[3] For binary classification datasets, each measure is computed over the entire $\boldsymbol{H}$. For multiclass datasets with labels $y_1, ..., y_m$, we enumerate all label pairs $(y_j, y_k)$, compute the measure using the corresponding two label clusters, i.e. hidden states of queries with labels $y_j, y_k$, before averaging across all pairs to obtain a scalar summary for $\boldsymbol{H}$. As shown in Appendix B, this is justified by the fact that separability and alignment influence the accuracy of multiclass classification through the binary classification between each label pair.

- **Singular alignment (B.2)**. To approximate $\boldsymbol{u}^*$, we apply Singular Value Decomposition (SVD) to the mean-centered $\boldsymbol{H}$ and extract the top-$r$ right singular vectors. These are the directions where $\boldsymbol{H}$ has the greatest spread and can approximate $\boldsymbol{u}^*$. We compute the maximum absolute cosine similarities between each of these vectors and $\frac{\boldsymbol{E}_{y_A} - \boldsymbol{E}_{y_B}}{\|\boldsymbol{E}_{y_A} - \boldsymbol{E}_{y_B}\|_2}$ as a proxy for directional alignment.

- **Variance-based alignment (B.3)**. To measure separability along $\frac{\boldsymbol{E}_{y_A} - \boldsymbol{E}_{y_B}}{\|\boldsymbol{E}_{y_A} - \boldsymbol{E}_{y_B}\|_2}$, we first measure the proportion of total variance in $\boldsymbol{H}$ that lies along this direction. A higher value indicates that this direction is a principal axis of variation—suggesting stronger directional alignment.

- **Mean-based alignment (B.4)**. We project the difference in means of label clusters—hidden states with labels $y_A$ and $y_B$—onto $\frac{\boldsymbol{E}_{y_A} - \boldsymbol{E}_{y_B}}{\|\boldsymbol{E}_{y_A} - \boldsymbol{E}_{y_B}\|_2}$ and divide it by the pooled projected variance. A higher value implies better separation margin along $\frac{\boldsymbol{E}_{y_A} - \boldsymbol{E}_{y_B}}{\|\boldsymbol{E}_{y_A} - \boldsymbol{E}_{y_B}\|_2}$ and better directional alignment.

- **Composite alignment (B.5)**. To unify the mean- and variance-based alignment, we define composite alignment as their product. It captures both the magnitude and the relative significance of separation along the label-difference direction, serving as a robust proxy for directional alignment.

### C. Indirect measure: Effective dimension

As an indirect indicator of separability and alignment, we compute the effective dimension of $\boldsymbol{H}$—the number of directions along which the hidden states exhibit significant variance. A low effective dimension suggests that hidden states concentrate along a small number of directions, potentially those associated with the label unembedding vectors, and are thus easier to separate.

## 5  Experiments and Results

**Models**  We experiment on the following 7 models: Llama2-7B, Llama2-13B, Llama2-70B [Touvron et al., 2023], Llama3-8B, Llama3-70B [Grattafiori et al., 2024], Gemma-2B, and Gemma-7B [Gemini Team et al., 2024]. Unless otherwise stated, we report the results on Llama2-70B.

**Datasets**  We conduct experiments on two major types of classification datasets: **text classification** (SUBJ [Wang and Manning, 2012], SST-2 [Socher et al., 2013], TREC [Li and Roth, 2002]) and **natural language inference** (SNLI [MacCartney and Manning, 2008], RTE [Dagan et al., 2005], and CB [De Marneffe et al., 2019]). We further include a generation dataset (detailed in Subsection 5.1) to test the generalizability of our results in settings other than classification. We also use a generated dataset to explore the scenario where ground-truth labels of queries are not presented in the demonstrations (discussed in Appendix I.5).

**ICL setting**  We set the number of in-context demonstrations to be 8 unless otherwise stated. The demonstration samples are chosen randomly, except for one experiment that investigates the effect of using demonstrations procured by kNN retrieval [Liu et al., 2022]. For a detailed exposition of the implementation of models, datasets, prompt templates, etc., refer to Appendix D.

### 5.1  Layer-wise trends of separability and alignment

**ICL as a two-stage process**  Figure 2 shows the values of six geometric measures at all layers of Llama2-70B in both zero-shot and ICL settings averaged across datasets (output alignment is deferred to Subsection 5.2 due to its strong link to the semantic information of hidden states; results for other models are in Appendix F). A key observation is that Figure 2 reveals a striking **phase transition** in the evolution along the layers of ICL hidden states. **Phase 1** (Figure 1 (B)): In the early layers, directional alignment measures take low values, whereas separability increases rapidly. **Phase 2** (Figure 1 (D)): In the subsequent layers, this trend reverses: separability plateaus, while all directional alignment measures spike simultaneously, which suggests that the label unembedding difference direction starts to capture more variance and separate label clusters more effectively. It causes the semantic information of query labels to be predominant in $\boldsymbol{H}$, which ultimately leads to final output alignment. These findings are consistent with prior work showing that task information emerges only after the initial layers [Sia et al., 2024]. The alternation between two phases is also captured by the effective dimension of $\boldsymbol{H}$, which first rises and then declines as alignment starts to improve, reflecting a gradual concentration of hidden states along label unembedding directions. In contrast, this transition is much weaker in the zero-shot setting, where effective dimension rises

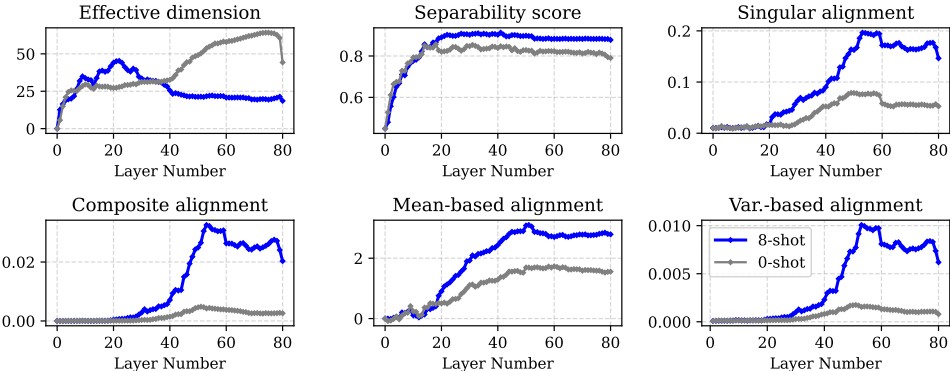

Figure 2: **Comparison of trends in separability and alignment measures: ICL vs. zero-shot.** Under ICL, a clear **phase transition** emerges: separability increases first and then alignment surges. The effective dimension first rises and then declines. This pattern is missing in the zero-shot setting. Accuracy gains from ICL over zero-shot are reflected in alignment, not separability measures.

monotonously and the rise in alignment measures is not nearly as pronounced. Hence, this phase transition pattern characterizes ICL as a **two-stage process**: an initial phase enhancing separability, followed by a refinement phase that improves alignment.

**ICL improves accuracy primarily through alignment**  A key observation is that ICL and zero-shot settings show only a small gap in separability scores—despite an 80% (Figure 2, upper middle) difference in classification accuracy. This implies that the semantic separation of query hidden states is intrinsic to LLM inference rather than a main effect of ICL (see Appendix E.1 for the practical implication of this), making alignment the primary bottleneck in zero-shot performance. As shown in Figure 1 (C), the high-separability direction in zero-shot hidden states is misaligned with the label-difference vector, as reflected in low singular and composite alignment scores. Along this direction, label clusters remain poorly separated (low mean-based alignment) and explain little variance (low variance-based alignment). These findings align with prior work indicating that unembedding vectors often fail to capture the dominant separation axes in hidden state space [Cho et al., 2025b].

Given the volatility of ICL accuracy across settings with different demonstration numbers and input formats, we investigate whether the phase transition persists across settings, and whether accuracy differences can similarly be attributed to alignment properties of hidden states. We consider three settings: (1) Varying the number of demonstrations from 0 to 24 in increments of 4; (2) Changing the demonstration selection strategy to kNN retrieval where demonstrations with the closest zero-shot embeddings to each query are chosen; (3) Replacing original demonstration labels with semantically uninformative symbols like "@" and "#". Figure 3 reports the results averaged across datasets.

**Phase transition is a robust hallmark of ICL**  Figure 3 shows that the phase transition pattern persists across all three settings. In Figure 3 (A), it is evident at all demonstration counts except 0. Figure 3 (B) confirms that principled demonstration selection (via kNN) preserves instead of interfering with the transition. Even in Figure 3 (C), where labels are replaced by symbols, the two-stage pattern in effective dimension remains—distinguishing it from zero-shot, where effective dimension rises monotonically—though the post-transition alignment surge is dampened.

**Phase transition holds in open-ended settings**  To test whether the phase transition pattern occurs in tasks beyond classification, i.e., open-ended generative tasks, where ICL is also frequently applied, we consider the following task: the queries assume the format "Praise/Critique {subject}", where {subject} is the name of a food (e.g., tapas). The labels are reviews for the subject satisfying the sentiment requirement, generated using GPT-4o [OpenAI et al., 2024], for instance, "Praise tapas → The tapas assortment at this restaurant is absolutely delightful!". We track the layer-wise dynamics of the separability score and composite alignment[4] as representatives of the metrics evaluated on the 8-shot and 0-shot hidden states of this task. The results in Figure 4 mirror those in Figure 2, where

---

[4]Since a generation task has an indefinite and infinite label space, we let the label unembedding difference direction needed to calculate composite alignment be that between "positive" and "negative".

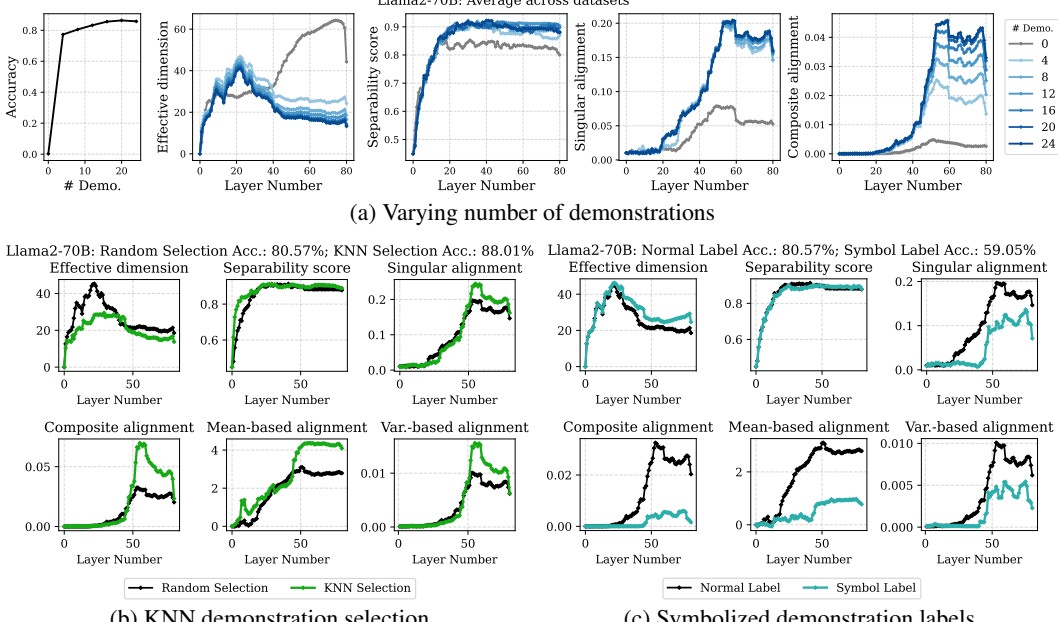

(a) Varying number of demonstrations

(b) KNN demonstration selection          (c) Symbolized demonstration labels

Figure 3: **Layer-wise trends of separability and alignment measures in different ICL settings.**
**(A)** Phase transition is evident under demonstration numbers from 4 to 24. Accuracy improvements of
increasing demonstrations are reflected by consistently improving alignment measures. **(B)** Changing
the demonstration selection method to kNN retrieval preserves phase transition and improves accuracy
through enhancing alignment. **(C)** Using uninformative demonstration labels hurts accuracy due to
decreased alignment of hidden states, yet a similar phase transition is evident.

the few-shot and zero-shot hidden states exhibit similar separability properties across layers, but the
surge in composite alignment beginning from mid-layers is specific to few-shot hidden states.

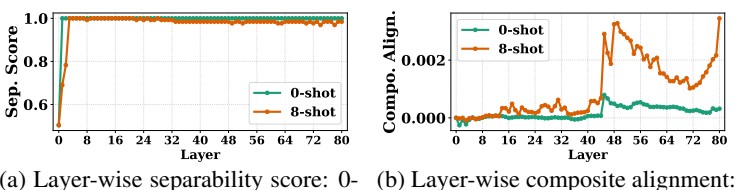

(a) Layer-wise separability score: 0-
shot vs. 8-shot

(b) Layer-wise composite alignment:
0-shot vs. 8-shot

Figure 4: **(A)** In the generation setting, the separability score exhibits
almost identical trends in the 0-shot and 8-shot cases. **(B)** In the 8-shot
case, the phase-transition of composite alignment is pronounced.

**Alignment explains accuracy differences across ICL settings** We find that:
accuracy differences between ICL settings are consistently reflected in alignment measures. Specifically, in Figure 3 (A), increasing the number of demonstrations from 4 to 24 produces minimal changes
in separability, while alignment measures improve in a consistent manner as the trend curves are
shifted versions of one another. Effective dimension also decreases steadily, suggesting a higher
concentration of hidden states along label-unembedding directions. Figure 3 (B) similarly shows that
the accuracy gains from kNN-selected demonstrations correspond to stronger alignment and lower
effective dimension. In Figure 3 (C), the accuracy drop from symbolic label replacement aligns with
a flattened post-transition rise in alignment measures, resembling the case of zero-shot.

## 5.2 Semantic interpretation of alignment dynamics

To understand how the surge in directional alignment after the phase transition leads to high output
alignment and successful decoding of task-related labels at the final layer, we conduct a case study
on the SST-2 dataset to interpret hidden-state alignment dynamics.

**Surge in alignment concurs with the encoding of label-relevant semantics** First, we compute
output alignment (logit lens accuracy) across all layers and compare its trajectory to directional
alignment measures. As shown in Figure 5, their strong correlation indicates that output alignment
rises in sync with directional alignment. This suggests that as alignment improves, label-relevant

semantics are injected into hidden states via consecutive layer updates, enabling correct label decoding even at intermediate layers. To support this, we decode the top right singular vectors of hidden states near the transition point. Figure 5 (B) reveals a sharp semantic shift: while layer 20 produces irrelevant tokens, the SST-2 label token negative emerges as the second most probable token in layer 21—immediately post-transition. This confirms that post-transition layers elevate label-relevant directions in both the semantic and variance structure of the hidden states.

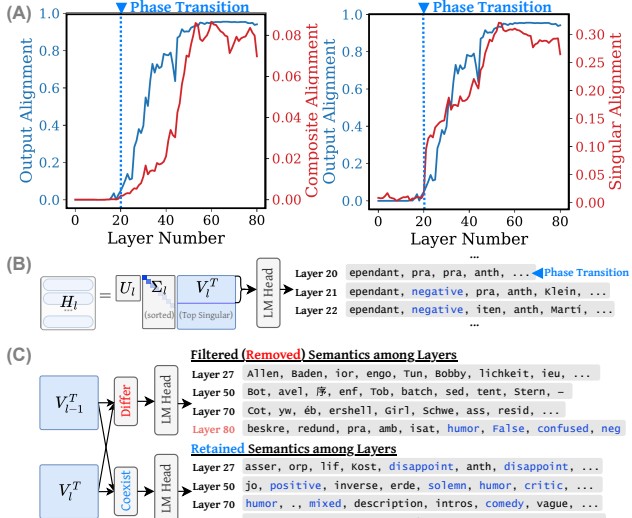

Figure 5: Dynamics of alignment measures and semantics of post-transition hidden states. **(A)** Strong correlation between output and directional alignment; **(B)** Surge in alignment measures concur with encoding of label-related semantics; **(C)** Post-transition layers, except for the last ones, filter out unrelated semantics and retain the related ones. Refer to Appendix G for more semantics decoding cases.

**Layer-wise pattern of semantic retention and filtering** To better understand how exactly layer updates after the phase transition encode relevant semantics into hidden states, we analyze the semantic information retained or removed by each layer. Given the SVDs of the centered hidden states at two consecutive layers $\bar{H}_{l-1} = U_{l-1}\Sigma_{l-1}V_{l-1}^\top$ and $\bar{H}_l = U_l\Sigma_l V_l^\top$, we compute the projection of the previous layer's singular directions onto the next layer's as $V_{l-1}^\top V_l V_l^\top \in \mathbb{R}^{n \times d}$. The rows of this matrix with larger norms are the principal directions of $H_{l-1}$ that are retained in $H_l$, while those with smaller norms indicate filtered directions. By decoding these directions, we identify the semantic information retained or removed by each layer. The results are reported in Figure 5 (C). In most layers after the phase transition, label-related semantic directions are consistently retained, and unrelated directions are filtered, leading to a progressive amplification of label-relevant information in the hidden states and a suppression of the irrelevant semantics. However, in the final layers, this pattern reverses: label-related directions are filtered, and label-irrelevant semantics are reintroduced. This shift corresponds to the slight drop in output alignment in the final layers in Figure 5 (A), and aligns with prior findings that late layers in LLMs tend to encode high-frequency, semantically uninformative tokens [Sharma et al., 2023], which may interfere with the expression of task-relevant content. The same phenomenon also occurs in other datasets as presented in Appendix G.4.

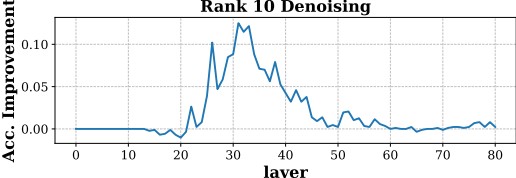

Figure 6: Accuracy gains of rank-10 denoising. See Appendix G.5 for results in other settings.

**Low-rank denoising enhances label-relevant semantics** To better understand the retention-and-filtering process, we test whether filtering can be accelerated—and label-relevant semantics enhanced—by denoising the hidden states. Specifically, we apply a rank-10 SVD approximation to the centered hidden states $\bar{H}$, retaining only the top 10 directions of variation and adding the mean back. This denoising should theoretically remove less informative components and amplify label-relevant semantics. We then feed the denoised hidden states into the unembedding layer and measure output accuracy. As shown in Figure 6, low-rank denoising yields accuracy gains exceeding 10%, especially in early post-transition layers where irrelevant semantics are just beginning to be filtered. These results validate low-rank denoising and support semantic retention and filtering as a core mechanism of post-transition layers. We explore the practical implication of this in Appendix E.2.

**Task vector properties of post-transition hidden states** As visualized by Figure 1 (D), the simultaneous enhancement of both alignment properties and label-relevant semantics after the phase transition qualifies ICL query hidden states in middle-to-late layers as task vectors [Hendel et al., 2023]. Due to their improved alignment properties, steering or replacing zero-shot hidden states

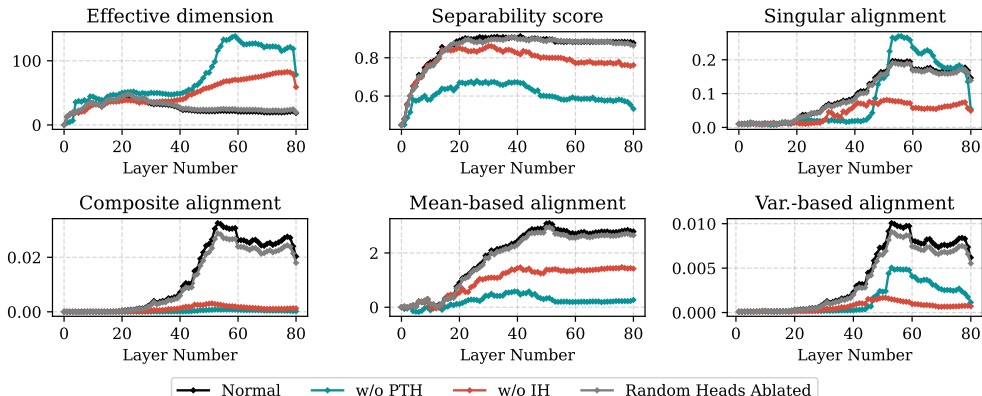

Figure 7: Effects of attention heads ablation on the layer-wise separability and alignment measures. Ablating PTH significantly reduces separability, and ablating IH significantly reduces alignment.

with ICL hidden states can align zero-shot hidden states better with the label unembedding vectors. Similarly, modifying zero-shot hidden states with ICL hidden states, which are label-informative, makes them more suited for the prediction of the correct labels.

## 5.3 Significance of critical attention heads

**PTHs** and **IHs** are attention heads with distinctive attention patterns. Given a token sequence like $[A][B_1][A][B_2]...[A][B_n][A]$, a PTH at the final $[A]$ attends to the immediately preceding $[B_n]$, while an IH attends to tokens such as $[B_1]$–$[B_n]$ following earlier $[A]$ positions. For example, in the ICL prompt "I don't like it. Answer: negative. I like it. Answer:", at the final ":", a PTH attends to the preceding "Answer", while an IH attends to "negative" after the earlier ":", effectively copying semantics of the demonstration label into the query's hidden state. By analyzing their effects on separability and alignment across layers, we gain a more fine-grained understanding of their role in ICL—beyond simply measuring their impact on final predictions.

For each dataset and model, we rank all attention heads by their PTH and IH scores, which quantify how strongly they exhibit corresponding attention patterns on test prompts. We then ablate (zero out) the outputs of the top 10% identified IHs and PTHs and measure the resulting effects on hidden state separability and alignment across layers as well as the output accuracy. As a control, we repeat the procedure with an equal number of randomly selected heads (excluding the top PTH/IH heads). Details on identifying PTHs and IHs are provided in Appendix H.

**PTHs induce separability, IHs induce alignment** The results in Figure 7 confirm that PTHs and IHs are crucial not only for verbalizing ICL outputs [Cho et al., 2025a] but also for structuring hidden states both separability-wise and alignment-wise. Ablating random attention heads has a negligible effect (as shown by overlapping **black** and gray curves). In contrast, ablating PTHs or IHs significantly alters the measures, but with distinct effects. Ablating PTHs substantially reduces separability and also influences mean-based alignment (and thus composite alignment) which concerns the separation margin of the label clusters, but has a minimal impact on other alignment measures. Conversely, ablating IHs leaves separability largely intact but severely disrupts all alignment measures, and has a far greater impact on output alignment as measured by the accuracy.

These results not only confirm a two-stage ICL process—first driven by separability, then by alignment—but also clarify its mechanism. PTHs, concentrated in early layers [Cho et al., 2025a], enhance separability by attending to query tokens and encoding distinct semantics into hidden states. IHs, emerging after a critical depth (30–40% of total layers [Halawi et al., 2024]), copy demonstration label embeddings into query hidden states via the residual stream, improving alignment with label unembedding directions. Furthermore, in Appendix I.5, we show that

Table 1: Dataset-mean layer positions of top IH and PTH heads by percentage levels with statistically significant differences.

| % | IH mean | PTH mean | p-value |
|---|---------|----------|---------|
| 1% | 40.203 | 33.379 | $1.01 \times 10^{-7}$ |
| 2% | 40.884 | 36.459 | $1.30 \times 10^{-6}$ |
| 5% | 42.238 | 36.422 | $2.06 \times 10^{-17}$ |
| 10% | 41.535 | 35.850 | $9.72 \times 10^{-29}$ |

PTHs and IHs contribute to ICL through the same mechanism even when the query's ground-truth label—previously considered essential for IHs [Cho et al., 2025a]—is absent from demonstrations.

**PTHs in early layers, IHs in late layers** To test whether PTHs and IHs align with our two-stage ICL characterization—inducing separability and alignment respectively—we examine their layer distributions. Specifically, we locate the top 1%, 2%, 5%, and 10% identified PTHs and IHs and perform Mann-Whitney U tests to assess differences across layers. As shown in Table 1, PTHs mainly occur in earlier layers, consistent with their role in inducing separability, while IHs appear later, with the differences being highly significant under the Mann-Whitney test.

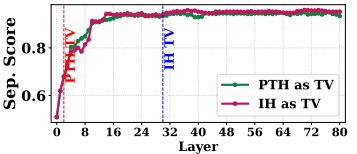

(a) Layer-wise separability score: PTH task vector

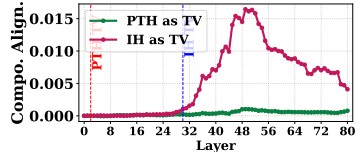

(b) Layer-wise composite alignment: IH task vector

Figure 8: **(A)** PTH task vector increases separability after injection; **(B)** Injecting IH task vector recreates the phase-transition in alignment.

**Geometric effects of IH and PTH outputs as task vectors** The ablation in Figure 7, though revealing how removing PTHs and IHs affects separability and alignment, does not show how they actively drive the geometric evolution of hidden states during inference.

To address this, we follow Todd et al. [2024] and inject their outputs as task vectors into zero-shot hidden states at selected layers to observe induced metric changes. We construct task vectors using outputs of the top 10% IHs and PTHs as in Appendix J. We inject the PTH-derived vector at layer $l = 2$ to track separability, and the IH-derived vector at $\frac{3}{8}L$ to track composite alignment. These layers correspond to where separability and alignment begin to shift markedly in Figure 2. Figure 8a shows that injecting PTH outputs as task vectors boosts separability (the green line surpassing the magenta line before convergence; see Appendix I.2 for similar trends across models), while injecting IH task vectors restores the ICL-specific spike in alignment metrics in zero-shot hidden states.

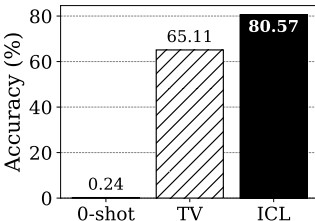

Figure 9: Average effect across datasets of steering zero-shot hidden states using IH outputs as task vectors.

**IH outputs as task vectors** Since we have shown that IH outputs with ICL demonstrations can compensate for alignment deficits in hidden states, we have clarified how IH outputs as task vectors influence the forward computation of models: they steer hidden states to align with label-token unembeddings so these tokens can be decoded. This also explains why IH outputs themselves decode label-related tokens [Todd et al., 2024], as they must align with label unembedding directions to perform such steering. To validate this conclusion on our classification datasets (not considered in the original study), we extract task vectors from identified IH outputs and add them to layer-30 zero-shot hidden states of Llama2-70B. The substantial accuracy gains in Figure 9 support this conclusion.

## 6 Conclusions and Limitations

**Conclusions** This work unifies two major perspectives on in-context learning (ICL)—the roles of special attention heads and task vectors—within a geometric framework centered on separability and alignment of hidden states. We theoretically show that these two properties fully determine ICL classification accuracy and empirically demonstrate that ICL across classification and generation tasks improves accuracy over zero-shot mainly by enhancing alignment, explaining why ICL query hidden states serve effectively as task vectors. Our analysis reveals a phase transition: early layers boost separability, while later layers refine alignment with label unembedding vectors by filtering out label-irrelevant semantics and preserving task-relevant directions, coinciding with the emergence of label semantics in hidden states. Zooming into layers, we identify complementary roles of **PTHs** and **IHs**: PTHs in early layers enhance separability, while IHs emerging post-transition promote alignment—clarifying both the phase transition and the effectiveness of IH outputs as task vectors.

**Limitations** Our analysis of the ICL hidden states of generation tasks is preliminary. In addition, our analysis focuses on inference-time behavior in pretrained models. Future work should investigate how the phase transition pattern of ICL hidden states emerges during training, which could shed light on the emergence of ICL capability.

**Acknowledgments**

This work was supported by JST FOREST Program (Grant Number JPMJFR232K, Japan) and the Nakajima Foundation. We used ABCI 3.0 provided by AIST and AIST Solutions with support from "ABCI 3.0 Development Acceleration Use".

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

# Appendices

## A  Proofs related to Theorem 1

We first show that $S^*$ is attained at some point $v^*$ in the hypersphere $\mathbb{S}^{d-1} = \{u : u \in \mathbb{R}^d, \|u\|_2 = 1\}$ though $\mathbb{S}^{d-1}$ is an infinite set and $S(u)$ is not continuous. Observe that each $h_i$ in $H$ partitions $\mathbb{S}^{d-1}$ into two hyperhemispheres. One is $\{u : u \in \mathbb{R}^d, \|u\|_2 = 1, u^T h_i \geq 0\}$ and the other is $\{u : u \in \mathbb{R}^d, \|u\|_2 = 1, u^T h_i < 0\}$. Hence, as long as the number of queries in the dataset $n$ is finite, $h_1, ..., h_n$ will partition $\mathbb{S}^{d-1}$ into a finite number of "cells". Each cell corresponds to the elements in $\mathbb{S}^{d-1}$ that satisfy a certain sign sequence $[\mathrm{sgn}(u^T h_1), ..., \mathrm{sgn}(u^T h_n)] \in \{1, 0, -1\}^n$. Hence, $S(u)$ is constant on each of the cells, and the supremum (or the maximum) value $S^*$ is attained on one of the cells.

We then show the proof of Theorem 1:

*Proof* First, consider the case where $\mathbb{V}$ only consists of $y_A$ and $y_B$. Then $E \in \mathbb{R}^{2 \times d}$ and the two rows are $E_{y_A}$ and $E_{y_B}$. Then we have

$$\text{Acc} = \frac{1}{n} \sum_{i=1}^{n} \mathbb{1}(y_i = \arg\max_{v_j} \frac{E_{v_j} h_i}{\sum_{v'_j \in \{y_A, y_B\}} E_{v'_j} h_i}) \tag{3}$$

$$= \frac{1}{n}\Big(\sum_{i \in \mathbb{N}_1} \mathbb{1}(E_{y_A} h_i \geq E_{y_B} h_i) + \sum_{i \in \mathbb{N}_2} \mathbb{1}(E_{y_A} h_i < E_{y_B} h_i) \qquad \text{definition of argmax} \tag{4}$$

$$= \frac{1}{n}\Big(\sum_{i \in \mathbb{N}_1} \mathbb{1}[(E_{y_A} - E_{y_B}) h_i \geq 0] + \sum_{i \in \mathbb{N}_2} \mathbb{1}[(E_{y_A} - E_{y_B}) h_i < 0] \qquad \text{reorganizing terms} \tag{5}$$

$$= \frac{1}{n}\Big(\sum_{i \in \mathbb{N}_1} \mathbb{1}[(\frac{E_{y_A} - E_{y_B}}{\|E_{y_A} - E_{y_B}\|_2}) h_i \geq 0] + \sum_{i \in \mathbb{N}_2} \mathbb{1}[(\frac{E_{y_A} - E_{y_B}}{\|E_{y_A} - E_{y_B}\|_2}) h_i < 0] \qquad \text{linearity} \tag{6}$$

$$\leq \sup_{u \in \mathbb{R}^d, \|u\|_2 = 1} \frac{1}{n}\Big(\sum_{i \in \mathbb{N}_1} \mathbb{1}(u^T h_i \geq 0) + \sum_{i \in \mathbb{N}_2} \mathbb{1}(u^T h_i < 0)\Big) \qquad \text{supremum} \tag{7}$$

$$= S^*. \tag{8}$$

Here, when $E_{y_A} h_i = E_{y_B} h_i$, without loss of generality, we break the tie by letting the predicted label to be $y_A$ for $x_i$.

When there are more tokens in $\mathbb{V}$, it holds true that $\mathbb{1}(y_i = \arg\max_{v_j} \frac{E_{v_j} h_i}{\sum_{v'_j \in \{y_A, y_B\}} E_{v'_j} h_i}) \leq \mathbb{1}(y_i = \arg\max_{v_j} \frac{E_{v_j} h_i}{\sum_{v'_j \in 1, ..., |\mathbb{V}|} E_{v'_j} h_i})$. Hence, we have $Acc = \frac{1}{n} \sum_{i=1}^{n} \mathbb{1}(y_i = \arg\max_{v_j} \frac{E_{v_j} h_i}{\sum_{v'_j \in 1, ..., |\mathbb{V}|} E_{v'_j} h_i}) \leq S^*$.

By the definition of $u^*$, we have

$$\frac{1}{n}\Big(\sum_{i \in \mathbb{N}_1} \mathbb{1}(u^{*,T} h_i \geq 0) + \sum_{i \in \mathbb{N}_2} \mathbb{1}(u^{*,T} h_i < 0)\Big) = S^*. \tag{9}$$

If $\frac{E_{y_A} - E_{y_B}}{|E_{y_A} - E_{y_B}|_2} = c u^*$ for some positive constant $c$, then

$$\frac{1}{n}\Big(\sum_{i \in \mathbb{N}_1} \mathbb{1}[(E_{y_A} - E_{y_B})^T h_i \geq 0] + \sum_{i \in \mathbb{N}_2} \mathbb{1}[(E_{y_A} - E_{y_B})^T h_i < 0]\Big) \tag{10}$$

$$= \frac{1}{n}\Big(\sum_{i \in \mathbb{N}_1} \mathbb{1}(u^{*,T} h_i \geq 0) + \sum_{i \in \mathbb{N}_2} \mathbb{1}(u^{*,T} h_i < 0)\Big). \tag{11}$$

We already showed that

$$\frac{1}{n} \sum_{i=1}^{n} \mathbb{1}(y_i = \arg\max_{v_j} \frac{E_{v_j} h_i}{\sum_{v'_j \in \{y_A, y_B\}} E_{v'_j} h_i}) \tag{12}$$

$$= \frac{1}{n}\Big(\sum_{i \in \mathbb{N}_1} \mathbb{1}[(E_{y_A} - E_{y_B})^T h_i \geq 0] + \sum_{i \in \mathbb{N}_2} \mathbb{1}[(E_{y_A} - E_{y_B})^T h_i < 0]\Big). \tag{13}$$

When $\max_{v \in \mathbb{V}, v \notin y_A, y_B} \boldsymbol{E}_v \boldsymbol{h}_i < \max(\boldsymbol{E}_{y_A} \boldsymbol{h}_i, \boldsymbol{E}_{y_B} \boldsymbol{h}_i), \forall i$, we have

$$\frac{1}{n} \sum_{i=1}^{n} \mathbb{1}(y_i = \arg\max_{v_j} \frac{\boldsymbol{E}_{v_j} \boldsymbol{h}_i}{\sum\limits_{v_j' \in \{y_A, y_B\}} \boldsymbol{E}_{v_j'} \boldsymbol{h}_i}) \tag{14}$$

$$= \frac{1}{n} \sum_{i=1}^{n} \mathbb{1}(y_i = \arg\max_{v_j} \frac{\boldsymbol{E}_{v_j} \boldsymbol{h}_i}{\sum\limits_{v_j \in 1, \ldots, |\mathbb{V}|} \boldsymbol{E}_{v_j'} \boldsymbol{h}_i}) \tag{15}$$

$$= \text{Acc.} \tag{16}$$

Hence, we have $\text{Acc} = S^*$ when $\max_{v \in \mathbb{V}, v \notin y_A, y_B} \boldsymbol{E}_v \boldsymbol{h}_i < \max(\boldsymbol{E}_{y_A} \boldsymbol{h}_i, \boldsymbol{E}_{y_B} \boldsymbol{h}_i), \forall i$ and $\frac{\boldsymbol{E}_{y_A} - \boldsymbol{E}_{y_B}}{\|\boldsymbol{E}_{y_A} - \boldsymbol{E}_{y_B}\|_2} = c\boldsymbol{u}^*$.
$\square$

# B  Generalization of Theorem 1

In this section[5], we consider a generalization of Theorem 1 where $x_1, \ldots, x_n$ are allowed to have $m < n$ labels, i.e. $y_i \in \{y_1, \ldots, y_m\}, \forall i$. Same as before, let $\mathbb{N}_j = \{i : y_i = y_j\}, j = 1, \ldots, m$. $\mathbb{N}_j, j = 1, \ldots, m$ are pairwise disjoint and $\bigcup_{j=1}^{m} |\mathbb{N}_j| = \{1, \ldots, n\}$. Let $S_{j,k}(\boldsymbol{u}) = \frac{1}{|\mathbb{N}_j| + |\mathbb{N}_k|} (\sum_{i \in \mathbb{N}_j} \mathbb{1}(\boldsymbol{u}^T \boldsymbol{h}_i \geq 0) + \sum_{i \in \mathbb{N}_k} \mathbb{1}(\boldsymbol{u}^T \boldsymbol{h}_i < 0))$ and let $\boldsymbol{u}_{j,k}^* = \arg\max_{\boldsymbol{u} \in \mathbb{S}^{d-1}} S_{j,k}(\boldsymbol{u})$ for $j, k \in 1, \ldots, m$, and $S_{j,k}(\boldsymbol{u}_{j,k}^*) = S_{j,k}^*$. For each $j$ and $k$, $\boldsymbol{u}_{j,k}^*$ is guaranteed to be in $\mathbb{S}^{d-1}$ because only $|\mathbb{N}_j| + |\mathbb{N}_k| < n$ hyperplanes are involved in separating separating $\mathbb{S}^{d-1}$, resulting in a smaller number of cells (guaranteed to be finite) compared to the case of $x_1, \ldots, x_n$ having only $y_1, y_2$ as labels. Then, we have the following result.

**Theorem 2**  $\text{Acc} \leq \frac{1}{n(m-1)} (\sum_{k=1}^{m} \sum_{j=k+1}^{m} (|\mathbb{N}_k| + |\mathbb{N}_j|) S_{j,k}^*)$. *The equality is achieved when for all unordered pairs $(j,k), j, k \in 1, \ldots, m$, $\max_{v \in \mathbb{V}, v \notin \{y_j, y_k\}} \boldsymbol{E}_v \boldsymbol{h}_i < \max(\boldsymbol{E}_{y_j} \boldsymbol{h}_i, \boldsymbol{E}_{y_k} \boldsymbol{h}_i), \forall i \in N_j \cup N_k$ and $\boldsymbol{E}_{y_j} - \boldsymbol{E}_{y_k} = c_{j,k} \boldsymbol{u}_{j,k}^*$ for some positive constant $c_{j,k}$.*

The theorem states that in the case of the queries having more than two labels, the dataset classification accuracy is controlled by the **pairwise separability and alignment** properties of the hidden states. After calculating the separability of the hidden states of queries with labels $y_j$ and $y_k$ for each $(j,k)$, the dataset classification accuracy is then upper-bounded by these separability values weighted by the number of queries with different labels.

Similarly, the accuracy can match this upper bound if each pairwise hidden states collection is maximally separated along a direction parallel to the difference of the unembedding vectors of the corresponding labels. This theorem shows that the otherwise difficult multiclass classification problem is geometrically equivalent to the combination of a series of binary classification subproblems. And the geometric properties of the multiclass classification problem can be understood by studying the much more tractable geometry of the constituent binary subproblems.

---

[5]Here, we focus on the multiway classification, thus we change our notation to label categories from $y_A$ and $y_B$ into $y_m$, where $m \in \mathbb{Z}^+$.

*Proof* First, note the following relations

$$\text{Acc} = \frac{1}{n}\sum_{i=1}^{n} \mathbb{1}(y_i = \arg\max_{v_j} \frac{\boldsymbol{E}_{v_j}\boldsymbol{h}_i}{\sum\limits_{j'\in 1,\ldots,|\mathbb{V}|}\boldsymbol{E}_{v_j'}\boldsymbol{h}_i})$$

$$\leq \frac{1}{n}\sum_{i=1}^{n} \mathbb{1}(y_i = \arg\max_{v_j} \frac{\boldsymbol{E}_{v_j}\boldsymbol{h}_i}{\sum\limits_{v_j'\in\{y_1,\ldots,y_k\}}\boldsymbol{E}_{v_j'}\boldsymbol{h}_i})$$

$$= \frac{1}{n}(\sum_{k=1}^{m}\sum_{x_i\in\mathbb{N}_k} \mathbb{1}(\boldsymbol{E}_{y_k}\boldsymbol{h}_i \geq \boldsymbol{E}_{y_j}\boldsymbol{h}_i, \forall j \neq k, j \in 1,\ldots,m))$$

$$= \frac{1}{n}(\sum_{k=1}^{m}\sum_{x_i\in\mathbb{N}_k} \min_{j\neq k, j\in 1,\ldots,m}\mathbb{1}(\boldsymbol{E}_{y_k}\boldsymbol{h}_i \geq \boldsymbol{E}_{y_j}\boldsymbol{h}_i)) \qquad \text{indicator function is either 1 or 0}$$

$$\leq \frac{1}{n}(\sum_{k=1}^{m}\sum_{x_i\in\mathbb{N}_k} \frac{1}{m-1}\sum_{j,j\neq k, j\in 1,\ldots,m}\mathbb{1}(\boldsymbol{E}_{y_k}\boldsymbol{h}_i \geq \boldsymbol{E}_{y_j}\boldsymbol{h}_i))$$

$$= \frac{1}{n(m-1)}(\sum_{k=1}^{m}\sum_{j,j\neq k, j\in 1,\ldots,m}\sum_{x_i\in\mathbb{N}_k}\mathbb{1}(\boldsymbol{E}_{y_k}\boldsymbol{h}_i \geq \boldsymbol{E}_{y_j}\boldsymbol{h}_i))$$

$$= \frac{1}{n(m-1)}(\sum_{k=1}^{m}\sum_{j,j\neq k, j\in 1,\ldots,m}\sum_{x_i\in\mathbb{N}_k}\mathbb{1}(\frac{\boldsymbol{E}_{y_k}-\boldsymbol{E}_{y_j}}{\|\boldsymbol{E}_{y_k}-\boldsymbol{E}_{y_j}\|_2}\boldsymbol{h}_i \geq 0)).$$

Note that the sum is over all ordered pairs $(j,k)$ s.t. $j,k \in 1,\ldots,m, j \neq k$. Consider the case of $(k=1, j=2)$ and $(k=2, j=1)$, then both $\mathbb{1}(\boldsymbol{E}_{y_k}\boldsymbol{h}_i \geq \boldsymbol{E}_{y_j}\boldsymbol{h}_i)$ and $\mathbb{1}(\boldsymbol{E}_{y_j}\boldsymbol{h}_i \geq \boldsymbol{E}_{y_k}\boldsymbol{h}_i))$ appear in the sum. Hence, the sum can be taken over all unordered pairs $(j,k)$ and rewritten as:

$$\frac{1}{n(m-1)}(\sum_{k=1}^{m-1}\sum_{j=k+1}^{m}\sum_{x_i\in\mathbb{N}_k\cup\mathbb{N}_j}(\mathbb{1}(\frac{\boldsymbol{E}_{y_k}-\boldsymbol{E}_{y_j}}{\|\boldsymbol{E}_{y_k}-\boldsymbol{E}_{y_j}\|_2}\boldsymbol{h}_i \geq 0) + \mathbb{1}(\frac{\boldsymbol{E}_{y_k}-\boldsymbol{E}_{y_j}}{\|\boldsymbol{E}_{y_k}-\boldsymbol{E}_{y_j}\|_2}\boldsymbol{h}_i < 0))) \qquad (17)$$

$$\leq \frac{1}{n(m-1)}(\sum_{k=1}^{m-1}\sum_{j=k+1}^{m}(|\mathbb{N}_k|+|\mathbb{N}_j|)S_{j,k}^*). \qquad (18)$$

Hence, we have

$$\text{Acc} \leq \frac{1}{n(m-1)}(\sum_{k=1}^{m-1}\sum_{j=k+1}^{m}(|\mathbb{N}_k|+|\mathbb{N}_j|)S_{j,k}^*). \qquad (19)$$

As a proof of concept, plug-in $m=2$, then the bound becomes $\frac{1}{n}(|N_1|+|N_2|)S^* = S^*$ since $|N_1|+|N_2| = n$, which matches the result in [Theorem 1](#).

To establish the equality conditions, as in the proof of [Theorem 1](#), if $\boldsymbol{E}_{y_j}-\boldsymbol{E}_{y_k} = c_{j,k}\boldsymbol{u}_{j,k}^*$ for some positive constant $c_{j,k}$, then

$$\sum_{i\in\mathbb{N}_k}\mathbb{1}[(\boldsymbol{E}_{y_k}-\boldsymbol{E}_{y_j})^T\boldsymbol{h}_i \geq 0] + \sum_{i\in\mathbb{N}_j}\mathbb{1}[(\boldsymbol{E}_{y_k}-\boldsymbol{E}_{y_j})^T\boldsymbol{h}_i < 0] \qquad (20)$$

$$= \sum_{i\in\mathbb{N}_k}\mathbb{1}(\boldsymbol{u}_{j,k}^{*,T}\boldsymbol{h}_i \geq 0) + \sum_{i\in\mathbb{N}_j}\mathbb{1}(\boldsymbol{u}_{j,k}^{*,T}\boldsymbol{h}_i < 0) = (|N_j|+|N_k|)S_{j,k}^*. \qquad (21)$$

We also know that if $\max_{v\in\mathbb{V},v\notin y_j,y_k}\boldsymbol{E}_v\boldsymbol{h}_i < \max(\boldsymbol{E}_{y_j}\boldsymbol{h}_i, \boldsymbol{E}_{y_k}\boldsymbol{h}_i), \forall i \in N_j\cup N_k$, then

$$\sum_{i\in\mathbb{N}_k}\mathbb{1}[(\boldsymbol{E}_{y_k}-\boldsymbol{E}_{y_j})^T\boldsymbol{h}_i \geq 0] + \sum_{i\in\mathbb{N}_j}\mathbb{1}[(\boldsymbol{E}_{y_k}-\boldsymbol{E}_{y_j})^T\boldsymbol{h}_i < 0] \qquad (22)$$

$$= \sum_{i\in\mathbb{N}_k\cup\mathbb{N}_j}\mathbb{1}(y_i = \arg\max_{v_j}\frac{\boldsymbol{E}_{v_j}\boldsymbol{h}_i}{\sum_{v_j'\in\{y_j,y_k\}}\boldsymbol{E}_{v_j'}\boldsymbol{h}_i}) \qquad (23)$$

$$= \sum_{i\in\mathbb{N}_k\cup\mathbb{N}_j}\mathbb{1}(y_i = \arg\max_{v_j}\frac{\boldsymbol{E}_{v_j}\boldsymbol{h}_i}{\sum\limits_{v_j\in 1,\ldots,|\mathbb{V}|}\boldsymbol{E}_{v_j'}\boldsymbol{h}_i}). \qquad (24)$$

Finally, note that

$$\text{Acc} = \frac{1}{n} \sum_{i=1}^{n} \mathbb{1}\left(y_i = \arg\max_{v_j} \frac{E_{v_j} h_i}{\sum_{v'_j \in \{y_1, y_2\}} E_{v'_j} h_i}\right) \tag{25}$$

$$= \frac{1}{n} \frac{1}{m-1} \sum_{k=1}^{m-1} \sum_{j=l+1}^{m} \sum_{i \in \mathbb{N}_j \cup \mathbb{N}_k} \mathbb{1}\left(y_i = \arg\max_{v_j} \frac{E_{v_j} h_i}{\sum_{v'_j \in 1,\dots,|\mathbb{V}|} E_{v'_j} h_i}\right). \tag{26}$$

because for each $\mathbb{N}_j$, the term $\mathbb{1}\left(y_i = \arg\max_{v_j} \frac{E_{v_j} h_i}{\sum_{v'_j \in 1,\dots,|\mathbb{V}|} E_{v'_j} h_i}\right)$ appears $(m-1)$ times in the summation for each $i \in \mathbb{N}_j$. Hence, combining everything, we have

$$\text{Acc} = \frac{1}{n} \frac{1}{m-1} \sum_{k=1}^{m-1} \sum_{j=k+1}^{m} \sum_{i \in \mathbb{N}_j \cup \mathbb{N}_k} \mathbb{1}\left(y_i = \arg\max_{v_j} \frac{E_{v_j} h_i}{\sum_{v'_j \in 1,\dots,|\mathbb{V}|} E_{v'_j} h_i}\right) \tag{27}$$

$$= \frac{1}{n(m-1)} \sum_{k=1}^{m-1} \sum_{j=k+1}^{m} (|N_j| + |N_k|) S_{j,k}^*. \tag{28}$$

The proof is thus complete. $\qquad\square$

## C  Calculation of geometric measures

1. **Separability score** We train the logistic classifier on half of the points in $H$ and take its prediction accuracy on the other half as the estimated maximum separability of $H$. We use the scikit-learn [Pedregosa et al., 2011] default implementation of the logistic classifier, and set the number of iterations to be 100.

2. **Output alignment** We feed $H$ into the output unembedding layer of the model to get the model's predictions of the query labels based on $H$ and calculate the percentage of correct predictions.

3. **Singular alignment** First obtain the centered version of $H$, denoted as $\bar{H}$. Then calculate its singular value decomposition, namely $\bar{H} = U \Sigma V^T$, and compute the absolute values of cosine similarities between the top-r singular vectors and the difference of label unembedding vectors $E_{y_A} - E_{y_B}$. The absolute value is taken since the sign of the singular vectors is arbitrary. In practice, we calculate the values using the top-2 singular vectors and report the maximum of the two.

4. **Variance-based alignment** First normalize $E_{y_A} - E_{y_B}$ as $\frac{E_{y_A} - E_{y_B}}{\|E_{y_A} - E_{y_B}\|_2}$. Then the ratio of variance (w.r.t. its mean) of $H$ explained along $\frac{E_{y_A} - E_{y_B}}{\|E_{y_A} - E_{y_B}\|_2}$ is $\frac{E_{y_A} - E_{y_B}}{\|E_{y_A} - E_{y_B}\|_2}^T \frac{\bar{H}^T \bar{H}}{n} \frac{E_{y_A} - E_{y_B}}{\|E_{y_A} - E_{y_B}\|_2}$, which we take to be the variance-based alignment of $\bar{H}$.

5. **Mean-based alignment** First compute the means of label clusters (subsets of $H$ with each of the two different labels), i.e. $\frac{1}{|\mathbb{N}_A|} \sum_{i \in \mathbb{N}_A} h_i$ and $\frac{1}{|\mathbb{N}_B|} \sum_{i \in \mathbb{N}_B} h_i$. Then take the difference and compute the value of its projection onto $\frac{E_{y_A} - E_{y_B}}{\|E_{y_A} - E_{y_B}\|_2}$, i.e. $\left(\frac{E_{y_A} - E_{y_B}}{\|E_{y_A} - E_{y_B}\|_2}\right)^T \left(\frac{1}{\mathbb{N}_A} \sum_{i \in |\mathbb{N}_A|} h_i - \frac{1}{|\mathbb{N}_B|} \sum_{i \in \mathbb{N}_B} h_i\right)$. Then, calculate the weighted average of the variances of the two label clusters along $\frac{E_{y_A} - E_{y_B}}{\|E_{y_A} - E_{y_B}\|_2}$, i.e. $\frac{|\mathbb{N}_A| \text{Var}(\{h_i^T \frac{E_{y_A} - E_{y_B}}{\|E_{y_A} - E_{y_B}\|_2} : i \in \mathbb{N}_A\}) + |\mathbb{N}_B| \text{Var}(\{h_i^T \frac{E_{y_A} - E_{y_B}}{\|E_{y_A} - E_{y_B}\|_2} : i \in \mathbb{N}_B\})}{|\mathbb{N}_A| + |\mathbb{N}_B|}$. Finally, divide the projected difference between means by the square root of the weighted average of the projected variance.

6. **Composite alignment** Multiply the variance-based alignment by the mean-based alignment.

7. **Effective dimension** Given the SVD $\bar{H} = U \Sigma V^T$ and the singular values $\sigma_1, \dots, \sigma_n$, the effective dimension of $H$ is $\frac{(\sum_{i=1}^{n} \sigma_i^2)^2}{\sum_{i=1}^{n} \sigma_i^4}$.

Table 2: Prompt templates and labels for different datasets.

| Dataset | Template | Label |
|---------|----------|-------|
| SST-2 | `{Sentence}` **Sentiment:** `{Label}` | positive / negative |
| SUBJ | `{Sentence}` **Type:** `{Label}` | subjective / objective |
| TREC | Question: `{Sentence}` **Type:** `{Label}` | abbreviation / entity / description / human / location / number |
| SNLI | The question is: `{Premise}`? The answer is: `{Hypothesis}` `{Label}` | true / maybe / false |
| RTE | The question is: `{Premise}`? The answer is: `{Hypothesis}` `{Label}` | true / false |
| CB | The question is: `{Premise}`? The answer is: `{Hypothesis}` `{Label}` | true / maybe / false |
| Famous People | `{Person Name}` is a: `{Label}` | actor / politician / singer / scientist / writer / athlete |

Table 3: Mappings used to replace ground-truth labels of different datasets to symbols

| Dataset | Label Mapping |
|---------|---------------|
| SST-2 | negative/positive $\rightarrow$ @/# |
| SUBJ | objective/subjective $\rightarrow$ @/# |
| TREC | abbreviation / entity / description / person / number / location / $\rightarrow$ @/#/!/$/&/* |
| SNLI | true/maybe/false $\rightarrow$ @/#/! |
| RTE | true/false $\rightarrow$ @/# |
| CB | true/maybe/false $\rightarrow$ @/#/! |

# D   Implementation details

**Models**   We use the official huggingface implementation of all models. All models with more than 10B parameters are quantized to 4bit.

**Datasets**   We use the official huggingface implementation of all datasets except for the Famous People dataset crafted by ourselves. The Famous People dataset is generated using ChatGPT-4o [OpenAI et al., 2024]. The dataset has 180 datapoints, each is the name of a famous person with a profession label from one of six categories: actor, politician, singer, scientist, writer, athlete. The dataset is balanced as there are 30 data points for each label.

**ICL setting**   For each dataset except for the Famous People dataset, we select demonstrations from the train set and queries from the test set, or the validation set if the ground-truth labels for the test set are not provided. We keep only the first 10000 datapoints in the train set for demonstration selection if the train set has more than 10000 entries, and the first 1000 data points in the test set/validation set for accuracy evaluation. For the famous people dataset, we use the first 90 data points for demonstration selection and the remaining 90 data points for testing. For experiments involving kNN demonstration selection, we first use the respective LLMs to encode the demonstrations and queries, and select the demonstrations with embeddings closer to that of the query in $l_2$ distance for each query. We use the prompt formats detailed in Table 2.

**Devices**   All experiments with Llama2-7B, Llama2-13B, Gemma-2B, and Gemma-7B are conducted with an A800 GPU. All experiments with Llama3-8B, Llama2-70B, and Llama3-70B are conducted with an H200 GPU.

For the label mapping experiment in Subsection 5.1. We use the mappings listed in Table 3 to map the ground-truth labels of each dataset to symbols.

# E   Practical implications of the mechanistic findings

We discuss in this section how our geometric characterization of the model's inference process in the zero-shot or ICL setting can be converted to actionable and efficient methods to improve model performance.

## E.1   Improving classification by capitalizing on hidden states separability

In Figure 2 we see the high separability of hidden states of prompts from classification tasks which surpasses the accuracy of the actual label prediction by far, irrespective of whether demonstrations are

prepended to the labels. This implies that rather than direct decoding we can instead use the model as a feature extractor with which to train classifiers to perform the classification task, which is exactly how we evaluate the separability score. To demonstrate this, we train a logistic classifier on 50% of the last layer hidden states and evaluate the classification accuracy on the rest half. The results in Table 22 demonstrate that such a practice improves the classification accuracy with zero-shot prompts by over 70% across models, while only incurring the minimal cost of training an extra logistic classifier. Moreover, in Table 23 we see that training the classifier over 8-shot ICL hidden states also brings a solid ∼10% increase over the original ICL accuracy, which shows the generalizability of this lightweight method that leverages hidden states separability.

## E.2 Unsupervised enhancement of classification performance through low-rank denoising

In Figure 6 we see that low-rank denoising of the ICL hidden states matrix reveals the task-related directions as its dominant directions and enhances its alignment properties. We can thus apply this technique to the final layer hidden states instead of the intermediate layer ones (which is the focus of Figure 6) to directly improve classification performance. In Table 24 we apply rank-5 denoising to final layer hidden states before feeding them into the unembedding layer. This method is different from the classifier-based method in that it assumes no knowledge of the ground-truth label of each hidden state. The dataset-average results demonstrate that the rank-5 denoising can indeed improve the classification performance of all models with the only exception of Gemma-7B.

# F   Supplementary materials for Subsection 5.1

## F.1   Replication of Figure 2 for other models

In Figure 10-15, we provide the visualizations of the experiments presented in Figure 2 for other models, from which similar conclusions to those in Subsection 5.1 can be drawn. Alignment measures explain the accuracy difference between ICL and zero-shot across models, and the layer-wise trends of the measures explain the phase transition pattern for all models.

## F.2   Replication of Figure 3 for other models

In Figure 16-21 we provide the visualizations of the experiments presented in Figure 3 for other models to establish the generality of our conclusions. It is clear that the phase transition pattern is evident in different ICL settings across models, and alignment measures capture accuracy differences across ICL settings for all models.

## F.3   Replication of Figure 4 for other models

In Figure 53-58 we repeat the visualizations in Figure 4 for other models to examine where the generation task fits well into our framework. The results demonstrate that for other models, the phase-transition pattern in the composite alignment metric is specific to the 8-shot ICL hidden states, while ICL and zero-shot hidden states do not differ much in terms of the dynamics of the separability score.

# G   Supplementary materials for Subsection 5.2

## G.1   Replication of subplot (A) of Figure 5 for other datasets and models

In Figure 22- 28 we provide the visualizations of the experiments presented in subplot (A) of Figure 5, to establish the generalizability of our conclusion that the directional alignment and the encoding of label-related semantics into the hidden states happen almost simultaneously after the phase transition. The layer-wise trends of the directional alignment measures and output alignment exhibit high correlation across all datasets and models.

### G.2  Replication of subplot (B) of Figure 5 for other datasets

In Table 4-8, we present the tokens decoded from top singular directions near the phase transition breakpoint for other datasets. The results demonstrate that the sudden .semantic shift toward label-related tokens in the hidden states (as demonstrated by subplot (B) of Figure 5) is present in other datasets as well. Note that the phase transition happens at different layers for different datasets.

### G.3  Replication of subplot (B) of Figure 5 with replaced demonstration labels

To further examine the relationship between alignment and the encoding of task-related semantics, we design a scenario in which the ground-truth demonstration labels are consistently mapped to symbols, and the expected predictions for the query label becomes the mapped symbols as well. This forces a separation between semantic encoding and alignment, as the task-related semantics remain unchanged, but alignment must now occur with respect to the unembedding vectors of the symbols. Specifically, we conduct experiments on SST-2 and SNLI where we map the SST-2 labels (negative/positive) to @ and #, and the SNLI labels (true/maybe/false) to @, #, and ! within the demonstrations. We then inspect the tokens decoded from the top singular directions of the hidden states at each layer.

As shown in Table 9, the results reveal an interesting pattern. The first major shift in the layer-wise semantic content occurs at layer 24, where the ground-truth label tokens suddenly become dominant. This closely mirrors the pattern observed with normal demonstration labels in Table 6, and aligns with the point where output alignment begins to rise in Figure 22 (C). However, another drastic shift is observed at layer 45, where the decoded top tokens begin to include the symbol tokens. These symbols suddenly appear and start to compete with semantically relevant task tokens, marking the onset of alignment between the hidden states and the unembedding vectors of the symbols. This delayed alignment corresponds with the pattern seen in subplot (C) of Figure 3, where the phase transition under demonstrations with symbolized labels is postponed to later layers.

The competition between the true labels and the injected symbol labels persists through to the final layers. This likely contributes to the reduced accuracy in this setting, as the semantic influence of the substituted labels fails to fully override that of the ground-truth labels. Similar results for SST-2 are presented in Table 10, showing similar trends.

### G.4  Replication of subplot (C) of Figure 5 for other datasets

Subplot (C) of Figure 5 reveals a two-stage pattern in the layer-wise retainment and filtering of semantic information, where most post-transition layers retain label-related semantics and filter out irrelevant ones, and the final layers do the opposite. We show in Table 11-15 that it exists in other datasets besides SST-2 as well.

### G.5  Replication of Figure 6 for other datasets and models

We first provide the results of rank-10 approximations on all 6 datasets for Llama2-70B in Figure 42. The conclusion is that low-rank denoising significantly increase the output alignment significantly acorss all datasets, particularly in the innitial layers after the phase transition. This confirms the generality of our findings in Subsection 5.2. Additionally, in the final layers, the output alignment also increases on on SST-2, SUBJ, TREC, and SNLI, though not on RTE and CB.

In Figure 43-48, we present results for other models. These are consistent with the findings from Llama2-70B. Specifically, the pattern that low-rank denoising improves output alignment more in text classification datasets than in natural language inference datasets holds across models. This suggests a higher level of linguistic complexity in natural language inference tasks—likely because such tasks require reasoning over both a premise and a hypothesis, whose combined semantics are more difficult to compress and more prone to information loss during low-rank projection.

### G.6  Ablation experiments for the rank hyperparameter in low-rank approximations

In this section, we explore the effect of varying the number of retained ranks in rank-$r$ SVD approximations of Llama2-70B hidden states across all datasets. Specifically, we investigate how increasingly

aggressive rank reductions—setting $r = 1$, 2, and 5—impact performance. The results, shown in Figure 49, Figure 50, and Figure 51, yield several notable findings.

For binary classification tasks such as SST-2, SUBJ, and RTE, more aggressive rank reduction sometimes leads to better output alignment, particularly in layers after the phase transition where alignment with label unembedding vectors begins to emerge. In some cases, even a rank-1 approximation preserves or enhances alignment-related performance. However, for three-way classification tasks like SNLI and RTE, aggressive reductions (especially at $r = 1$ or 2) can cause accuracy drops in later layers. For TREC, which involves six label classes, rank-2 or rank-1 approximations degrade accuracy across all layers.

These results align with the intuition that datasets with more labels require higher-dimensional hidden state representations to preserve the necessary separability and alignment structures among label clusters.

## H  Experiment details concerning the identification of PTHs and IHs

For each dataset, we use the first 50 queries to identify the set of **IH**s and **PTH**s. Denote the queries as $x_1, ..., x_{50}$ each with token length $s(x_1), ..., s(x_{50})$.

**Identification of PTHs**  For each $x_i$, the LLM will generate an attention tensor $\boldsymbol{Attn}_i \in \mathbb{R}^{L \times N_h \times s(x_i) \times s(x_i)}$, where $L$ is the total number of layers, $N_h$ is the number of attention heads per layer. Hence, for the $n_h^{th}$ attention head at layer $l$, there is a corresponding attention matrix $\boldsymbol{Attn}(l, n_h)_i \in \mathbb{R}^{s(x_i) \times s(x_i)}$, where $\boldsymbol{Attn}(l, n_h)_{i,j,k}$ is the attention of this attention head at the $k^{th}$ token to the $j^{th}$ token in $x_i$. The PTH score of this attention head on $x_i$ is thus defined as $\sum_{k=2}^{s(x_i)} \boldsymbol{Attn}(l, n_h)_{i,k-1,k}$, which measures the total attention the attention head assigns at each token position to the previous token. Consequently, the PTH score of the $n_h^{th}$ attention head at layer $l$ on all the queries is $\sum_{i=1}^{50} \sum_{k=2}^{s(x_i)} \boldsymbol{Attn}(l, n_h)_{i,k-1,k}$. We calculate the PTH scores for all $(l, n_h)$ pairs and choose the top 10% attention heads as the identified **PTH**s.

**Identification of IHs**  For each $x_i$, randomly select 8 demonstrations and prepare them to $x_i$. The resultant ICL prompt (denoted as $X_i$ with length $s(X_i)$), with a slight abuse of notation, takes the format of $\langle x_{i,1} \rangle : \langle y_{i,1} \rangle, ..., \langle x_{i,8} \rangle : \langle y_{i,8} \rangle, \langle x_i \rangle :$, where $\langle x_{i,k} \rangle$ represents the sentence part of demonstration $k$ ("I like this movie. Sentiment") and $\langle y_{i,k} \rangle$ the label ("positive") separated from the sentence part by a colon (:), and $\langle x_i \rangle$ is the sentence part for the query. At the position of the final ":", an IH will attend to tokens after the previous ":"s, i.e. the label tokens $\langle y_{i,1} \rangle, ..., \langle y_{i,8} \rangle$. Let $\mathbb{I}_i$ denote the indices of the label tokens in $X_i$. The IH score of the $n_h^{th}$ attention head at layer $l$ over the 50 queries is thus $\sum_{i=1}^{50} \sum_{k \in \mathbb{I}_i} \in \boldsymbol{Attn}(l, n_h)_{i,k,s(X_i)}$, i.e. we are measuring the total attention a head assigns at the final ":" position to the positions of all the label tokens summed over all 50 queries. We calculate the IH scores for all $(l, n_h)$ pairs and choose the top 10% attention heads as the identified **IH**s.

## I  Supplementary materials for Subsection 5.3

### I.1  Replication of Figure 7 for other models

In Figure 29-34, we present the results of attention heads ablations across models. The results support our conclusions regarding the significance of PTHs in enhancing the separability of the ICL hidden states and the significance of IHs in aligning them with the label unembedding vectors.

### I.2  Replication of Figure 8 for other models

In Figure 59-64, we present the results of analyzing the geometric contribution of PTHs and IHs in other models to the evolution of hidden states by injecting their outputs as task vectors. The results clearly demonstrate their respective significance in fostering the separability and alignment of hidden states. In particular, the PTH task vectors of other models exhibit greater potential to increase hidden-state separability than the result shown in Figure 8.

### I.3 Replication of Table 1 for other models

In Table 16-21, we compare the layer distributions of the top 1%, 2%, 5%, and 10% identified IHs and PTHs in other models and conduct Mann–Whitney U tests to assess the significance of the differences. The results demonstrate a consistent difference in the layer positions of PTHs and IHs across models, with PTHs significantly preceding IHs on average—the difference being most prominent in the two Gemma-family models, Gemma-2B and Gemma-7B.

### I.4 Replication of Figure 9 for other models

In Figure 65-70, we show the results of injecting task vectors derived from IH outputs for other models. The task vectors are similarly injected at layer $\frac{3}{8}L$, where $L$ is the total number of layers of the respective models. The results demonstrate that IH outputs qualify as effective task vectors across models, with the exception of Gemma-2B, as they considerably improve prediction accuracy in the zero-shot setting after being injected into the zero-shot hidden states.

### I.5 Results of attention heads ablation in the ICL setting where query label is not in context

The results for other models, reported in Figure 29-34, confirm the generality of our conclusions across model sizes and architectures. To test whether these findings hold under different ICL settings as well, we replicate the attention head ablation experiments in a special ICL scenario ("In-weight Learning") studied by Reddy [2024] and Chan et al. [2022], where none of the in-context demonstrations share the query's label. This setting is useful for analyzing attention heads—especially **IHs**, which are known for copy-like behavior—when direct label copying is not possible.

To this end, we construct a Famous People dataset following the procedures in Appendix D. Each data point is a sentence in the format "{Famous Person Name} is a:", and the label is the profession of the famous person. Once combined, they form an input sequence like, e.g., "Taylor Swift is a: Singer." that can be used as either a query or a demonstration. For each query, we sample $k = 4$ demonstrations, ensuring that none share the query's profession label. We then ablate PTHs and IHs to examine their influence on model performance. As a control, we also evaluate the measures in the setting where at least one demonstration does share the query's label, allowing us to assess how the availability of directly copyable labels affects the impact of head ablation.

The results in Figure 35 resemble those in standard ICL settings in that ablating **PTHs** significantly impairs separability, while ablating **IHs** affects alignment. However, there are notable differences. Most importantly, the accuracy drop from ablating PTHs is greater than that from ablating IHs—opposite to the standard case, where IH ablation has a larger impact. This highlights the critical role of having the query label present in the context for IHs to function effectively. The importance of label presence is further confirmed by the substantial accuracy gain observed when the query label is forced into the context.

We provide results analogous to Figure 35 for other models in Figure 36–41, which demonstrate similar results.

### I.6 Results of attention heads ablation in the zero-shot setting

In this section, we investigate the effect of attention head ablation on the separability and alignment measures of zero-shot hidden states. The results averaged across datasets, as provided in Figure 52, reveal that the zero-shot setting is fundamentally different from ICL. In the zero-shot setting, ablating IHs has practically no effect on either separability or alignment measures, similar to ablating random heads, because there are no demonstrations from which to copy. On the contrary, ablating PTHs substantially compromises all separability and alignment measures. This illustrates that the contribution of PTHs to the separability of hidden states is by no means restricted to the ICL setting and highlights the significance of PTHs as key structural components of LLMs.

## J  Experiment details of using IH outputs to construct task vectors

For each dataset, we first construct 8-shot ICL prompts using the first 50 queries. The demonstrations used are exactly the same as the ones used to evaluate the 8-shot ICL accuracy for each dataset. Then,

following the procedures described in Todd et al. [2024], we obtain the average output (across the 50 prompts) of each identified top 10% IHs and PTHs at the final token positions, and sum the average outputs of all IHs to obtain the task vector.

In terms of the steering experiment, we add the task vector to the hidden state of the final token of each query at Layer-30 of Llama2-70B, and let the modified hidden state flow through subsequent layers. While calculating the accuracy, we exclude the first 50 queries and only evaluate accuracy on the remaining queries in the test set to ensure an independence between the task vector and the queries involved.

## K   Broader impacts

Understanding the internal mechanisms of large language models is increasingly critical as these models are deployed in high-stakes applications ranging from education and healthcare to legal and governmental decision-making. This work contributes to that understanding by providing a geometric framework that explains in-context learning through the lens of hidden state dynamics, specifically separability and alignment. By identifying the role of specific architectural components—such as Previous Token Heads and Induction Heads—in shaping model behavior, our study brings interpretability to a domain often criticized for opacity.

The primary positive societal impact of this research lies in its potential to improve the transparency, controllability, and safety of LLMs. Better mechanistic understanding can inform the development of more robust models that are less reliant on spurious correlations and more capable of structured generalization. For example, insights into how semantic information is injected and refined layer by layer may support diagnostic tools that detect when a model is failing to align its internal representations with task-relevant signals. Additionally, the SVD-based techniques we propose for identifying and amplifying task-relevant components could be used to improve the efficiency or reliability of model outputs, particularly in low-resource or privacy-sensitive settings where retraining is infeasible.

However, a deeper understanding of model internals also introduces risks. The ability to precisely manipulate hidden states or attention mechanisms may enable adversarial behaviors, such as constructing prompts that selectively suppress or amplify certain outputs for political or financial gain. Moreover, techniques for filtering and steering semantic content could be misused to hide bias or simulate alignment without genuine safety improvements. As such, we encourage future work to pair interpretability research with rigorous safety and ethics evaluations, and to involve interdisciplinary expertise when applying these findings to real-world systems.

Overall, we view our contributions as a step toward more interpretable and accountable AI, but we emphasize that interpretability alone is not a guarantee of ethical deployment. Responsible application of these insights requires careful consideration of both technical and societal factors.

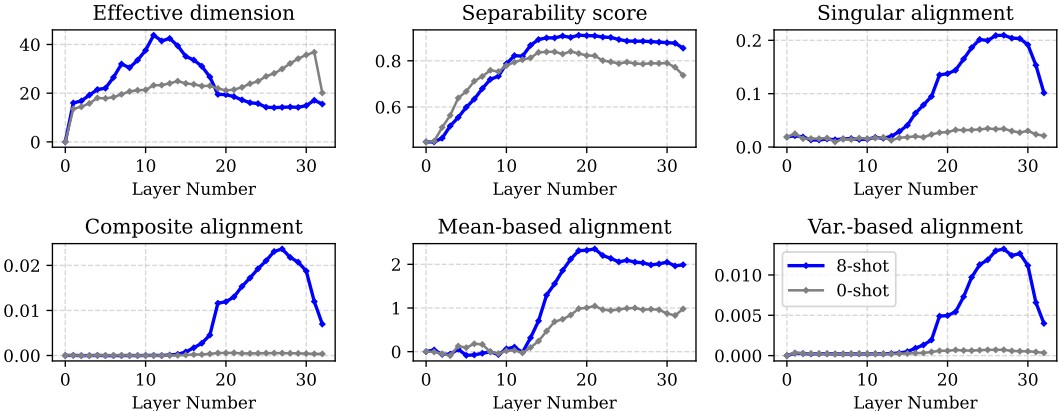

Figure 10: Comparison of trends in separability and alignment measures of Llama3-8B hidden states between ICL and zero-shot

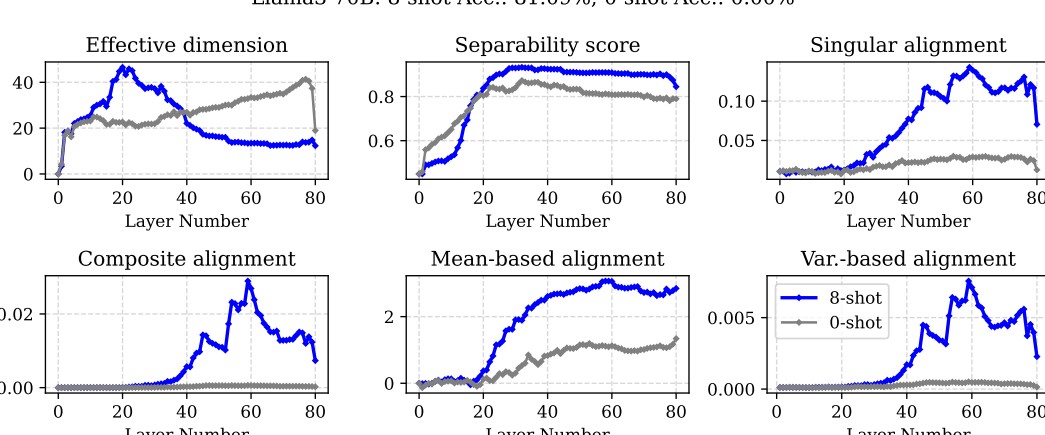

Figure 11: Comparison of trends in separability and alignment measures of Llama3-70B hidden states between ICL and zero-shot

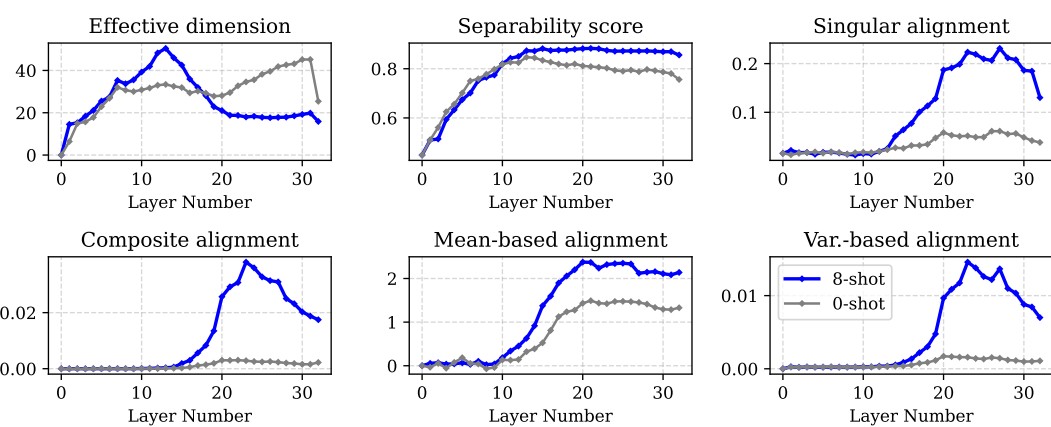

Figure 12: Comparison of trends in separability and alignment measures of Llama2-7B hidden states between ICL and zero-shot

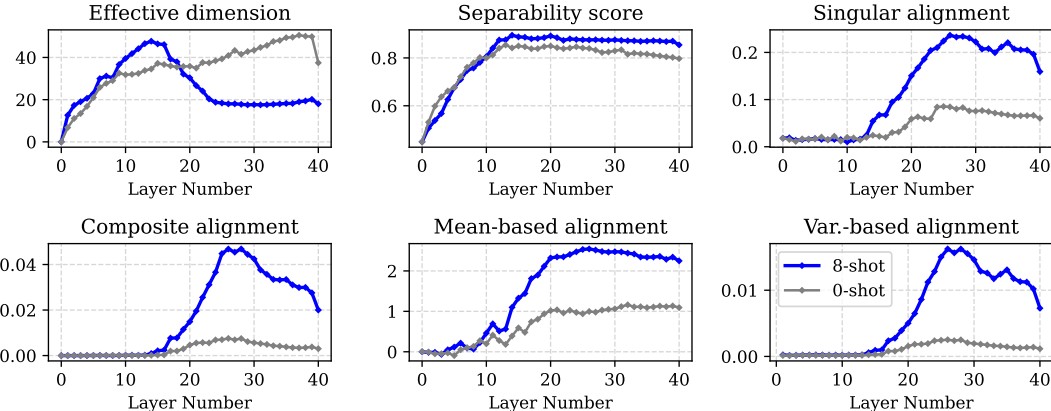

Figure 13: Comparison of trends in separability and alignment measures of Llama2-13B hidden states between ICL and zero-shot

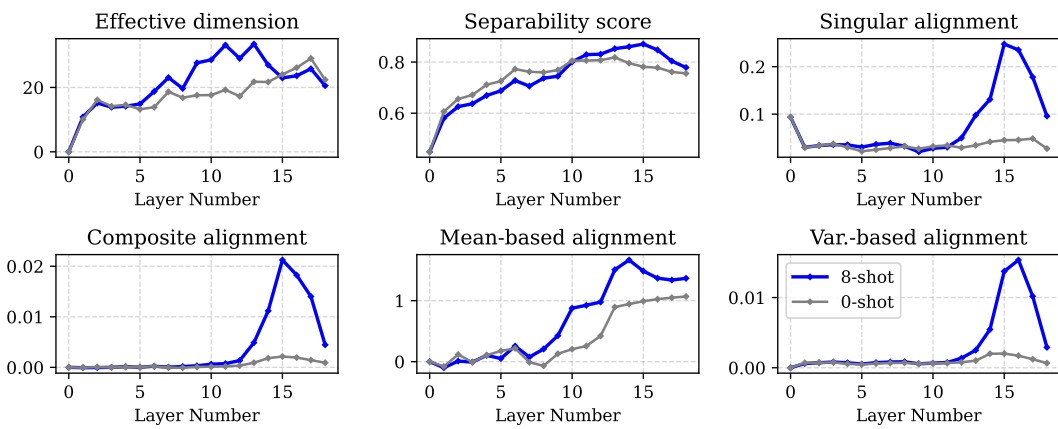

Figure 14: Comparison of trends in separability and alignment measures of Gemma-2B hidden states between ICL and zero-shot

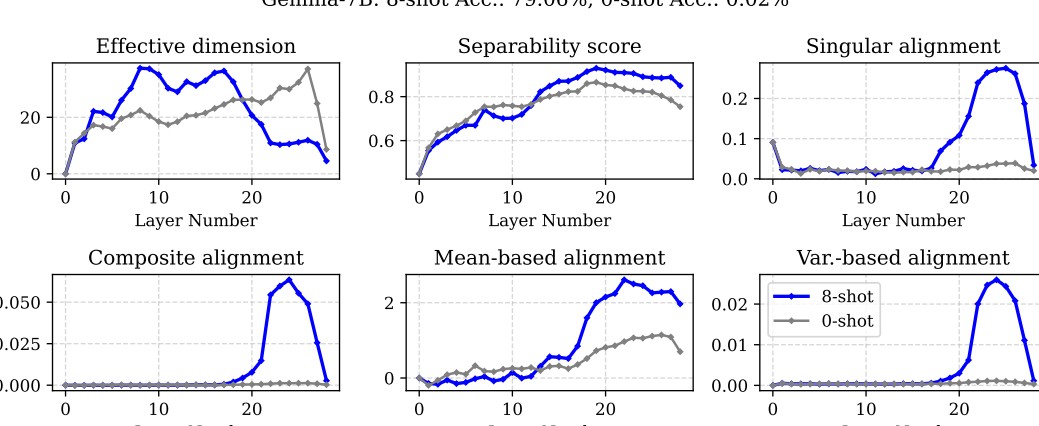

Figure 15: Comparison of trends in separability and alignment measures of Gemma-7B hidden states between ICL and zero-shot

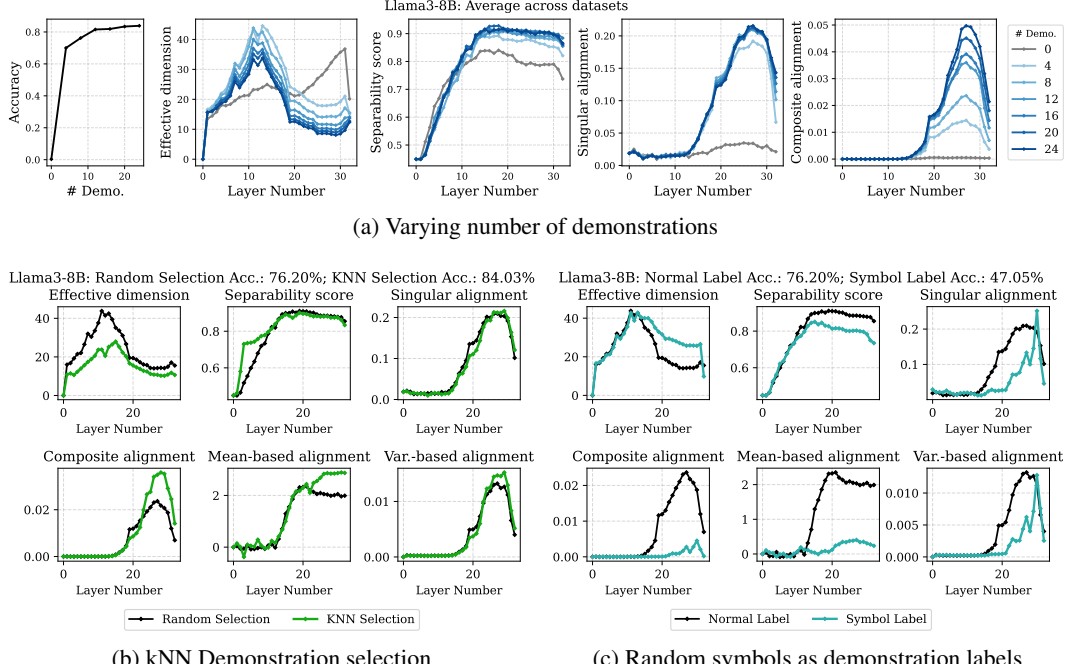

(a) Varying number of demonstrations

(b) kNN Demonstration selection

(c) Random symbols as demonstration labels

Figure 16: Layer-wise trends of separability and alignment measures of Llama3-8B hidden states in different ICL settings

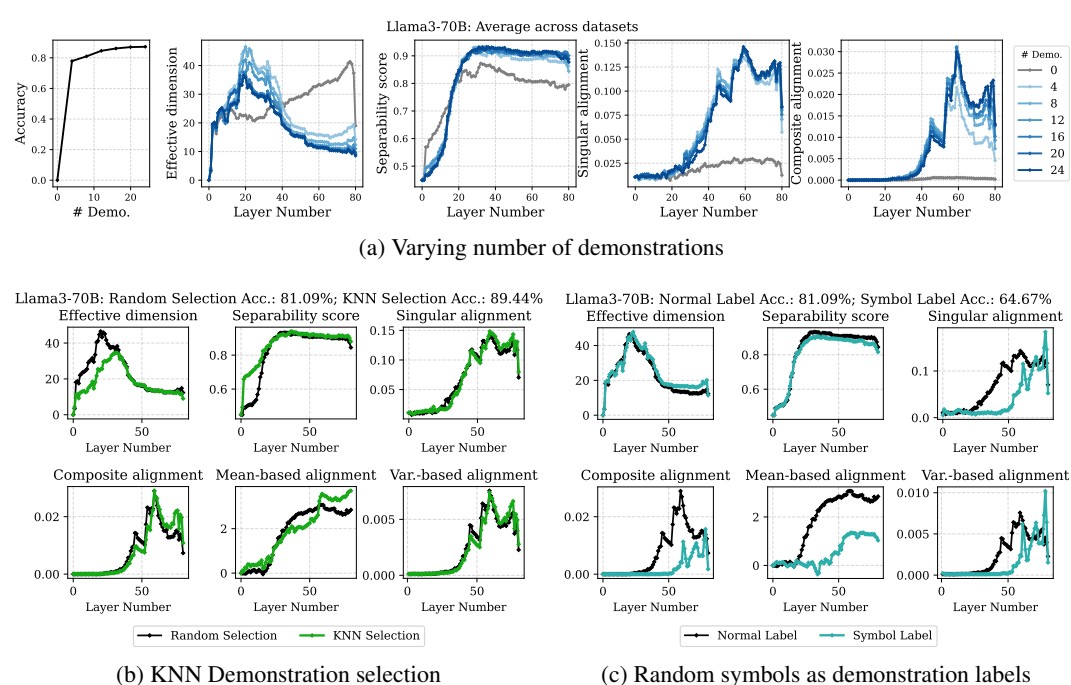

(a) Varying number of demonstrations

(b) KNN Demonstration selection

(c) Random symbols as demonstration labels

Figure 17: Layer-wise trends of separability and alignment measures of Llama3-70B hidden states in different ICL settings

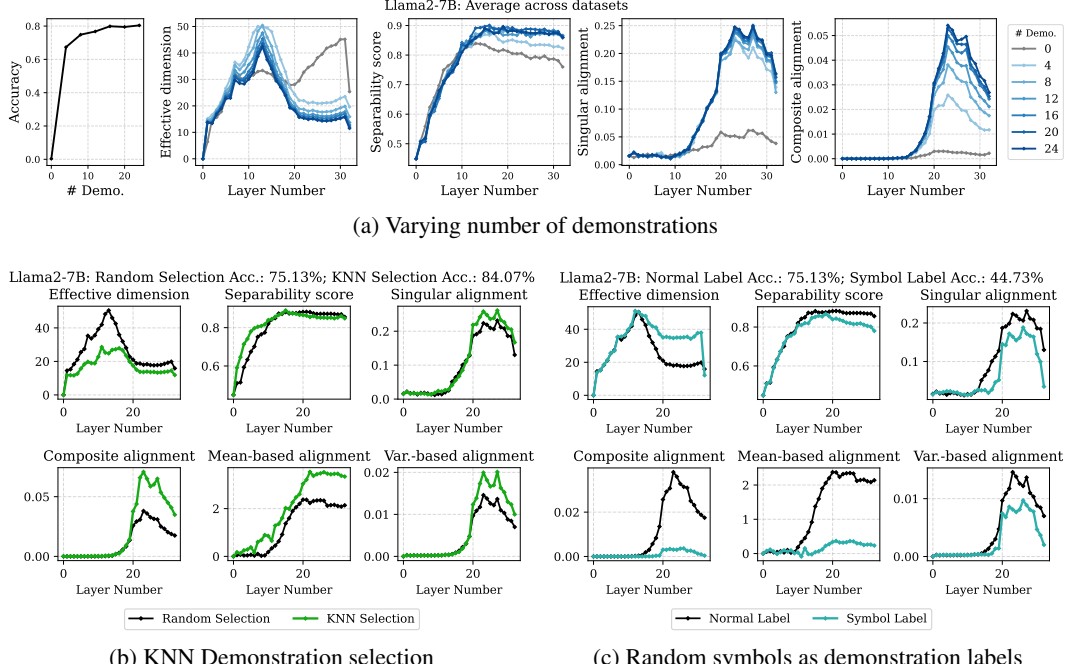

(a) Varying number of demonstrations

(b) KNN Demonstration selection

(c) Random symbols as demonstration labels

Figure 18: Layer-wise trends of separability and alignment measures of Llama2-7B hidden states in different ICL settings

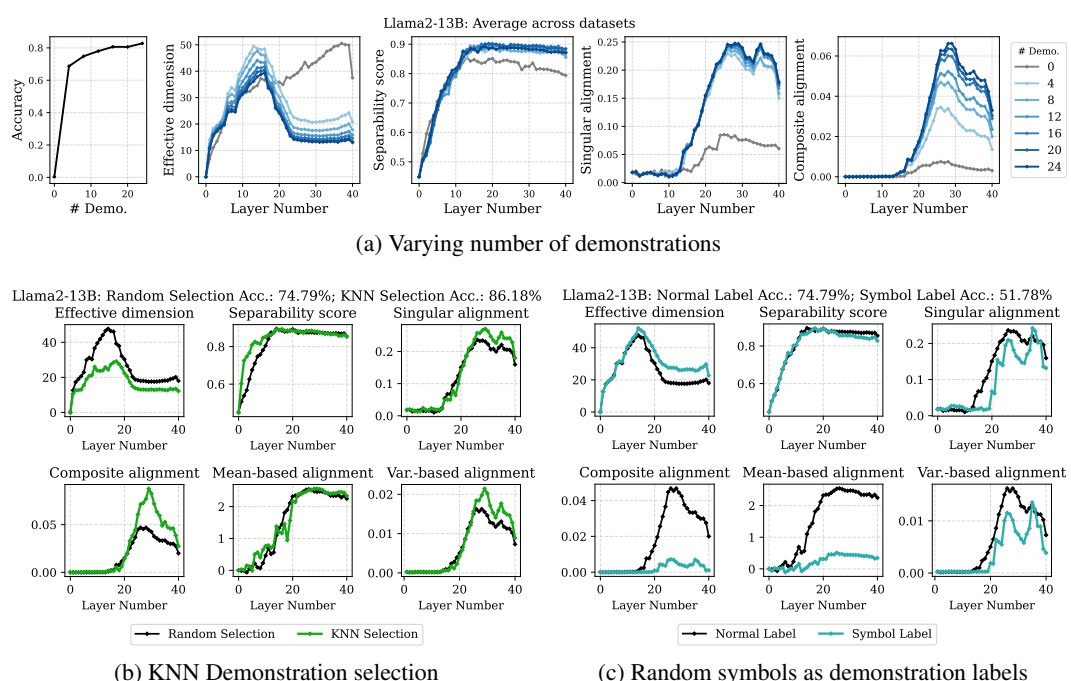

(a) Varying number of demonstrations

(b) KNN Demonstration selection

(c) Random symbols as demonstration labels

Figure 19: Layer-wise trends of separability and alignment measures of Llama2-13B hidden states in different ICL settings

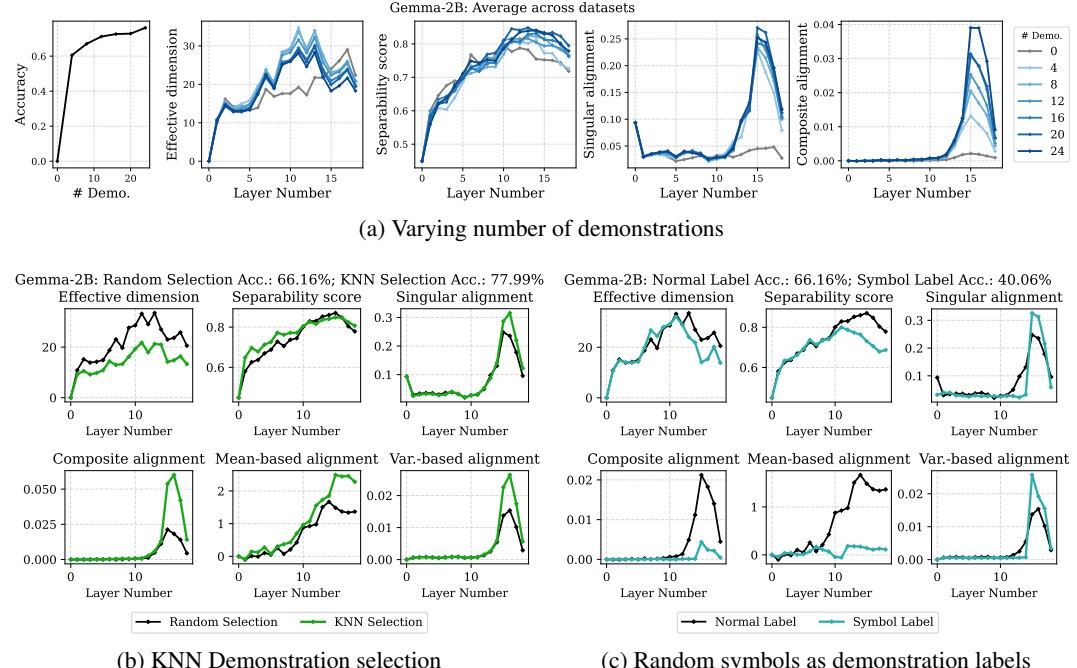

(a) Varying number of demonstrations

(b) KNN Demonstration selection

(c) Random symbols as demonstration labels

Figure 20: Layer-wise trends of separability and alignment measures of Gemma-2B hidden states in different ICL settings

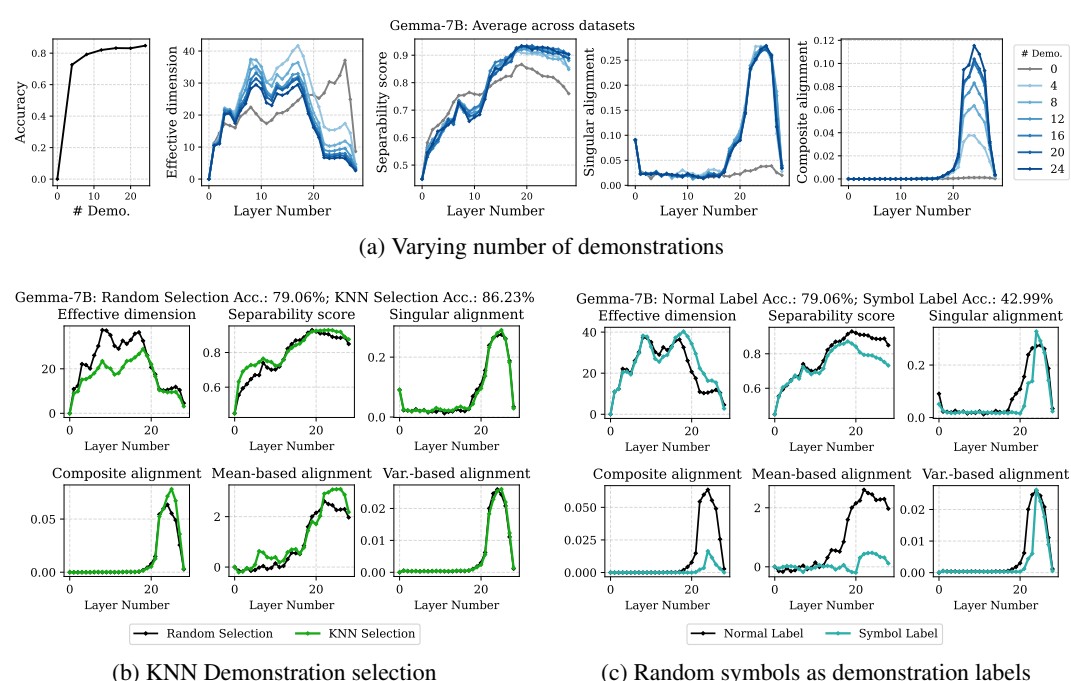

(a) Varying number of demonstrations

(b) KNN Demonstration selection

(c) Random symbols as demonstration labels

Figure 21: Layer-wise trends of separability and alignment measures of Gemma-7B hidden states in different ICL settings

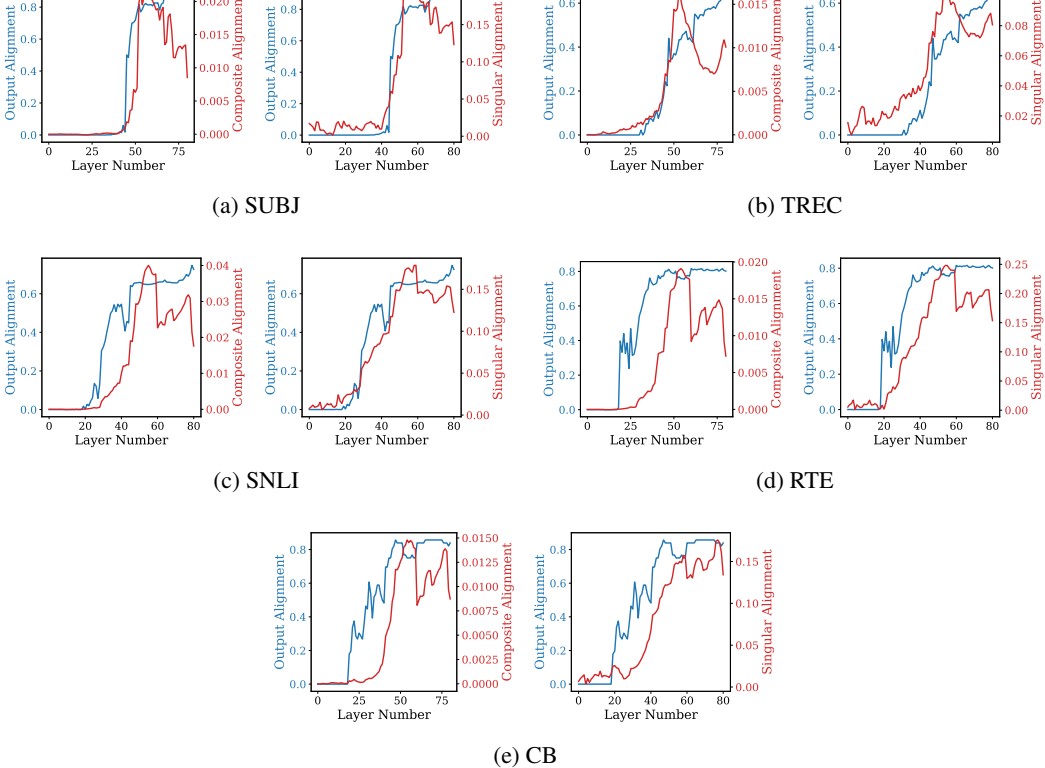

Figure 22: Output and directional alignment of Llama2-70B ICL hidden states with label unembedding vectors on various datasets.

Table 4: Tokens decoded from top 10 singular directions from layers near the phase transition on SUBJ

| Layer | Tokens |
|-------|--------|
| Layer 51 | opinion, humor, genre, adm, facts, plot, description, explanation, pont, humor |
| Layer 52 | **objective**, humor, ., adm, dram, plot, enthus, categor, pont, subject |
| Layer 53 | **objective**, humor, ., plot, fact, humor, enthus, explanation, character, humor |

Table 5: Tokens decoded from top 10 singular directions from layers near the phase transition on TREC

| Layer | Tokens |
|-------|--------|
| Layer 18 | Ras, Float, Physics, Heimat, Zob, rela, ijd, lbl, alphabet, uther |
| Layer 19 | **Geography**, name, Physics, **map**, >, Ele, **location**, Vall, alphabet, Tests |
| Layer 20 | **definitions**, histor, **maps**, **Geography**, amos, inn, **map**, babel, etr, letters |

Table 6: Tokens decoded from top 10 singular directions of layers near the phase transition on SNLI

| Layer | Tokens |
|-------|--------|
| Layer 24 | imagination, inen, exceptions, hner, igin, amerikanischer, Fich, float, opposite, orf |
| Layer 25 | dup, **unknown**, **similarity**, Dy, Sci, ete, nost, Fich, endorf, **Correct** |
| Layer 26 | erde, **unknown**, **similarity**, context, apper, architecture, unity, **ambigu**, alg, py |

Table 7: Tokens decoded from top 10 singular directions from layers near the phase transition on RTE

| Layer | Tokens |
|-------|--------|
| Layer 21 | Eva, pul, ikel, orp, Mol, enk, Dal, adrat, hor, üng |
| Layer 22 | **correct**, nor, urch, Boh, VP, aqu, Dal, subscribe, currency, sign |
| Layer 23 | **correction**, zero, tempo, Boh, VP, lak, FA, iva, currency, trag |

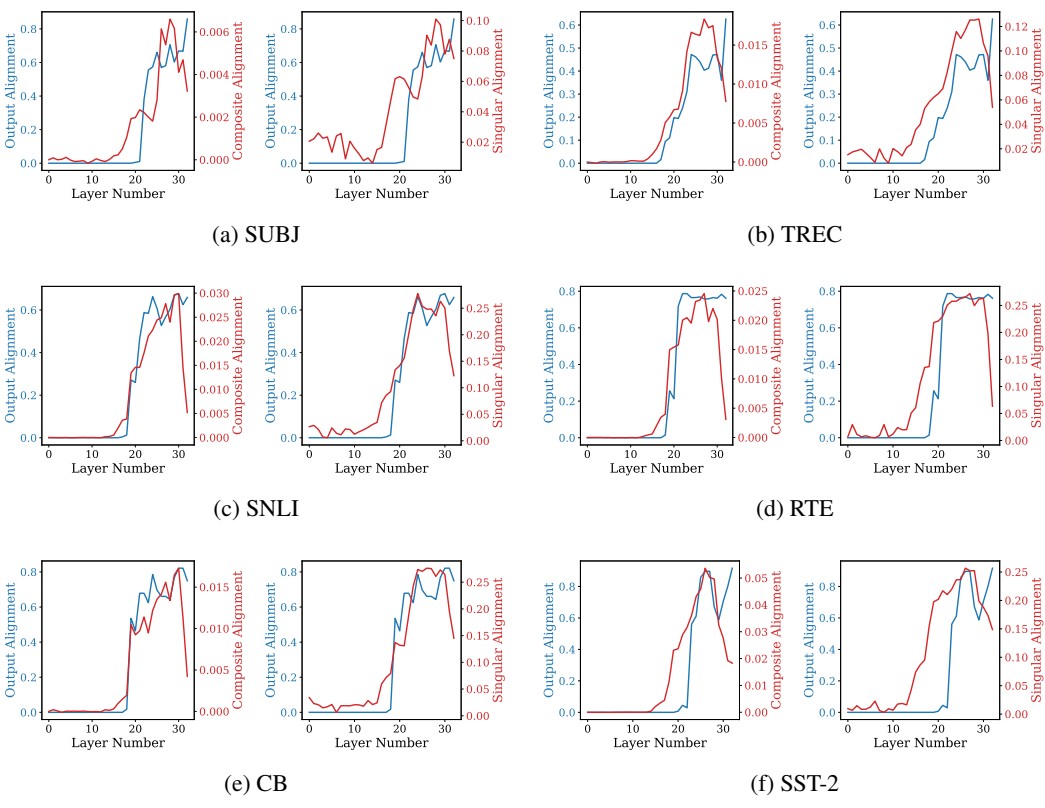

Figure 23: Output and directional alignment of Llama3-8B ICL hidden states with label unembedding vectors on various datasets.

Table 8: Tokens decoded from top 10 singular directions from layers near the phase transition on CB

| Layer | Tokens |
|---|---|
| Layer 22 | TR, eign, >, qual, Joan, Icon, ebol, рит, ąż, Kontrola |
| Layer 23 | deg, Cet, Bou, SV, ando, Simon, **True**, lav, iből, **opposition** |
| Layer 24 | **negative**, schap, ftrag, court, ando, iada, rib, Martí, counter, **answer** |

Table 9: Tokens decoded from top 10 singular directions with labels replaced on SNLI

| Layer | Tokens |
|---|---|
| Layer 23 | Tier, aro, cum, Vector, FB, Eur, Fon, pending, Dre, Fant |
| Layer 24 | **False**, aten, beck, ardo, ⅓, Sci, attan, Kaiser, tant, Fant |
| Layer 25 | **False**, Cop, cgi, Dou, ⅓, Ra, hur, TF, Dre, nehm |
| Layer 44 | false, combination, inois, multiply, scenario, group, action, group, prec, Solo |
| Layer 45 | False, **#**, xspace, @, **#**, neutral, @, large, sentiment, False |
| Layer 46 | False, @, contr, **#**, @, neutral, cheer, group, ppo, monot |
| Layer 78 | false, False, $., @@, @, false, $., false, False, # |
| Layer 79 | false, !, ., #, True, , !., Hash, Street, 群 |
| Layer 80 | twe, ., X, !, :#, ##, action, !, #, Street |

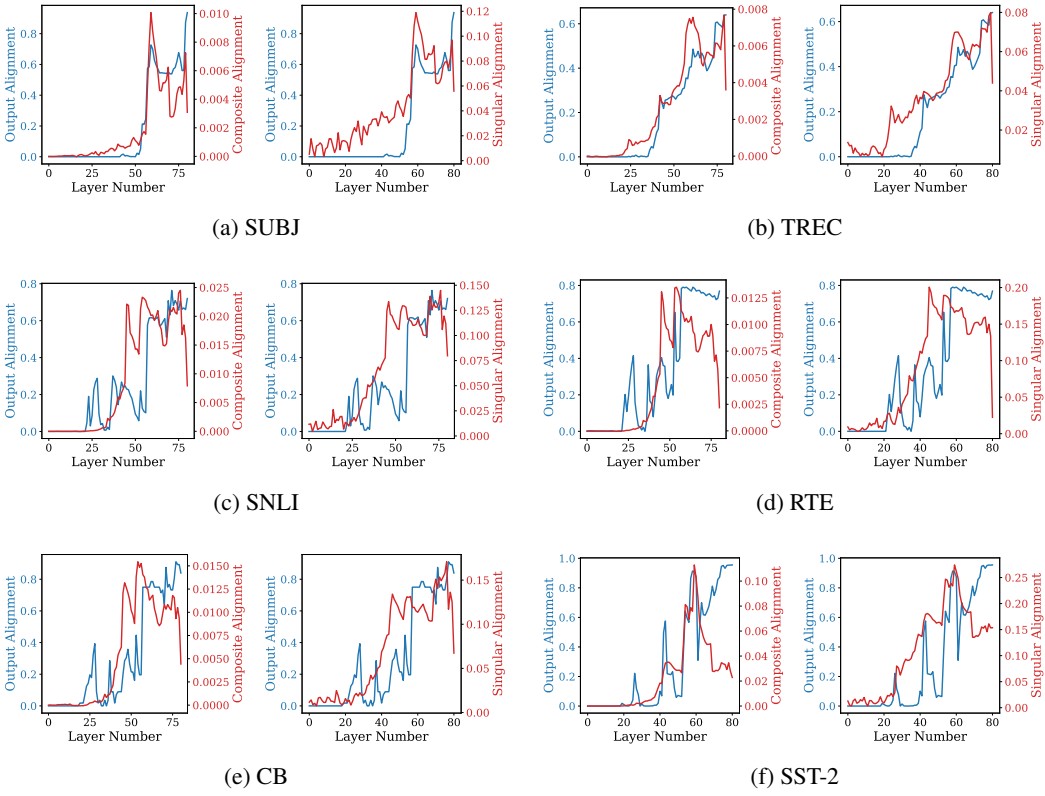

Figure 24: Output and directional alignment of Llama3-70B ICL hidden states with label unembedding vectors on various datasets.

Table 10: Tokens decoded from top 10 singular directions with labels replaced on SST-2

| Layer | Tokens |
|---|---|
| Layer 24 | poste, Cant, Jug, Lars, >, zak, zig, geh, eg, alus |
| Layer 25 | rott, igi, endi, noreferrer, **positive**, zak, Aur, Mans, ŭ, humor |
| Layer 26 | rott, Armen, Lau, noreferrer, **negative**, mee, cot, tu, ƒ, cita |
| Layer 40 | ipage, pra, Kle, estaven, Maj, CURLOPT, extreme, humor, ⊙, anel |
| Layer 41 | ViewById, pra, meno, Einzelnach, ziel, @, disappoint, Meyer, L, humor |
| Layer 42 | negative, pra, nar, Einzelnach, **#**, humor, extreme, humor, embargo, **#** |
| Layer 78 | negative, пози, elt, neutral, hyper, $., , positive, inea, intros |
| Layer 79 | positive, ( {, *., neutral, Tru, ???, , ?, humor, Sent |
| Layer 80 | :(, positive, ·, !, disappoint, positive, about, positive, wod, intros |

Table 11: Tokens decoded from top retained and filtered directions in layer updates on SUBJ

| | | **Retained Tokens** |
|---|---|---|
| **Stage** | **Layer** | **Tokens** |
| Stage 1 | Layer 54 | humor, description, explanation, **objective**, humor, shock, **subject**, character, comparison, humor |
| | Layer 59 | humor, plot, pra, facts, **objective**, hypoth, **subject**, met, explanation, **subject** |
| | Layer 66 | ., **objective**, Plot, explan, prom, **subject**, character, fact, **objective**, sci |
| Stage 2 | Layer 80 | plot, observation, IM, Thom, verb, hyper, quar, Em, ps, Guillaume |

| | | **Filtered Tokens** |
|---|---|---|
| **Stage** | **Layer** | **Tokens** |
| Stage 1 | Layer 54 | oro, Kant, Harr, Werner, Bast, desert, Kent, Uns, ola, Fall |
| | Layer 59 | iter, Arg, halten, enser, Lok, Kontrola, tring, atif, ias, Sof |
| | Layer 66 | conj, dens, shal, amo, Lisa, lish, Sob, narrow, UMN, co |
| Stage 2 | Layer 80 | **subject**, Met, chron, intros, mix, emot, humor, Type, **jective**, **subject** |

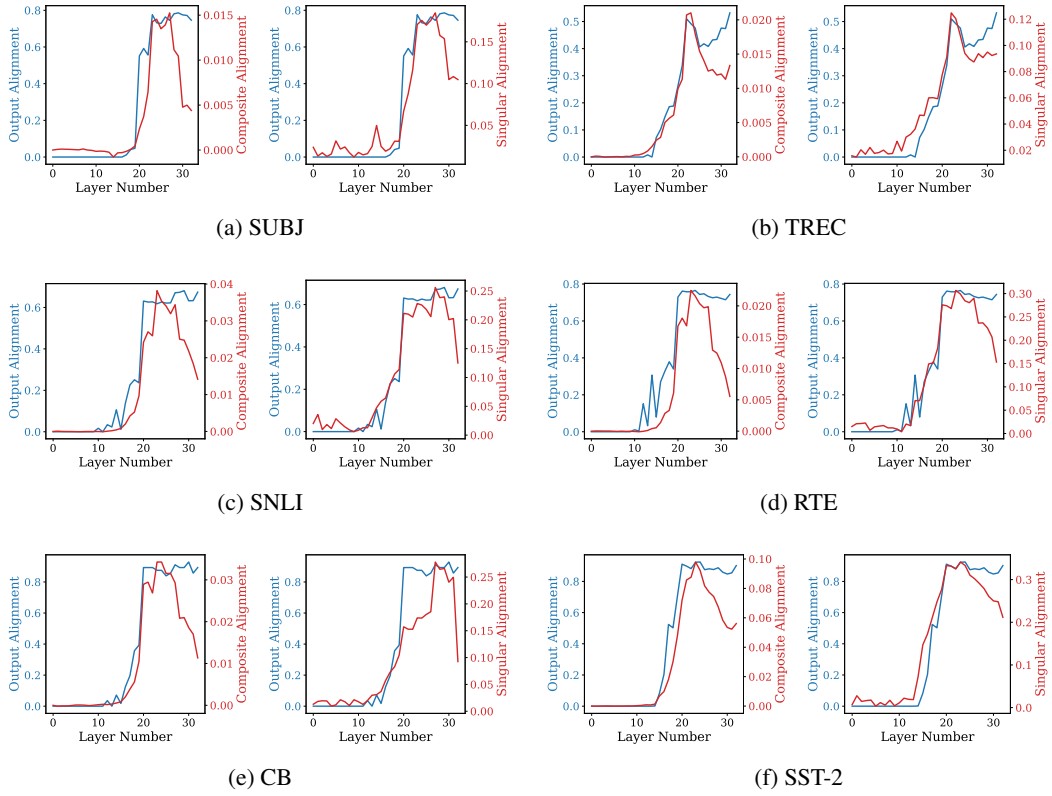

Figure 25: Output and directional alignment of Llama2-7B ICL hidden states with label unembedding vectors on various datasets.

Table 12: Tokens decoded from top retained and filtered directions in layer updates on TREC

| | | **Retained Tokens** |
|---|---|---|
| **Stage** | **Layer** | **Tokens** |
| Stage 1 | Layer 30 | pione, taste, **definition**, temporal, precision, science, amen, lich, irectory, enta |
| | Layer 50 | **description**, subst, **person**, **numbers**, interpret, **location**, wars, **location**, answer, date |
| | Layer 70 | нау, date, **number**, **location**, **abbre**, **currency**, **location**, **number**, **description**, Creative |
| Stage 2 | Layer 80 | hol, Heidel, Desp, eth, olis, **description**, origin, Type, **description**, arto |

| | | **Filtered Tokens** |
|---|---|---|
| **Stage** | **Layer** | **Tokens** |
| Stage 1 | Layer 30 | ako, dn, amen, ainer, overlay, aar, CHAP, storm, synchron, tober |
| | Layer 50 | Gordon, Chinese, acci, Grace, penas, zak, gas, Kü, ps, prototype |
| | Layer 70 | mas, fit, ioni, Salt, ru, oci, être, quet, thro, Sever |
| Stage 2 | Layer 80 | , **number**, list, discipline, organ, entertain, weather, bum, poll, coff |

Table 13: Tokens decoded from top retained and filtered directions in layer updates on SNLI

| | | **Retained Tokens** |
|---|---|---|
| **Stage** | **Layer** | **Tokens** |
| Stage 1 | Layer 25 | pac, ício, charm, beck, hist, rap, iella, **fact**, **contradiction**, engol |
| | Layer 50 | ., **probable**, **false**, **uncertain**, solo, **neutral**, tender, **unknown**, gender, **unlikely** |
| | Layer 70 | ., reverse, **false**, scenario, **maybe**, reverse, identity, gender, conce, Gram |
| Stage 2 | Layer 80 | rola, Dic, Kra, Column, Common, motor, ..., ge, Salt, kern |

| | | **Filtered Tokens** |
|---|---|---|
| **Stage** | **Layer** | **Tokens** |
| Stage 1 | Layer 25 | brief, uclide, Ath, bul, références, EC, asta, Rot, nia, Taml |
| | Layer 50 | aho, oba, zu, umble, prepared, Mey, jer, Bras, irre, Maj |
| | Layer 70 | Yu, Mare, iner, artificial, overlap, ilder, aret, elev, constru, Storm |
| Stage 2 | Layer 80 | im, , fl, **maybe** , **similarity**, sem, habit, Ta, gender, action |

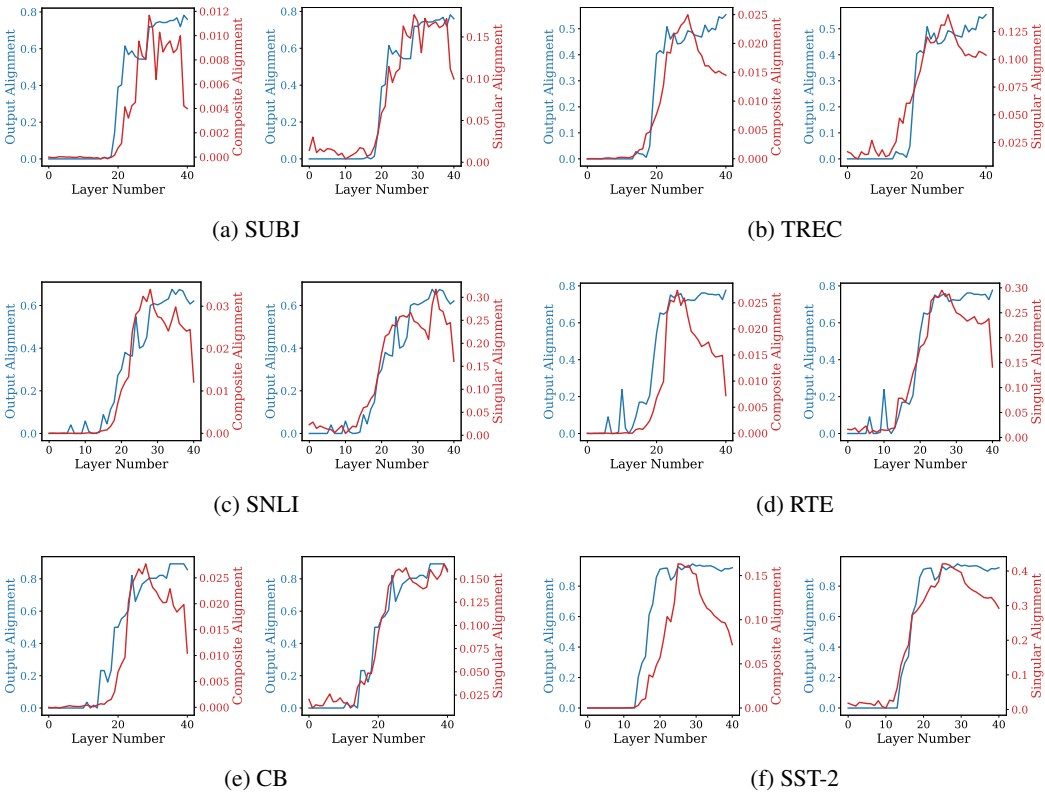

Figure 26: Output and directional alignment of Llama2-13B ICL hidden states with label unembedding vectors on various datasets.

Table 14: Tokens decoded from top retained and filtered directions in layer updates on RTE

| | | **Retained Tokens** |
|---|---|---|
| **Stage** | **Layer** | **Tokens** |
| Stage 1 | Layer 33 | Wikipedia, disput, **yes**, Jakob, alias, Ven, essel, indirect, **False**, isu |
| | Layer 50 | **false**, alarm, irrelevant, **opposite**, nost, naming, duplicates, misunder, Geography, location |
| | Layer 68 | :(, Geography, lament, sto, **impossible**, opinion, ken, numer, **negative**, **true** |
| Stage 2 | Layer 80 | Ale, Ad, Hub, Ke, lub, Terra, Haz, Ta, part, Andre |

| | | **Filtered Tokens** |
|---|---|---|
| **Stage** | **Layer** | **Tokens** |
| Stage 1 | Layer 33 | cust, ▼, Mün, Lucas, itel, Barb, inha, Hed, FI, nica |
| | Layer 50 | лові, CA, Um, wob, prot, jär, 望, ang, 孝, Roman |
| | Layer 68 | fert, CA, bach, Word, RL, gon, Charlie, MS, nen, Feld |
| Stage 2 | Layer 80 | SR, techni, gr, **unclear**, trag, **opposite**, Susan, estimate, rare, murder |

Table 15: Tokens decoded from top retained and filtered directions in layer updates on CB

| | | **Retained Tokens** |
|---|---|---|
| **Stage** | **Layer** | **Tokens** |
| Stage 1 | Layer 25 | Bou, **True**, , Cet, ando, elin, лом, Pear, stabil, lav |
| | Layer 49 | **true**, **unlikely**, **uncertain**, absence, **False**, predictions, avoided, myth, predictions, asym |
| | Layer 70 | **true**, **false**, **agreement**, **maybe**, predictions, silent, negative, disapp, áv, predictions |
| Stage 2 | Layer 80 | Giov, future, often, Frank, tend, imet, Sid, fran, Intent, **maybe** |

| | | **Filtered Tokens** |
|---|---|---|
| **Stage** | **Layer** | **Tokens** |
| Stage 1 | Layer 25 | slant, acci, Laurent, cht, brie, chamber, olare, gart, una, vie |
| | Layer 49 | Dra, cito, nick, Verm, moz, neglect, abel, WR, anger, uchar |
| | Layer 70 | ottom, nation, Cooper, exhaust, reign, hem, Um, diag, Ri, wa |
| Stage 2 | Layer 80 | **false**, esi, , **False**, **Yes**, often, aer, ", Mod, George |

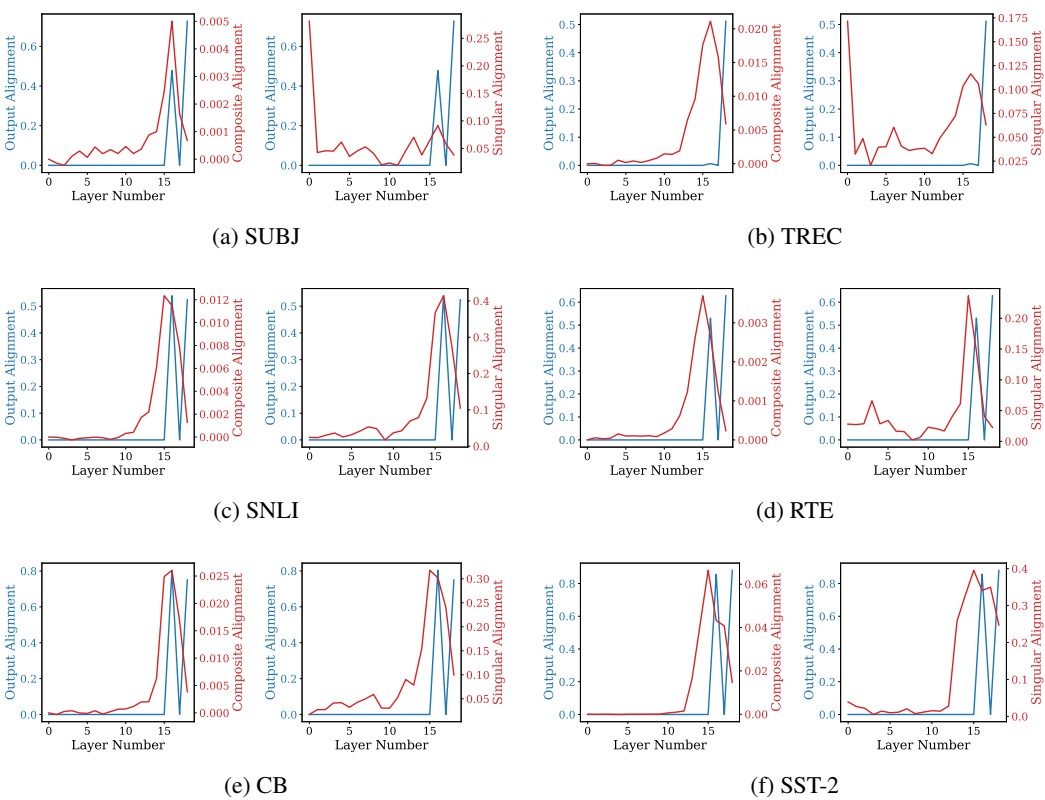

Figure 27: Output and directional alignment of Gemma-2B ICL hidden states with label unembedding vectors on various datasets.

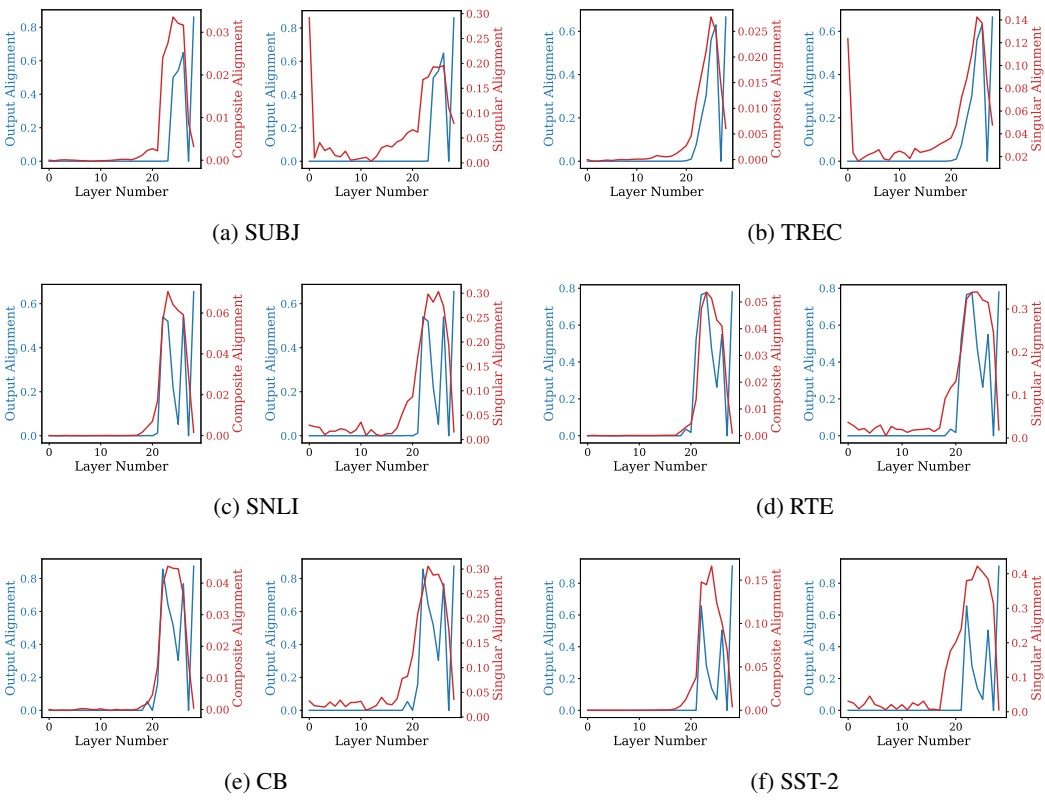

Figure 28: Output and directional alignment of Gemma-7B ICL hidden states with label unembedding vectors on various datasets.

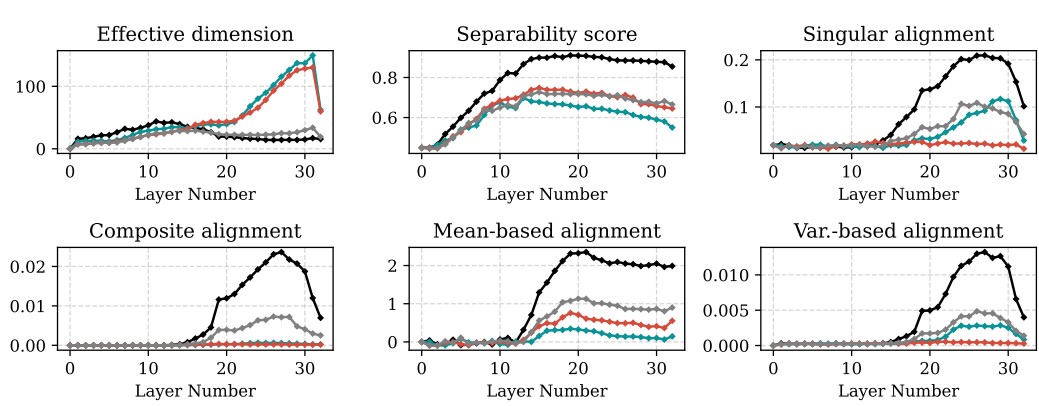

Figure 29: Effects of attention heads ablation on the layer-wise separability and alignment measures of Llama3-8B hidden states averaged across all datasets

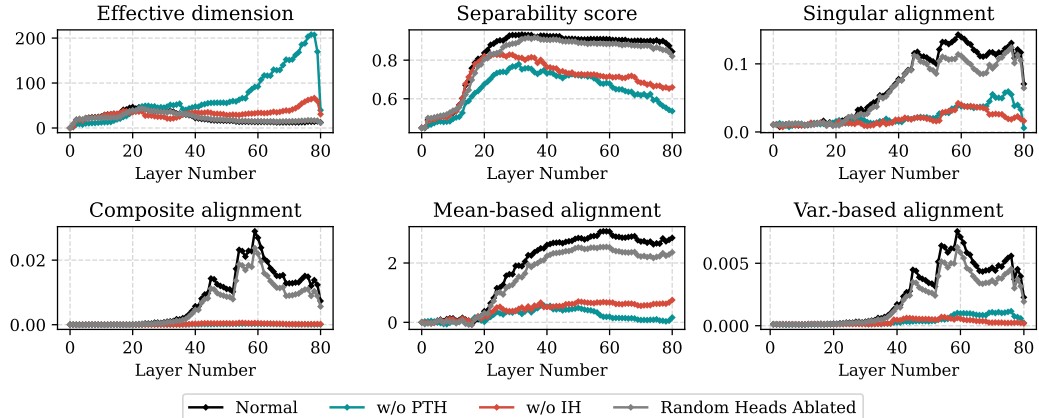

Figure 30: Effects of attention heads ablation on the layer-wise separability and alignment measures of Llama3-70B hidden states averaged across all datasets

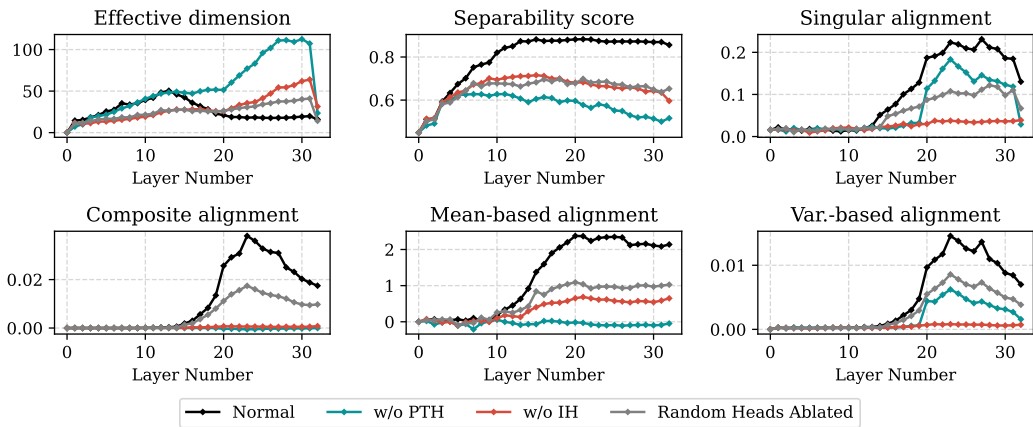

Figure 31: Effects of attention heads ablation on the layer-wise separability and alignment measures of Llama2-7B hidden states averaged across all datasets

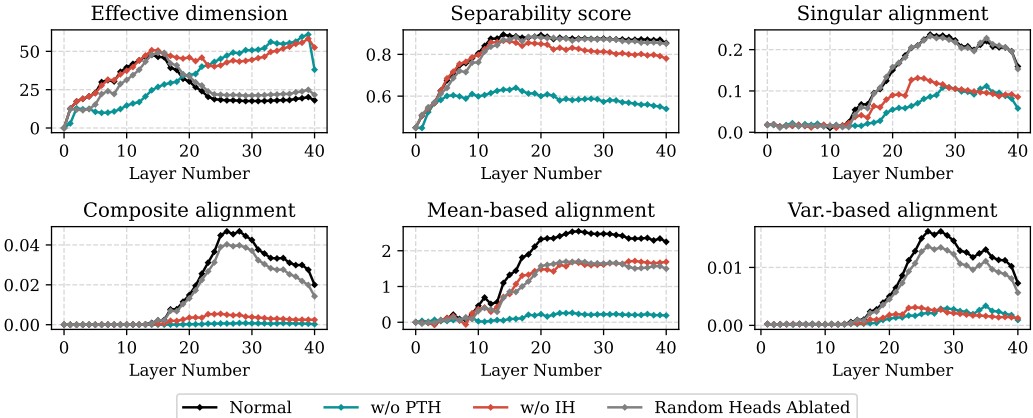

Figure 32: Effects of attention heads ablation on the layer-wise separability and alignment measures of Llama2-13B hidden states averaged across all datasets

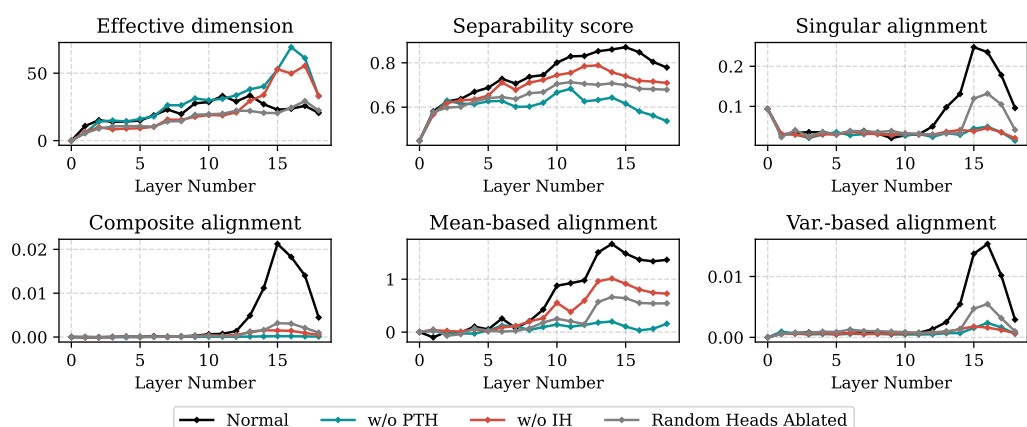

Figure 33: Effects of attention heads ablation on the layer-wise separability and alignment measures of Gemma-2B hidden states averaged across all datasets

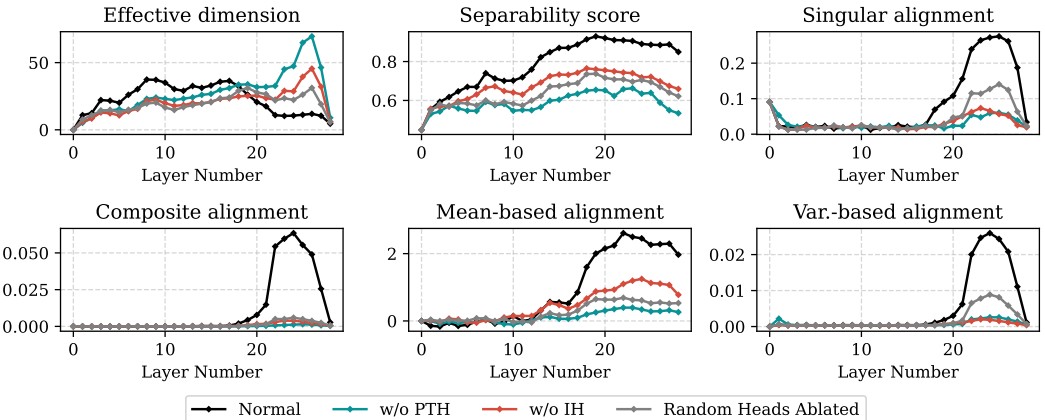

Figure 34: Effects of attention heads ablation on the layer-wise separability and alignment measures of Gemma-7B hidden states averaged across all datasets

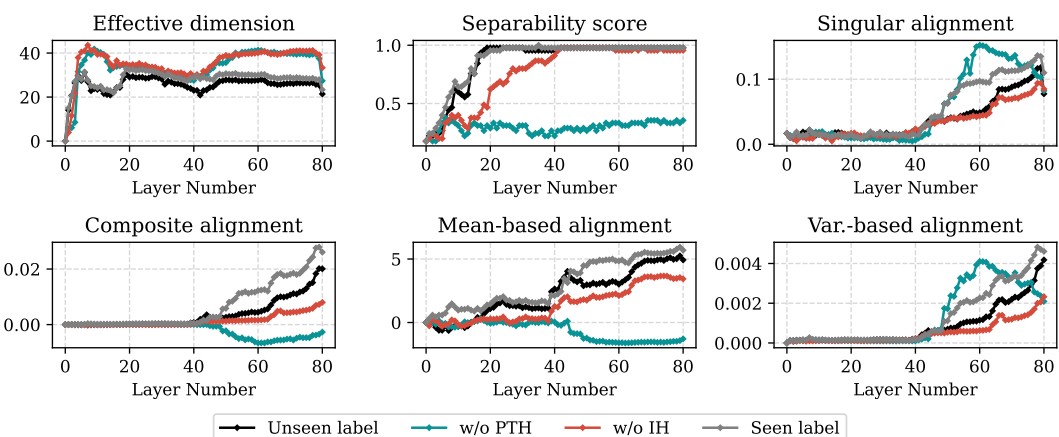

Figure 35: Effects of attention heads ablation on the layer-wise separability and alignment measures of Llama2-70B hidden states on Famous People dataset.

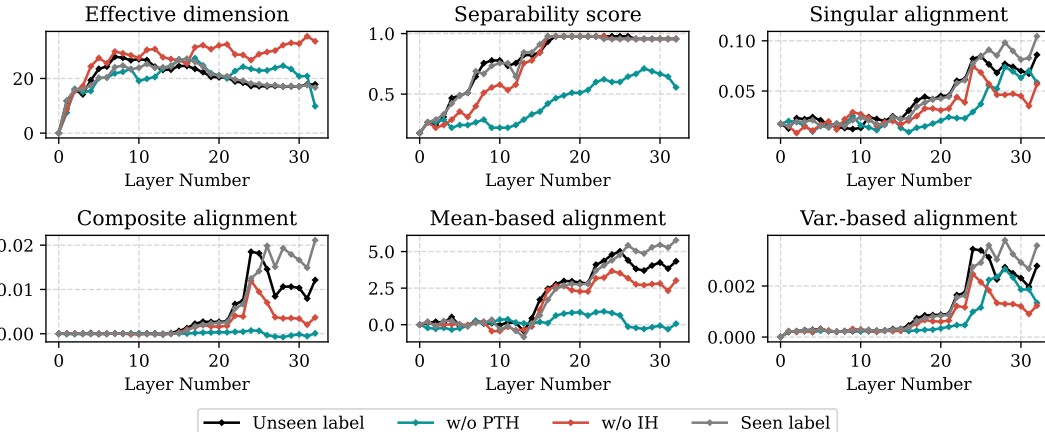

Figure 36: Effects of attention heads ablation on the layer-wise separability and alignment measures of Llama3-8B hidden states on Famous People dataset.

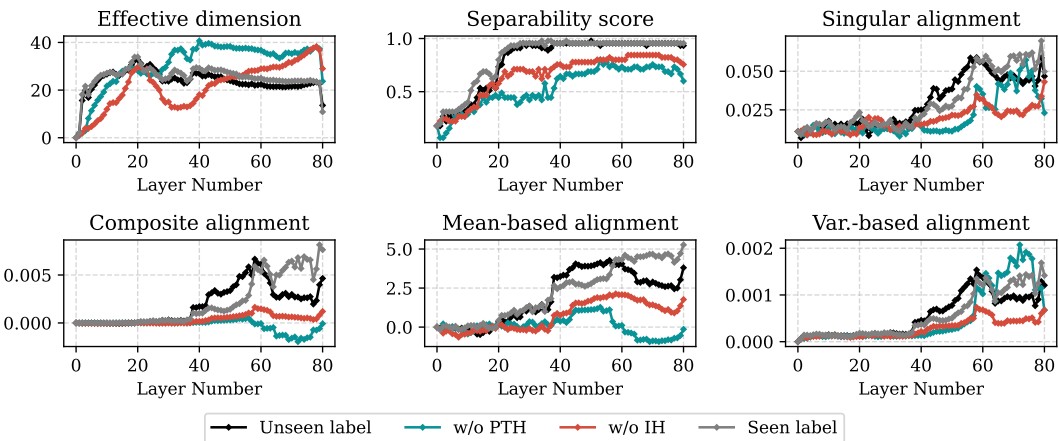

Figure 37: Effects of attention heads ablation on the layer-wise separability and alignment measures of Llama3-70B hidden states on Famous People dataset.

Llama2-7B Famous
Unseen label Acc.: 54.44% | w/o PTH Acc.: 1.11% | w/o IH Acc.: 3.33% | Seen label Acc.: 72.22%

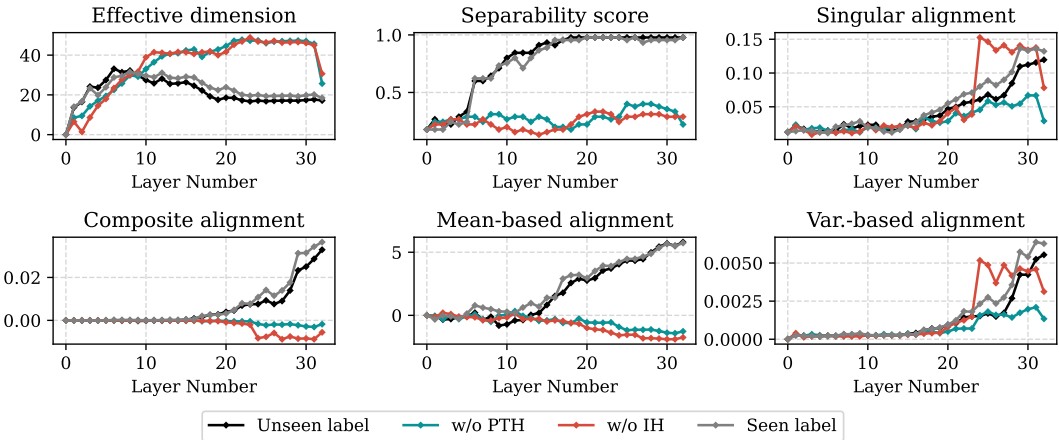

Figure 38: Effects of attention heads ablation on the layer-wise separability and alignment measures of Llama2-7B hidden states on Famous People dataset.

Llama2-13B Famous
Unseen label Acc.: 52.22% | w/o PTH Acc.: 0.00% | w/o IH Acc.: 0.00% | Seen label Acc.: 67.78%

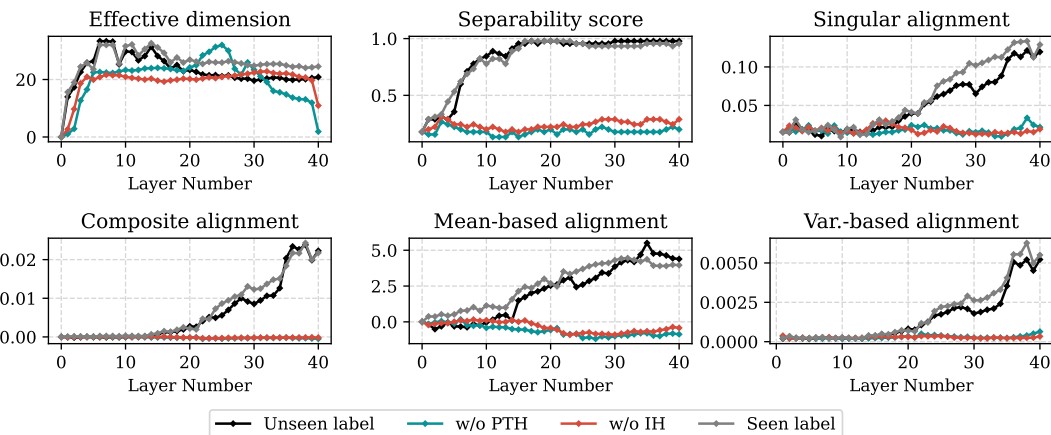

Figure 39: Effects of attention heads ablation on the layer-wise separability and alignment measures of Llama2-13B hidden states on Famous People dataset.

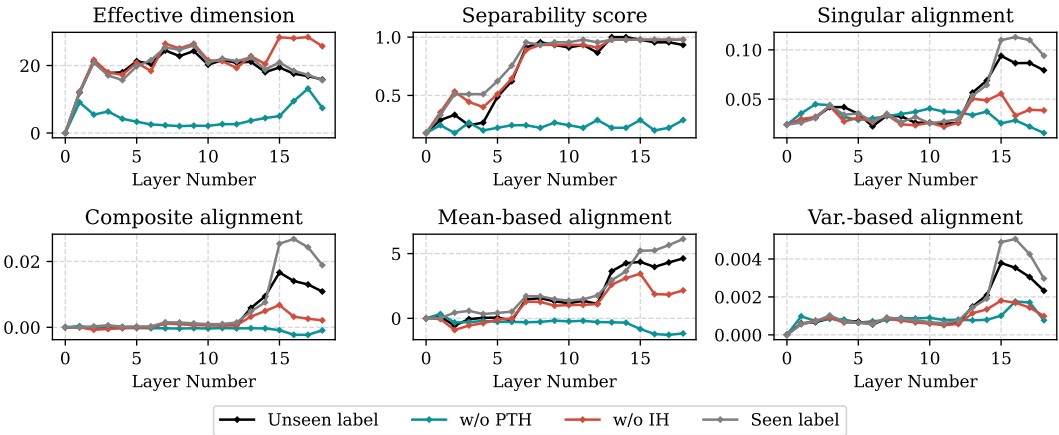

Figure 40: Effects of attention heads ablation on the layer-wise separability and alignment measures of Gemma-2B hidden states on Famous People dataset.

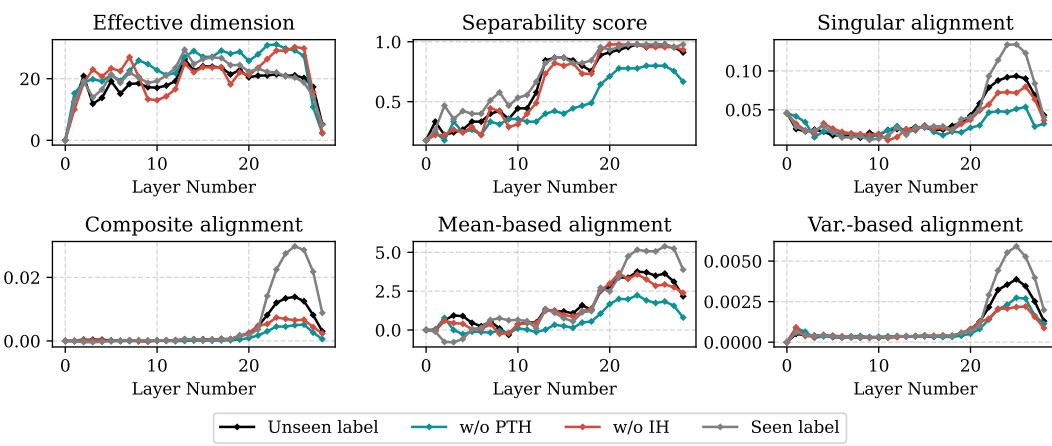

Figure 41: Effects of attention heads ablation on the layer-wise separability and alignment measures of Gemma-7B hidden states on Famous People dataset.

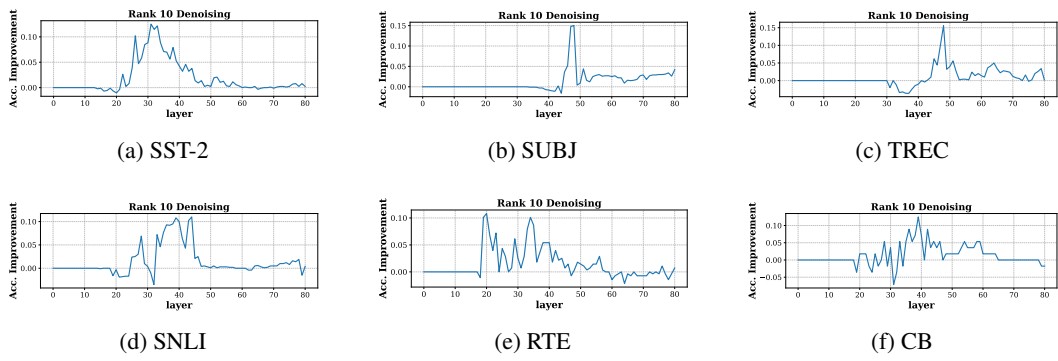

Figure 42: Accuracy gains of rank-10 denoising over layers on all datasets with Llama2-70B

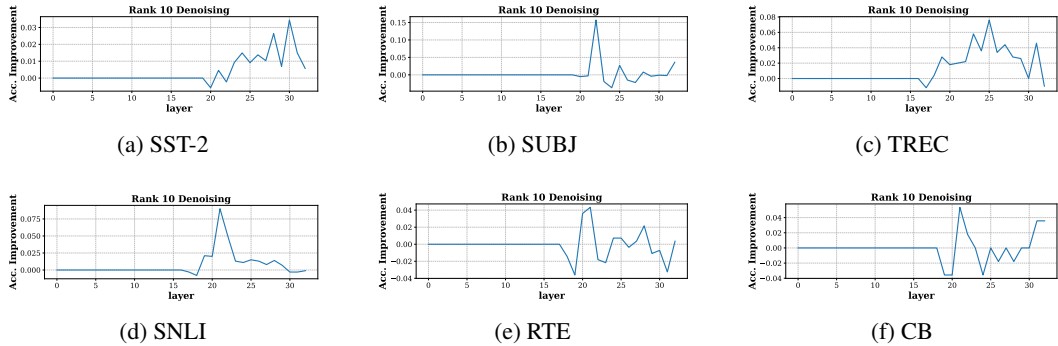

Figure 43: Accuracy gains of rank-10 denoising over layers on all datasets with Llama3-8B

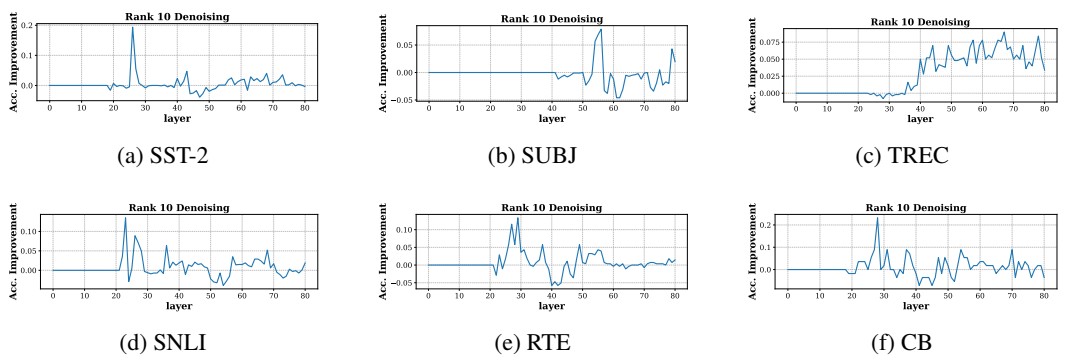

Figure 44: Accuracy gains of rank-10 denoising over layers on all datasets with Llama3-70B

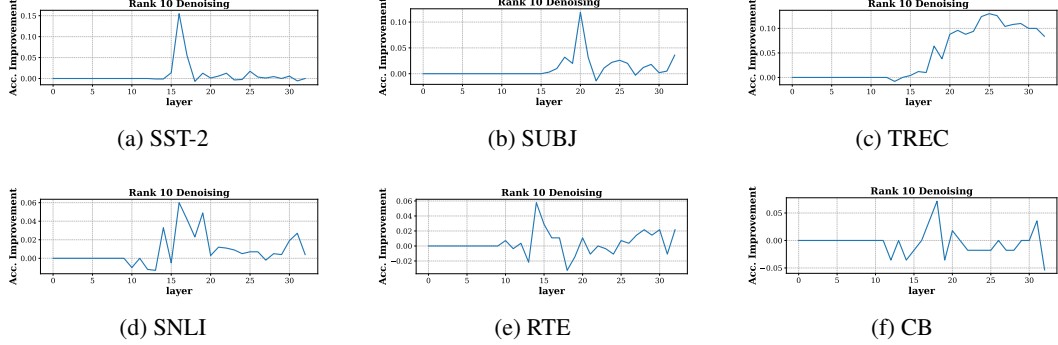

Figure 45: Accuracy gains of rank-10 denoising over layers on all datasets with Llama2-7B

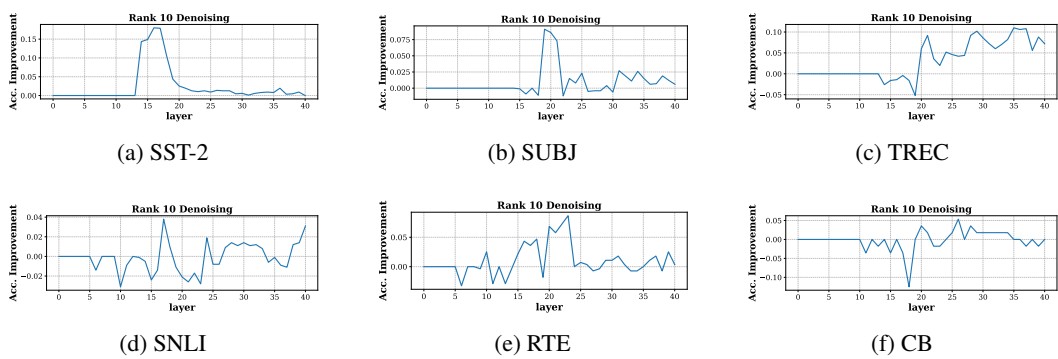

Figure 46: Accuracy gains of rank-10 denoising over layers on all datasets with Llama2-13B

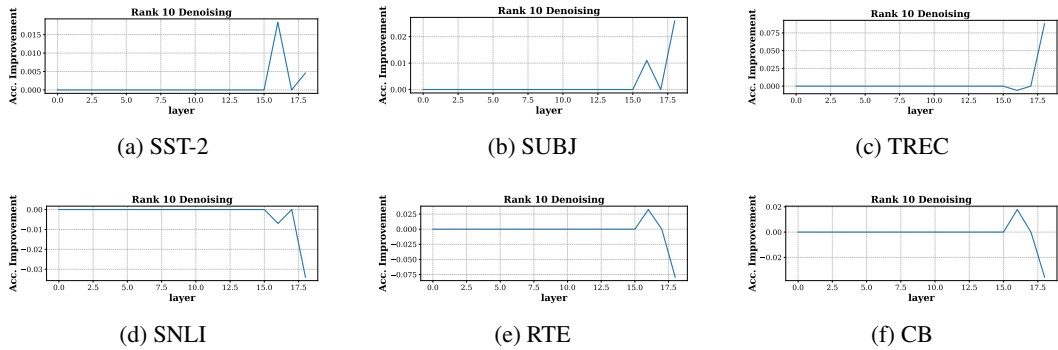

Figure 47: Accuracy gains of rank-10 denoising over layers on all datasets with Gemma-2B

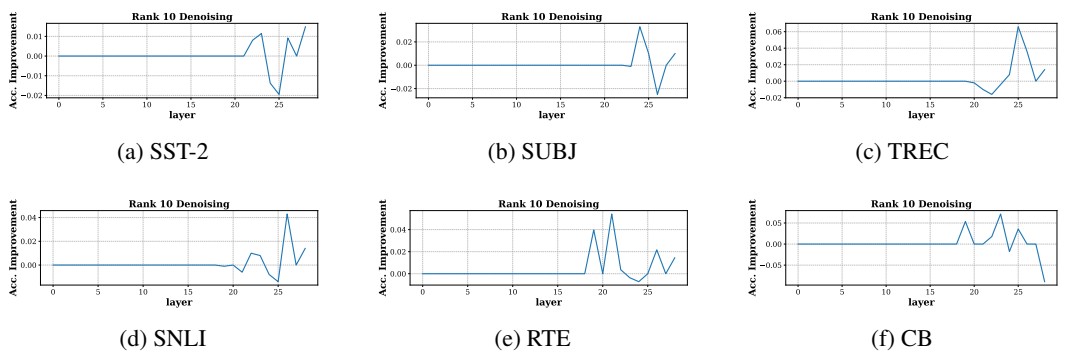

Figure 48: Accuracy gains of rank-10 denoising over layers on all datasets with Gemma-7B

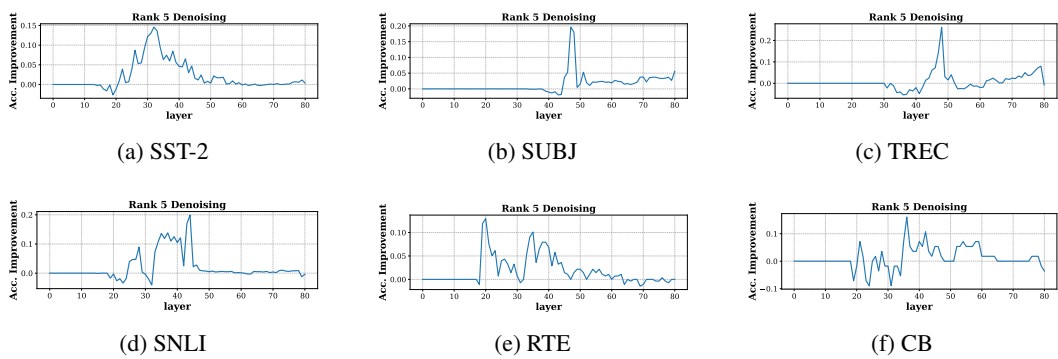

Figure 49: Accuracy gains of rank-5 denoising over layers on all datasets with Llama2-70B

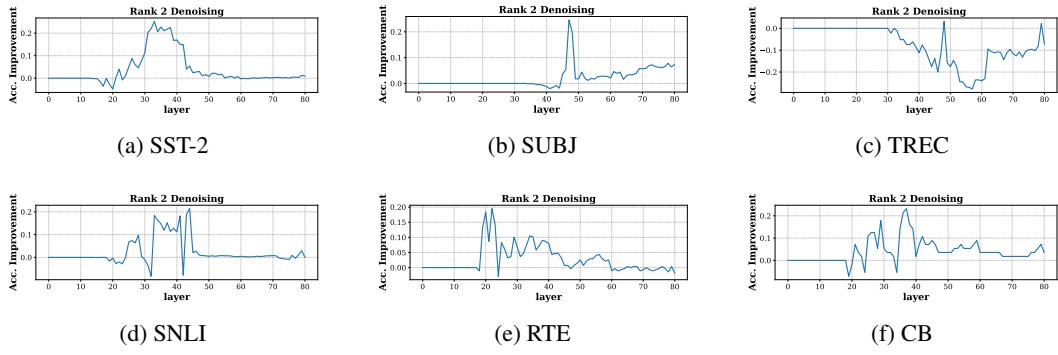

Figure 50: Accuracy gains of rank-2 denoising over layers on all datasets with Llama2-70B

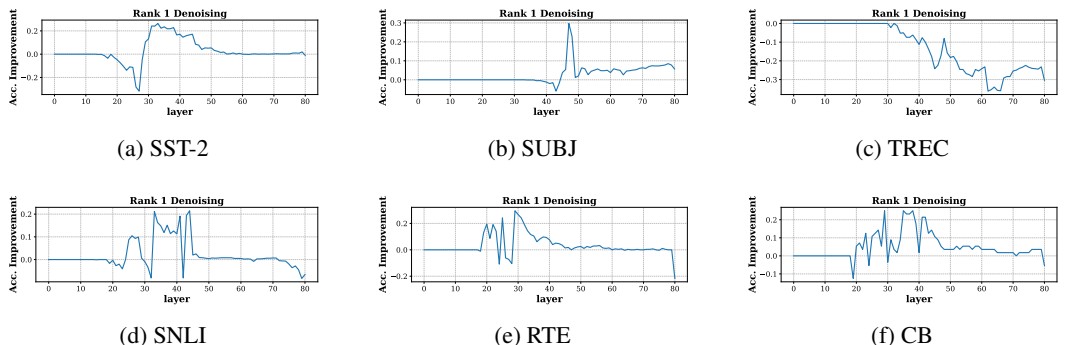

Figure 51: Accuracy gains of rank-1 denoising over layers on all datasets with Llama2-70B

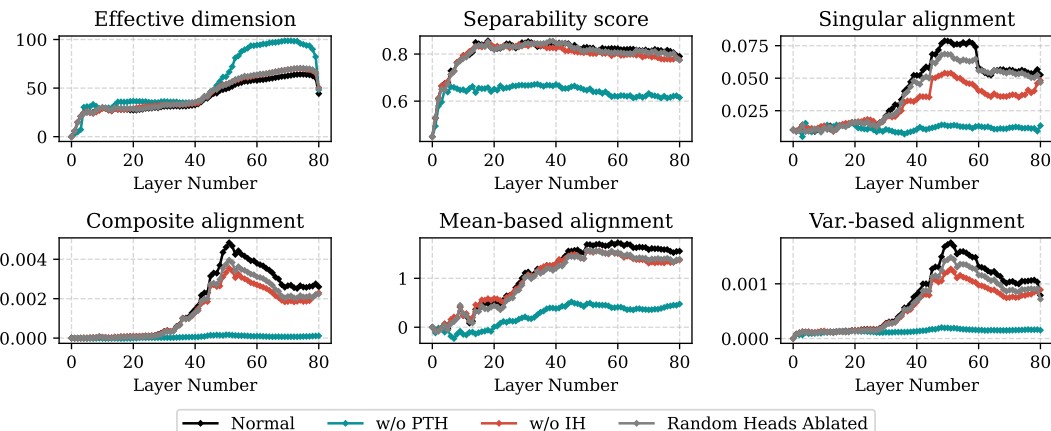

Figure 52: Average effects across datasets of attention heads ablation on the layer-wise separability and alignment measures of Llama2-70B hidden states in the zero-shot setting.

Table 16: Mean layer values of IH and PTH along with p-values of Mann-Whitney U test on Llama3-8B.

| % Level | IH mean layer | PTH mean layer | p-value of Mann-Whitney U test |
|---------|---------------|----------------|-------------------------------|
| 1%      | 15.883        | 9.5667         | $5.06 \times 10^{-7}$         |
| 2%      | 16.392        | 12.658         | $3.99 \times 10^{-5}$         |
| 5%      | 16.493        | 13.833         | $2.29 \times 10^{-5}$         |
| 10%     | 16.435        | 14.364         | $4.46 \times 10^{-6}$         |

Table 17: Mean layer values of IH and PTH along with p-values of Mann-Whitney U test on Llama3-70B.

| % Level | IH mean layer | PTH mean layer | p-value of Mann-Whitney U test |
|---------|---------------|----------------|-------------------------------|
| 1%      | 47.624        | 45.755         | 0.5573                        |
| 2%      | 46.273        | 46.977         | 0.1347                        |
| 5%      | 44.354        | 47.277         | $1.18 \times 10^{-7}$         |
| 10%     | 43.278        | 44.257         | $8.38 \times 10^{-4}$         |

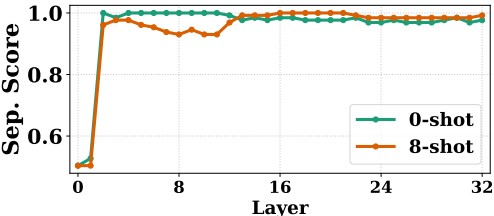
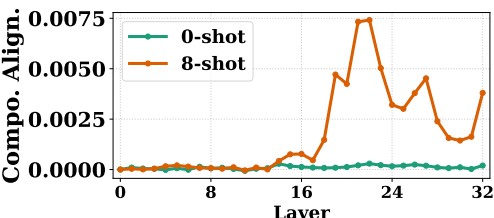

(a) Layer-wise separability score: 0-shot vs. 8-shot    (b) Layer-wise composite alignment: 0-shot vs. 8-shot

Figure 53: Layer-wise trends of separability score and composite alignment of 0-shot and 8-shot hidden states of Llama3-8B.

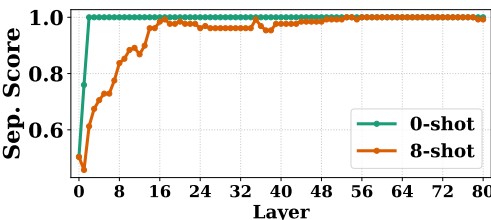
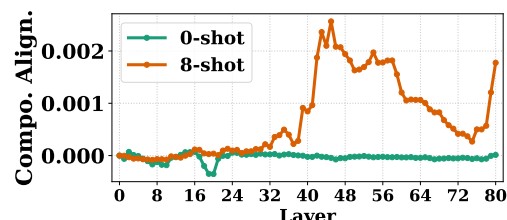

(a) Layer-wise separability score: 0-shot vs. 8-shot    (b) Layer-wise composite alignment: 0-shot vs. 8-shot

Figure 54: Layer-wise trends of separability score and composite alignment of 0-shot and 8-shot hidden states of Llama3-70B.

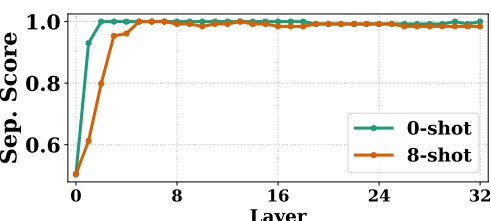
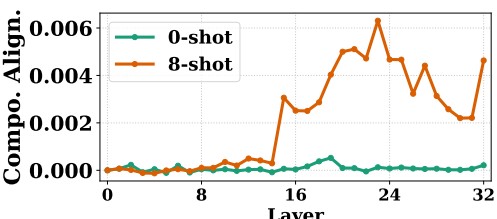

(a) Layer-wise separability score: 0-shot vs. 8-shot    (b) Layer-wise composite alignment: 0-shot vs. 8-shot

Figure 55: Layer-wise trends of separability score and composite alignment of 0-shot and 8-shot hidden states of Llama2-7B.

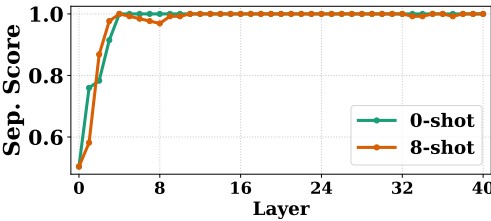
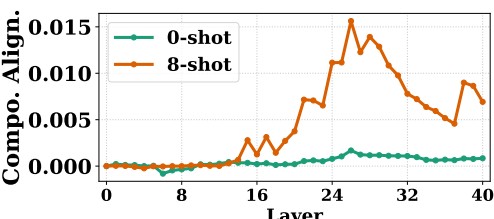

(a) Layer-wise separability score: 0-shot vs. 8-shot    (b) Layer-wise composite alignment: 0-shot vs. 8-shot

Figure 56: Layer-wise trends of separability score and composite alignment of 0-shot and 8-shot hidden states of Llama2-13B.

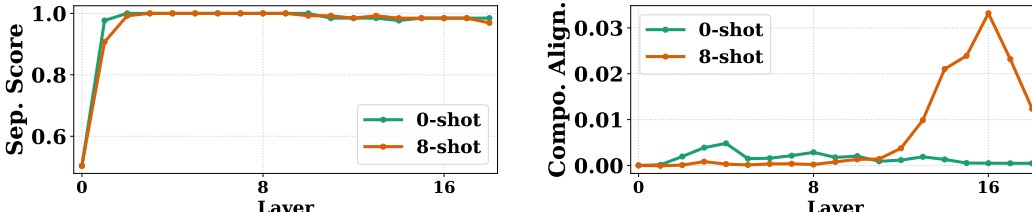

(a) Layer-wise separability score: 0-shot vs. 8-shot     (b) Layer-wise composite alignment: 0-shot vs. 8-shot

Figure 57: Layer-wise trends of separability score and composite alignment of 0-shot and 8-shot hidden states of Gemma-2B.

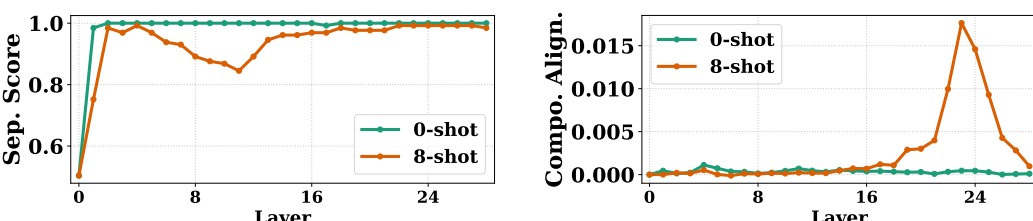

(a) Layer-wise separability score: 0-shot vs. 8-shot     (b) Layer-wise composite alignment: 0-shot vs. 8-shot

Figure 58: Layer-wise trends of separability score and composite alignment of 0-shot and 8-shot hidden states of Gemma-7B.

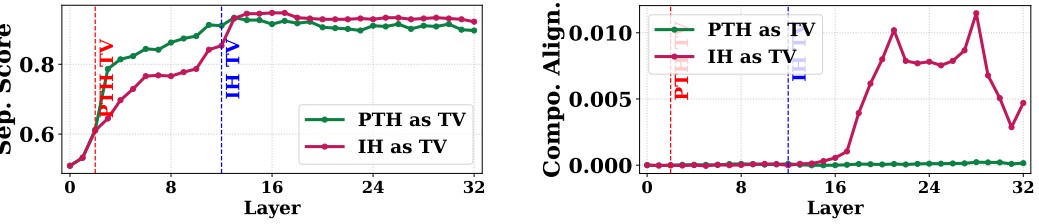

(a) Layer-wise separability score: PTH task vector     (b) Layer-wise composite alignment: IH task vector

Figure 59: Effects of injecting PTH and IH outputs as task vectors on the geometric properties of Llama3-8B zero-shot hidden states.

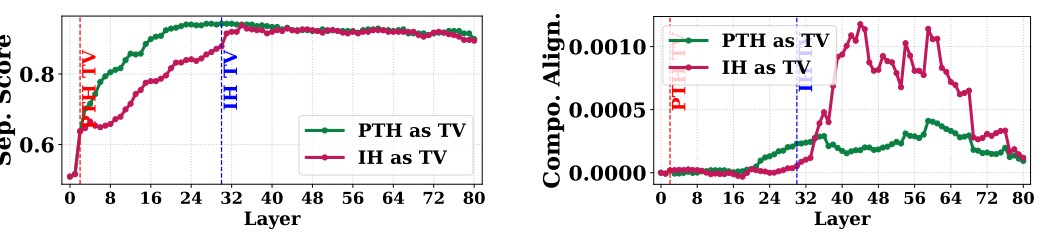

(a) Layer-wise separability score: PTH task vector     (b) Layer-wise composite alignment: IH task vector

Figure 60: Effects of injecting PTH and IH outputs as task vectors on the geometric properties of Llama3-70B zero-shot hidden states.

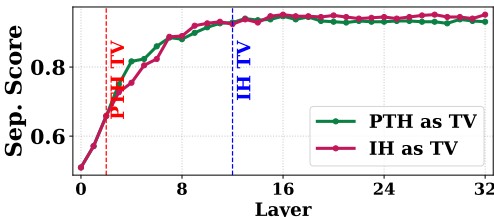
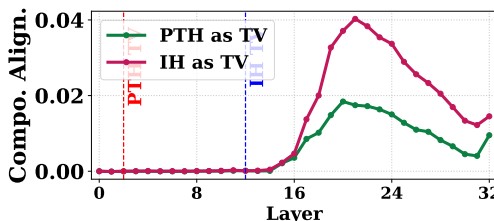

(a) Layer-wise separability score: PTH task vector

(b) Layer-wise composite alignment: IH task vector

Figure 61: Effects of injecting PTH and IH outputs as task vectors on the geometric properties of Llama2-7B zero-shot hidden states.

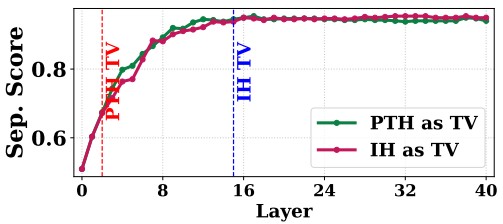
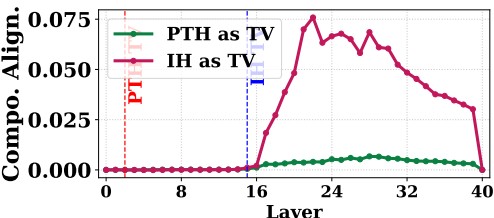

(a) Layer-wise separability score: PTH task vector

(b) Layer-wise composite alignment: IH task vector

Figure 62: Effects of injecting PTH and IH outputs as task vectors on the geometric properties of Llama2-13B zero-shot hidden states.

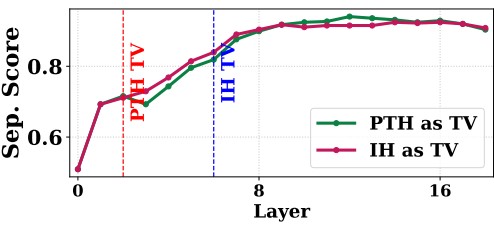
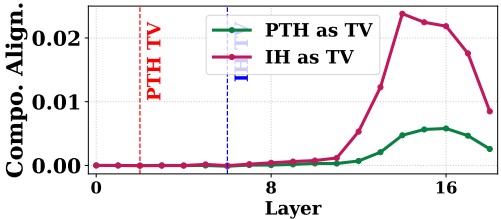

(a) Layer-wise separability score: PTH task vector

(b) Layer-wise composite alignment: IH task vector

Figure 63: Effects of injecting PTH and IH outputs as task vectors on the geometric properties of Gemma-2B zero-shot hidden states.

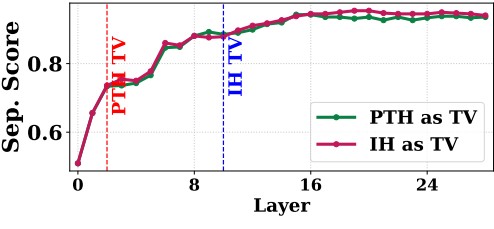
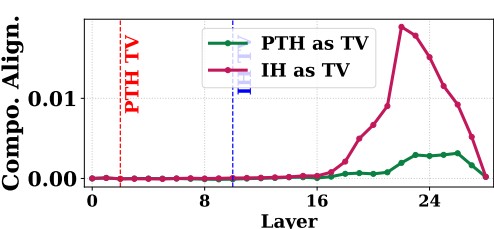

(a) Layer-wise separability score: PTH task vector

(b) Layer-wise composite alignment: IH task vector

Figure 64: Effects of injecting PTH and IH outputs as task vectors on the geometric properties of Gemma-7B zero-shot hidden states.

Table 18: Mean layer values of IH and PTH along with p-values of Mann-Whitney U test on Llama2-7B.

| % Level | IH mean layer | PTH mean layer | p-value of Mann-Whitney U test |
|---------|---------------|----------------|--------------------------------|
| 1% | 15.883 | 8.5833 | $5.62 \times 10^{-7}$ |
| 2% | 16.133 | 10.108 | $9.78 \times 10^{-9}$ |
| 5% | 16.301 | 12.092 | $4.12 \times 10^{-8}$ |
| 10% | 16.405 | 12.559 | $7.27 \times 10^{-13}$ |

Table 19: Mean layer values of IH and PTH along with p-values of Mann-Whitney U test on Llama2-13B.

| % Level | IH mean layer | PTH mean layer | p-value of Mann-Whitney U test |
|---------|---------------|----------------|--------------------------------|
| 1% | 21.312 | 17.094 | $1.75 \times 10^{-2}$ |
| 2% | 20.135 | 18.026 | $9.44 \times 10^{-2}$ |
| 5% | 19.960 | 16.927 | $6.08 \times 10^{-6}$ |
| 10% | 19.888 | 16.343 | $3.08 \times 10^{-13}$ |

Table 20: Mean layer values of IH and PTH along with p-values of Mann-Whitney U test on Gemma-2B.

| % Level | IH mean layer | PTH mean layer | p-value of Mann-Whitney U test |
|---------|---------------|----------------|--------------------------------|
| 1% | 9.0000 | 1.6667 | $8.58 \times 10^{-3}$ |
| 2% | 10.917 | 5.0000 | $7.91 \times 10^{-4}$ |
| 5% | 11.571 | 4.4286 | $2.22 \times 10^{-10}$ |
| 10% | 12.214 | 6.1667 | $2.48 \times 10^{-12}$ |

Table 21: Mean layer values of IH and PTH along with p-values of Mann-Whitney U test on Gemma-7B.

| % Level | IH mean layer | PTH mean layer | p-value of Mann-Whitney U test |
|---------|---------------|----------------|--------------------------------|
| 1% | 18.708 | 9.0833 | $2.01 \times 10^{-5}$ |
| 2% | 19.062 | 9.1250 | $1.96 \times 10^{-6}$ |
| 5% | 18.742 | 7.6212 | $6.31 \times 10^{-21}$ |
| 10% | 18.523 | 9.0530 | $4.44 \times 10^{-32}$ |

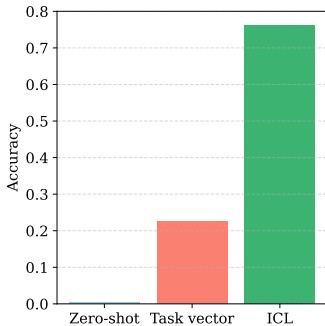

Figure 65: Average effect across datasets of steering Llama3-8B zero-shot hidden states using task vectors created from IH outputs.

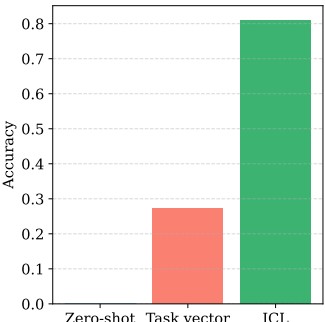

Figure 66: Average effect across datasets of steering Llama3-70B zero-shot hidden states using task vectors created from IH outputs.

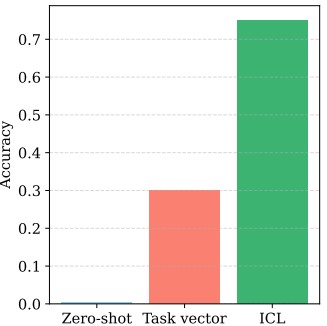

Figure 67: Average effect across datasets of steering Llama2-7B zero-shot hidden states using task vectors created from IH outputs.

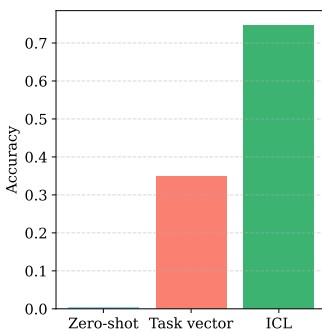

Figure 68: Average effect across datasets of steering Llama2-13B zero-shot hidden states using task vectors created from IH outputs.

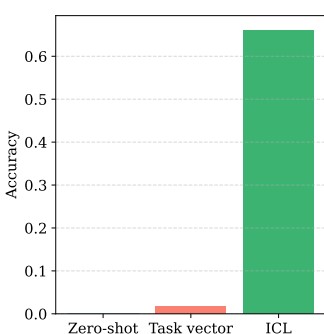

Figure 69: Average effect across datasets of steering Gemma-2B zero-shot hidden states using task vectors created from IH outputs.

Figure 70: Average effect across datasets of steering Gemma-7B zero-shot hidden states using task vectors created from IH outputs.

Table 22: Comparison of classification accuracy: training classifier v.s. direct decoding with zero-shot prompts.

| Model | 0-shot (%) | 0-shot+classifier (%) | Improvement |
|-------|-----------|----------------------|-------------|
| Llama3-8B | 0.30% | 73.71% | +73.41% |
| Llama3-70B | 0.00% | 78.97% | +78.97% |
| Llama2-7B | 0.31% | 75.63% | +75.32% |
| Llama2-13B | 0.36% | 79.71% | +79.35% |
| Llama2-70B | 0.24% | 79.02% | +78.78% |
| Gemma-2B | 0.00% | 75.56% | +75.56% |
| Gemma-7B | 0.02% | 75.43% | +75.41% |

Table 23: Comparison of classification accuracy: training classifier v.s. direct decoding with zero-shot prompts.

| Model | 8-shot (%) | 8-shot+classifier (%) | Improvement |
|---|---|---|---|
| Llama3-8B | 76.20% | 85.47% | +9.27% |
| Llama3-70B | 81.01% | 84.46% | +3.45% |
| Llama2-7B | 74.83% | 85.59% | +10.76% |
| Llama2-13B | 74.82% | 85.44% | +10.62% |
| Llama2-70B | 80.52% | 87.76% | +7.24% |
| Gemma-2B | 67.03% | 77.87% | +10.84% |
| Gemma-7B | 79.19% | 84.94% | +5.75% |

Table 24: Comparison of classification accuracy: regular ICL v.s. low-rank denoising of hidden states

| Model | 8-shot (%) | 8-shot+denoising (%) | Improvement |
|---|---|---|---|
| Llama3-8B | 76.20% | 78.58% | +2.38% |
| Llama3-70B | 81.01% | 82.40% | +1.39% |
| Llama2-7B | 74.83% | 77.68% | +2.85% |
| Llama2-13B | 74.82% | 79.52% | +4.70% |
| Llama2-70B | 80.52% | 80.72% | +0.20% |
| Gemma-2B | 67.03% | 67.25% | +0.22% |
| Gemma-7B | 79.19% | 66.11% | -13.08% |

