# OpenReview forum: "Unifying Attention Heads and Task Vectors via Hidden State Geometry in In-Context Learning"
_NeurIPS.cc/2025/Conference — NeurIPS 2025 poster_

### Official Review · Reviewer_LiD7 · 2025-06-22

**Clarity:** 3
**Significance:** 2
**Originality:** 3
**Rating:** 5
**Confidence:** 4

**Summary:**

This work presents a careful analysis of the inner workings of LLMs in the domain of ICL for classification tasks. It introduces separability and alignment as key geometric properties governing the success of ICL, formalizes them theoretically, and provides compelling empirical evidence that ICL operates as a two-stage process, first enhancing separability, then refining alignment, through the progression of information across model layers.

**Questions:**

How can the proposed interpretive framework for in-context learning (ICL) be extended to support generation and other non-classification tasks, and does it reveal similar alignment and separability patterns in those settings?

Can alignment mechanisms in zero-shot ICL be improved by explicitly leveraging the observed separability in hidden representations?

Important: what practical gains can such methods achieve in downstream performance?

**Ethical Concerns:**

["NO or VERY MINOR ethics concerns only"]

**Final Justification:**

My concerns have been comprehensively addressed in the rebuttal, and I have upgraded my rating.

**Limitations:**

adequately addressed

**Quality:**

3

**Strengths And Weaknesses:**

Strengths

* The central claim of a two-phase mechanism is substantiated through a comprehensive empirical evaluation involving seven models, six datasets, and multiple demonstration selection strategies (including random, kNN-based, and label-symbolized). The geometric framework, featuring a suite of alignment metrics (e.g., singular, composite, mean/var-based), is well-motivated and clearly described. Intervention-based experiments such as low-rank denoising and attention head ablations further strengthen the case, which validates semantic filtering hypothesis and distinguishes between the role of different attention heads in this process.

* Beyond quantitative metrics, the authors also offer qualitative insights into the semantics of hidden states. They show that label-relevant tokens emerge post-transition, and that semantic directions are filtered or preserved across layers in a way that aligns with task structure. This interpretability—achieved via SVD projection analysis and singular vector decoding—adds an important dimension to the work.

* The paper is clearly written, conceptually engaging, and well-structured. The logical flow of claims, evidence, and interpretation is easy to follow, making it an enjoyable read.

Weaknesses

* The scope of the analysis is narrowly focused on classification tasks within ICL, limiting its generality. The framework is not extended to generation or other types of ICL tasks. This raises concerns about the broader impact of the findings.

* While the analysis is rich, the practical implications are underdeveloped. For example, the paper convincingly shows that alignment is the main bottleneck in zero-shot settings, but does not propose a method to improve zero-shot classification by leveraging the separability already present in hidden representations. This leaves the work primarily as an interpretive contribution, rather than one with actionable outcomes.

* The paper emphasizes the connection between model components (e.g., attention heads) and hidden state geometry, but falls short in delivering empirical evidence for this claim. The identification of PTHs and IHs, though central to the proposed mechanism, is relegated to an appendix with little discussion in the main text. Moreover, the distribution of these heads across layers—which would directly support the two-phase hypothesis—is not reported.

---

> ### Author Rebuttal · Authors · 2025-07-31
>
> **Dear Reviewer LiD7,**
>
> Thank you for your insightful review of our manuscript, which we greatly appreciate. Below, we respond to the weaknesses and questions you raised.
>
> ---
>
> ## **W1 & Q1: Limitations of solely focusing on classification tasks**
>
> We appreciate your view that structuring our analysis solely around classification tasks seems limiting. Per your instruction, and to address your concern that our geometric framework of ICL may not generalize to generation tasks, **we repeat our experiments on a text generation dataset** with the following setup. The demonstrations and queries take the format "Write a favourable/unfavourable review of {subject}:" where {subject} is the name of a kind of food (e.g., tapas). The labels are GPT-generated favourable or unfavourable reviews about the {subject}. Eight demonstrations and a query are concatenated to form each ICL-style prompt, and we compute our separability and alignment measures, as well as the effective dimension, on the dataset of prompts. The query labels needed for training the classifier to compute the separability score are determined based on whether a favourable or unfavourable review is requested in the query, and we also compute the alignment measures w.r.t. the unembedding vectors of the "positive" and "negative" tokens to address the issue that labels for generation tasks comprise multiple tokens and take indefinite forms. Below, we report the results concerning the separability score and singular alignment (as a representative of alignment measures) across all layers of Llama3-8B in the zero-shot and ICL settings. (All experimental results presented in the rebuttal are conducted on Llama3-8B.)
>
> |SingularAlignment|...|14|15|16|17|18|19|20|21|22|23|24|25|26|27|28|29|30|31|32|
> |----------------|---|--|--|--|--|--|--|--|--|--|--|--|--|--|--|--|--|--|--|--|
> |zero-shot|...|0.0304|0.0026|0.0163|0.0132|0.0200|0.0158|0.0147|0.0076|0.0057|0.0084|0.0088|0.0054|0.0064|0.0135|0.0155|0.0278|0.0278|0.0310|0.0108|
> |ICL|...|0.0270|0.0360|0.0410|0.0361|0.0545|0.0899|0.0762|0.0836|0.0918|0.0802|0.0731|0.0717|0.0766|0.0781|0.0547|0.0498|0.0613|0.0626|0.0810|
>
> |Separability|...|14|15|16|17|18|19|20|21|22|23|24|25|26|27|28|29|30|31|32|
> |------------|---|--|--|--|--|--|--|--|--|--|--|--|--|--|--|--|--|--|--|--|
> |zero-shot|...|0.992|0.992|0.985|0.985|0.977|0.969|0.969|0.961|0.961|0.969|0.969|0.969|0.969|0.969|0.977|0.969|0.977|0.992|1|
> |ICL|...|1|1|1|1|1|1|1|1|1|1|1|1|1|1|1|1|1|1|1|
>
> *Table 1: Layer-wise trend of singular alignment and separability score for the generation task: ICL (8-shot) and zero-shot*
> The results demonstrate the rise in alignment measures in the middle layers that is unique to the ICL setting and show that both ZSL and ICL hidden states have high separability, thereby demonstrating the **generalizability of our framework to generation tasks**.
>
> ---
>
> ## **W2 & Q2: Limited Practical Implication & Improving Zero-Shot Performance**
>
> We respect your concern about the practical significance of our work, particularly for zero-shot learning. We would like to point out that our findings about the highly separable zero-shot hidden states can directly be used to improve zero-shot performance, without having to explicitly change the alignment properties of the hidden states as you mentioned. Concretely, we can simply train a classifier on the hidden states of some zero-shot prompts as labels, and use its predictions on other prompts as the predicted labels. For the Llama3-8B model, this approach increases the **zero-shot accuracy averaged across datasets from 0.3 to 73.71** (i.e., the mean separability score for the Llama3-8B final layer hidden states presented in the paper). Apart from being lightweight, this approach also requires only a small amount of labeled data. Using just 30 labeled prompts for training achieves 72.94 accuracy on the SST-2 dataset (in the calculation of the separability score, half of the prompts in the dataset are used to train the classifier).
>
> ---
>
> ## **W3 Part 1: Direct Evidence for the Geometric Significance of IHs and PTHs**
>
> We agree with your suggestion that evidence directly supporting the association between alignment and IHs, as well as separability and PTHs beyond the ablation experiments, will make our case more convincing. Hence, we conduct experiments where we construct task vectors using the outputs of top-scoring IHs and PTHs and add the task vectors to the final token's zero-shot hidden state. Due to the additive nature of attention heads' outputs, this experiment mimics the actual effect of IHs and PTHs on the hidden state residual stream. We inject the PTH task vector at layer 2 and the IH task vector at layer 16 of the 32-layered Llama3-8B. Then we directly record the dynamics of the separability and alignment measures near the layers of injection. We report the separability score for the PTH task vector and output alignment (logit lens accuracy) for the IH task vector (averaged across datasets). The results are reported below.
>
> |Separability Score|0|1|2|3|4|5|6|7|8|9|10|...|
> |----------------|--|--|--|--|--|--|--|--|--|--|--|---|
> |zero-shot (PTH as TV)|0.4491|0.4532|0.5189|0.6960|0.6924|0.7121|0.7396|0.7495|0.7689|0.7832|0.8059|...|
> |zero-shot|0.4491|0.4491|0.4651|0.5178|0.5548|0.5989|0.6340|0.6797|0.7199|0.7331|0.7873|...|
>
> |Output Alignment|...|14|15|16|17|18|19|20|21|22|23|24|25|26|27|28|29|30|31|32|
> |---------------|---|--|--|--|--|--|--|--|--|--|--|--|--|--|--|--|--|--|--|--|
> |zero-shot (IH as TV)|...|0|0|0.4431|0.4524|0.4605|0.4526|0.4533|0.4533|0.4537|0.4803|0.5074|0.5158|0.4985|0.5261|0.4633|0.4736|0.4933|0.4553|0.5476|
> |zero-shot|...|0|0|0|0|0|0.0003|0|0.0093|0.0006|0.0009|0.0025|0.0015|0.0019|0.0032|0.0017|0.0023|0.0003|0.0010|0.0023|
>
> *Table 2: Dynamics in separability and alignment measure after task vector injection using PTH and IH outputs as task vectors*
>
> The significant increase in separability and alignment after the injection of PTH and IH outputs respectively is strong and direct evidence of the geometric significance of PTHs and IHs.
>
> ---
>
> ## **W3 Part 2: Location and Identification of IHs and PTHs**
>
> Per your suggestion, we have calculated the mean layer index of the top 1%, top 2%, top 5%, and top 10% PTHs and IHs for Llama3.1-8B. We have also performed a Mann-Whitney U test to examine whether the difference in the layer distributions is significant, with PTHs significantly appearing earlier than IHs at all four levels. The results below show that PTHs indeed significantly precede IHs at all levels, thereby supporting our two-stage hypothesis regarding ICL. We will add these results, together with the explanation of how IHs and PTHs are identified which is currently in **Appendix G**, to Section 5.3 of the camera-ready version of this paper.
>
> |%Level|IH mean layer|PTH mean layer|p-value of Mann-Whitney U test|
> |------|-----------|------------|---------------------------|
> |1%|15.883|9.5667|5.0594e-07|
> |2%|16.392|12.658|3.9983e-05|
> |5%|16.493|13.833|2.2897e-05|
> |10%|16.435|14.364|4.4557e-06|
> *Table 3: Mean layer index of IHs and PTHs and the significance of difference in layer distributions at different percentage levels*
>
> ---
>
> ## **Q3: Implications for Downstream Performance**
>
> We agree with you that connecting our analysis to downstream performance is important. Previously, in Section **W2 & Q2: Limited Practical Implication & Improving Zero-Shot Performance**, we already demonstrated how our findings can be leveraged to improve zero-shot performance. Moreover, our findings can also be leveraged to improve ICL performance. We would like to first draw your attention to Section 5.2 and Fig. 4 in particular, where we present a considerable increase in logits lens accuracy achieved through low-rank denoising, which exploits the alignment structure of the hidden states with label unembedding vectors. We apply rank-5 denoising to Llama3-8B final layer hidden states. As a result, we can increase the final layer output accuracy averaged across six datasets as reported in Table 4.
>
> Furthermore, the high separability score of ICL hidden states presented in Section 5.2 and Fig. 2 implies that we can also improve ICL accuracy by training a classifier on several ICL hidden states and using its prediction on the hidden states of the remaining prompts to get the label, resulting in an ICL calibration label akin to the one proposed by [1]. Using this method, we also manage to significantly increase the average ICL accuracy as shown in Table 4.
>
> |Method|ICL|ICL+Denoising|ICL+Classifier|
> |------|-----|----------------|----------------|
> |Accuracy|76.20|78.59|85.47|
> *Table 4: Improving ICL accuracy using denoising or classifier as calibration methods*
>
> In summary, **our findings lead to two methods to improve ICL performance that are lightweight, effective (the low-rank denoising method is also supervision-free), and can be seamlessly integrated into any ICL scenario involving text classification.**
>
>
> # References
>
> [1] Cho, Hakaze, et al. "Token-based decision criteria are suboptimal in in-context learning."

---

> > ### Comment · Reviewer_LiD7 · 2025-08-05
> >
> > My concerns have been comprehensively addressed in the rebuttal, and I have upgraded my rating.

---

> > > ### Author Response · Authors · 2025-08-06
> > >
> > > Thank you very much for your thoughtful reconsideration and for updating your rating. We're glad to hear that the rebuttal successfully addressed your concerns, and we sincerely appreciate your engagement with our work.

---

### Official Review · Reviewer_kMpH · 2025-06-30

**Clarity:** 3
**Significance:** 2
**Originality:** 2
**Rating:** 2
**Confidence:** 5

**Summary:**

This paper proposes a unified framework to explain the mechanism of in-context learning (ICL) by analyzing two geometric properties of query hidden states: **separability** and **alignment**. The key findings include:
1. **Early layers** (dominated by Previous Token Heads, PTHs) enhance separability (distinguishing hidden states of queries from different classes);
2. **Later layers** (dominated by Induction Heads, IHs) optimize alignment between hidden states and label unembedding vectors while filtering out task-irrelevant semantics (as visualized in **Figure 1B-D**).

The framework unifies the roles of attention heads (PTHs/IHs) and task vectors, explaining why IHs’ outputs serve as effective task vectors.

**Questions:**

1. Kahardipraja et al. (2025) also provide different genre of heads in ICL, as well as function vectors. I understand it is a cocurrent paper, but I would like to see the author's detailed comparision of the deferences in terms of definitions of heads and function vector. Indeed, [17] Jiang et al. (2025)'s finding is also similar (two phase as compression-expansion) but also contradict (task vector is formed at 17th layer similar to Merullo et al. (2024)) to the author's, and the (15) in their paper is the correct task vector definition according to Hendel et al (2023).

2. In terms of Hendel et al (2023), what should be the task vector and the black-box task-vector leveraging function, and can you uncover the black-box nature or its relationship to latent steering vectors of labels?

3. Han et al. (ICML Spotlight 2025) provided that Task Decodability (TD) quantifies how well latent tasks can be inferred from intermediate representations that predict ICL performance, and early layers count more-early-layer finetuning outperforms late-layer tuning. This contradicts the author's view where the former layers' role didn't change too much and ICL make little changes, compared with the latter layer's role in aligning to the label unembedding. My intuition is because the author doesn't refer to the task vector in Hendel et al (2023)'s definition, which, if exists would appear in mid-to-late layers. Also, from my own experimental experience, the sentence/token level semantic classification task's task vector defined in Hendel et al (2023) and its f(x, h) are non-trivial and non-linear -- The task vector extracted from "This book is bad: negative; The movie is fun: positive; Today I'm unhappy:", cannot be simply adopted to any query just through a simple addition in the latent space and produce a correct answer, and the function vector leveraging function f(x, h) is a black box. For single token case, only the single token factual recall ICL task in Merullo et al. (2024) is true in terms of the equivalence of function vector and latent steering vector where f(x, h) is linear. Given the existing findings, the author fail to concretly grasp the existing literature's finding, make conclusions on too restricted setting, and the impact is too limited despite the presentation is clear.

In all, I found the anthors misalign with the broad task vector literature and didn't make a good comparision and summerization. For example, the task vector defined in Hendel et al (2023), Merullo et al (2024), [17] Jiang et al. (2025) emerged in early-to-medium layers, not refering the the last layer's steering vector but indeed warrent greater interest in terms of In Context Learning (the task vector and its function vector leveraging function f(x, h) is not simply a steering in the latent space that help align the label unembedding). In fact, if the author prefer to define latent steering vectors that help align label unembedding as the task vector, it somewhat similar to the broad literature I mentioned above regarding using steering vector / linear mode connectivity started 2022, but these important discussions are currently missing in the author's manuscript.

**Ethical Concerns:**

["NO or VERY MINOR ethics concerns only"]

**Final Justification:**

The manuscript currently exhibits an insufficient and sometimes incorrect understanding of task vectors and ICL, and the necessary citations and conceptual distinctions are largely missing.

**Limitations:**

See Weakness and Question above. Since the task vector literature is so large, it is hasty to only focus on single token semantic classification, in the scenario that single token factural-recall (Merrulo et al) and other tasks have been well-documented. Not to mention that I heavily question about the authors' familarity about task vector celebrated many years in the community.

**Paper Formatting Concerns:**

No significant concerns.

**Quality:**

3

**Strengths And Weaknesses:**

**Strengths**:
1. **Theory**: link separability/alignment to ICL classification accuracy, providing a geometric interpretation of ICL mechanisms.
2. **Comprehensive experiments**:
   - Covers 7 major models (Llama2/3, Gemma) and 6 tasks (sentiment analysis/NLI, etc.);
   - Multi-angle validation (layer-wise dynamics, attention head ablation, task vector intervention). For example, **Figure 6** shows PTH ablation reduces separability, while IH ablation disrupts alignment.
3. **Clear mechanistic insights**:
   - Reveals a two-stage phase transition (**Figure 2**: separability ↑ in early layers, alignment ↑ in later layers);
   - Explains the efficacy of IHs’ outputs as task vectors (**Figure 7**: adding IH outputs improves zero-shot hidden state alignment).

**Weaknesses**:
1. **Incomplete literature research**. There is a vast scope (from 2022 to now) of "task vector" literature that the author fails to discuss their relationship to their work, to name a few:

[1]. Jiang et al. (ICLR Oral 2025) Unlocking the Power of Function Vectors for Characterizing and Mitigating Catastrophic Forgetting in Continual Instruction Tuning

[2]. Li et al. (ICLR Oral 2025) When is Task Vector Provably Effective for Model Editing? A Generalization Analysis of Nonlinear Transformers

[3]. Yoshida et al. (ICLR Poster 2025) Mastering Task Arithmetic: Jp as a Key Indicator for Weight Disentanglement

[4]. Cao et al. (ICLR Poster 2025) ParamΔ for Direct Weight Mixing: Post-Train Large Language Model at Zero Cost

[5]. He et al. (TMLR 2025) Localize-and-Stitch: Efficient Model Merging via Sparse Task Arithmetic

[6]. Lee et al. (ACL 2025) Dynamic Fisher-weighted Model Merging via Bayesian Optimization

[7]. Kahardipraja et al. (2025) The Atlas of In-Context Learning: How Attention Heads Shape In-Context Retrieval Augmentation

[8]. Zhang et al. (2025) Understanding Fact Recall in Language Models: Why Two-Stage Training Encourages Memorization but Mixed Training Teaches

[9]. Zeng et al. (2025) Efficient Model Editing with Task Vector Bases: A Theoretical Framework and Scalable Approach

[10]. Wortsman et al. (2022) Model soups; averacing weichts of multiple fine-tuned models improves accuracy without increasing inference time.

[11]. lharco et al. (2023). Editing Models with Task Arithmetic.

[12]. Matena et al. (2022). Merging Models with Fisher-Weighted Averaging.

[13]. Jin et al. (2023) Dataless Knowledge Fusion by Merging Weights of Language Models.

[14]. Bu et al. (2025) Provable In-Context Vector Arithmetic via Retrieving Task Concepts

[15]. Merullo et al. (2024)  Language models implement simple word2vec-style vector arithmetic

[16]. Jiang et al. (2024) On the origins of linear representations in large language models

[17]. Jiang et al. (2025) From Compression to Expansion: A Layerwise Analysis of In-Context Learning

[18]. Han et al. (ICML Spotlight 2025) Emergence and Effectiveness of Task Vectors in In-Context Learning: An Encoder Decoder Perspective

[19]. Zhou et al. (2024) Going Beyond Linear Mode Connectivity: The Layerwise Linear Feature Connectivity

[20]. Zhou et al. (2024) On the Emergence of Cross-Task Linearity in the Pretraining-Finetuning Paradigm

2. **In-accurate and In-sufficient Discussion of the Definition of Task Vector**. Indeed, the author would find their "task vector" view is more like a "latent steering vector" (in Jiang et al. (2024)) that utilized in many work not in the ICL setting I listed above. However, in vision settings and non In-Context setting, the work listed above have different definitions and formulation of (general) task vector but indeed similar to the author's, despite the author considers ICL scenario. However, according to Hendel et al. (2023), the function vector leveraging function f(x, h) is indeed not linear and remain a black-box function, only when operating on "single token factual-recall task" in Merullo et al. (2024), the "task vector" defined in Hendel et al. (2023) would operate a simple vector addition in the latent space. In the semantic classification task the author defined, the task vector defined in Hendel et al. (2023) is not a latent steering vector, and should not operate by vector addition as in Merullo et al. (2024). The task vector in Hendel et al. (2023) is, when attending to any query with the black box function vector leveraging function f(x, h), can conduct prediction in one-shot. (The "task vector" in Hendel et al. (2023) is more than linear latent steering vector.)

3. **Missing Discussions over ICA**. According to Yamagiwa et al. (2023), words embedding would be better whitening through ICA instead of PCA, where each embedding can be expressed as a composition of a few intrinsic interpretable axes. Therefore, in the author's scenario, the IHs' "filtering irrelevant task information" is like filtering those task-irrelecent ICA-decomposed interpratable components in the ICL prompt. Therefore, experiments with ICA should be complementary to the author's PCA's ones.

4. **Broader Impact might be mild**. See Questions.

Yamagiwa et al. (2023) Discovering Universal Geometry in Embeddings with ICA

---

> ### Author Rebuttal · Authors · 2025-07-31
>
> **Dear Reviewer kMpH,**
>
> Thank you for your careful review of our manuscript. Below we respond to the weaknesses and questions you raised.
>
>
> ## **W1 & W2 part 1: Insufficient discussion of task vector literature**
>
> We appreciate your suggestion to clarify our connection to the task vector literature. As you noted, the task vector literature is vast, with each work offering unique definitions and settings. **We reiterate that the main focus of our work is the geometry-inspired, process-oriented exposition of the ICL mechanism. That is why we focus only on task vectors in ICL settings—because they are relevant and serve as a motivation for our framework.**
>
> Following your suggestion, we will include a Discussion Section in the camera-ready version situating and fitting our work within the ICL-related task vector literature. The section will include the following parts:
>
> 1. **In-weight Task Vector.** The term Task Vector originates from the parameter difference between a fine-tuned model on a specific task and its pre-trained counterpart, typically constructed via Linear Mode Connectivity [2–6, 9–13, 19, 20, 29]. However, ICL involves no explicit parameter updates. Thus, despite sharing the same term, task vectors in ICL and those based on in-weight changes are conceptually hard to align. **We will include a brief discussion of in-weight task vectors as necessary background.**
> 2. **Linear Task Vector.** Linear Task Vectors [8, 14–16] typically shift hidden states from a query embedding (e.g., China, England) to a target expression (e.g., Beijing, London) in factual recall tasks. Our definition generalizes this idea: as shown in Figure 1, our task vectors also transform embeddings with mixed semantics (e.g., China, or a positive sentence) toward specific attributes (e.g., Beijing, or positive), and are not limited to simple translation in hidden space. This perspective both aligns with prior work and extends it. **We will include these references to highlight the connection.**
> 3. **Black-box Task Vector.** Some works [1, 7, 17, 18] define task vectors loosely as any vector that enables patching-and-improvement, without specifying their form. Our additive task vector, after passing through Transformer layers, can similarly induce such effects, thus serving as a precursor to black-box task vectors. **We will clarify this point in the final version.**
>
>
> ## **W3: ICA as an alternative method for interpreting hidden state dynamics**
>
> We agree that ICA is a viable method for extracting semantic directions. However, in our context, our SVD-based approach is preferable for two reasons.
>
> **Theoretical reason.** Our analysis targets the geometric evolution of hidden states. The top right singular vectors used in our analysis identify directions of maximum variation, crucial for capturing alignment with unembedding vectors. In contrast, the ICA components (columns of $\mathbf{A}$ in $\mathbf{H} = \mathbf{S}\mathbf{A}$) **lack geometric significance.** The column order is arbitrary, rendering most experiments in Section 5.2 infeasible.
>
> **Practical reason.** **ICA requires $n\geq d$**, but the 70B models we use have $d=8192$, exceeding most dataset sizes. The paper you mentioned uses $d=300$ and manually inspect activations of ICA components for semantic interpretation. In contrast, our semantic analysis depends on measuring component alignment with unembedding vectors which equals model dimension.
>
> Nonetheless, per your suggestion, we repeated the semantic filtering and retention experiment on SST-2 with Llama3-8B. Below is the result at layer 25:
>
> |layer 25|retained tokens|filtered tokens|
> |---|---|---|
> |SVD|extra, hud, neutral, rated, herent|PointerException, ldr, anus, ipt, allon|
> |ICA|elo, raries, injury, .DropDown, offensive|dead, ets, seksi, @Spring, 견|
>
> It shows that SVD and ICA yield similar outcomes; SVD even decodes the SST-2-relevant "neutral" token.
>
> ## **W4 & Q3 Part 2: Limitations of focusing solely on classification tasks**
>
> Please see Section **W1: Limitations of solely focusing on classification tasks** in our response to **Reviewer Ua9F**, where we show our geometric two-stage ICL hypothesis extends to text generation tasks.
>
>
> ## **Q1: Comparison between [7]'s findings and ours**
>
> Thank you for highlighting this relevant. Below, we outline two key distinctions.
>
> **Study setting.** Paper [7], despite having 'in-context learning' its title, studies rather a general RAG setting where the model can draw on both internal knowledge and external sources. Their "context" includes both prompt and retrieved information, which allows them to define head types (e.g., in-context heads) based on which part of the context influences outputs. **In contrast, our setting uses classical ICL: the model must learn an input-output mapping from a few-shot prompt and respond to a query.** This setting justifies our focus on classification, which exemplifies the mapping learning process. **The importance of IHs and PTHs in this setting has already been well-documented ([21]–[25]) and is considered common knowledge in the ICL interpretability community**, as stated in our Introduction and Related Work.
>
> **The approach of study** [7] identifies the attention heads with special roles by measuring the association between the activation of attention heads or their distribution of attention weights, to the output logits. For instance, their 'task heads' is defined to be the heads whose attention weights to the question in the prompt have a large effect on the model outputs. Then they inject such heads' outputs as task vector into the hidden state of a prompt with a different question and recovers the same answer. This approach of characterizing attention heads can be found in other studies([26],[27]), and is conceptually akin to the ablation-based approach in [23][24], in that they characterize heads only by **how much** they influence the model outputs, but not by **how** they influence the outputs through LLM internal workings
>
> **This is why our study is innovative because we propose a process-oriented perspective of analyzing the how attention heads inform LLM outputs**. Thus our approach to attention heads and function vectors is that we first find that IHs induce the increase in alignment in the two-stage ICL process, and then we succesfully construct task vectors out of IH outputs because we know that alignment is missing in the zero-shot setting.
>
>
> ## **Q2, Q3 Part 1, W2 Part 2: Hendel et al.'s task vectors—linearity, location, and adaptability**
>
> We respect your concern that the linearity of the task vector defined by Hendel et al. is evident in [15]'s setting (Country-Capital etc.) that mimics the simple word2vec scenario, but not as evident in classification tasks. We have addressed this issue in the footnote at **line 130**. The Hendel et al.-styled hidden state patching can be reformulated as an addition where the added vector is the difference between the ICL and zero-shot hidden states.  Since ICL hidden states are more aligned with label unembeddings than zero-shot ones, the difference vector also increases alignment when added to zero-shot states. We repeat Hendel et al.'s experiment with SST-2 and Llama3-8B at all layers to verify this. We have also decoded the top tokens from the patched ICL hidden state, the original hidden state and the difference vector extracted from one example.
>
> |Index|...|14|15|16|17|18|19|20|21|22|23|24|25|26|27|28|29|30|31|32|
> |---|---|---|---|---|---|---|---|---|---|---|---|---|---|---|---|---|---|---|---|---|
> |Accuracy| ...|0.0321|0.0883|0.2752|0.2282|0.2523|0.3165|0.3429|0.3372|0.3601|0.4587|0.4736|0.4817|0.5000|0.4759|0.4966|0.4977|0.4862|0.4885|0.5172|
>
> (Before layer 14, accuracy is ~0)
>
> |Layer 26|original $h$s|ICL $h$s|difference|
> |---|---|---|---|
> | Tokens| ..\n, asiswa, _exempt, in, واء|negative, negatively, positive, positively, in|negative, positive, -negative, negative, Negative |
>
> The results show that the benefit of intervention increases with layer-consistent with our finding that alignment improves in deeper layer. The decoded tokens show the difference vector can steer zero-shot states toward label-aligned directions. The ~50% accuracy is expected: since we patch using a dummy query's ICL state following Hendel et al., there's a 50% chance that its label matches the current one in the two-labeled SST-2. Note that Hendel et al.'s result **does not show that task vector intervention is not effective in late layers.** It is because that the patched ICL hidden state in deep layers aligns with the dummy query's label and on non-classification tasks the probability of two queries sharing a label is smaller. [15] also note the effectiveness of task vector in quasi-classification tasks (e.g. Name to Nationality) in late layers, and [28] find that task vectors can emerge in many layers from middle to late, depending on the nature of the task. These findings can reconcile the discrepancies between our findings and the results you mentioned concerning the where task vectors are most effective
>
> # References
> Follows the paper list you mentioned in the review...
>
> [21] Olsson, C. et al. "In-context learning and induction heads."
>
> [22] Singh, A. K. et al. "What needs to go right for an induction head?..."
>
> [23] Cho, H. et al. "Revisiting in-context learning inference circuit..."
>
> [24] Crosbie, J., Shutova, E. "Induction heads as an essential mechanism..."
>
> [25] Song, J. et al. "Out-of-distribution generalization via composition..."
>
> [26] Jin, Z. et al. "Cutting off the head ends the conflict..."
>
> [27] Bansal, H. et al. "Rethinking the role of scale for in-context learning..."
>
> [28] Yang, L. et al. "Yang, Liu, et al. "Task vectors in in-context learning: Emergence, formation, and benefit."
>
> [29] Neyshabur, B. et al. "What is being transferred in transfer learning?"

---

> > ### Comment · Reviewer_kMpH · 2025-08-01
> >
> > I would like to thank the authors for their detailed response.
> >
> > I generally disagree with the authors’ summary and understanding regarding the high-level task/function vector deeply studied in the community. Task/function is the high-level capability requiring compression of the information embedded in the context. As the name suggests, the high-level task/function vector is intended to **be responsible for the task/function capability** across literature, rather than to serve primarily as a concrete steering or alignment vector in the latent space — the latter being merely a narrower-scope side/outcome effect of the high-level task vector. [2–6, 9–13, 19, 20, 29] are weight vectors that can be responsible for certain tasks and transferable between models.
> >
> >
> > In contrast, in the case of ICL [1, 7-8, 14–18], the main power of ICL is to (i) first identify the underlying function/task based on the prompt/context with certain confidence over the choice, then (ii) excute the function/task to the query. Here, as shown in those studies, the extracted high-level task/function information is extracted as task/function vector in the latent space, with subsequent layers utilizing this vector to conduct the function/task accompanying with the query representation. The increase of steering/alignment for label unembedding effect you validated is the process of (ii), a consequence/subsequent effect after (i) - formulating the correct function/task vector in former layers. For classification tasks, the high-level task/function is "classify the emotion/attitude of the text", and the low-level task implementation (conduct the function over the query) includes alignment to the label unembeddings.
> >
> > For example, your example could be something like: *“I don’t like it. Answer: positive. I like it. Answer:”* — meaning the model’s task is *“state the opposite of the actual sentiment”*, rather than the standard *“sentiment classification”*. In this case, the high‑level task/function vector should be different from that for a prompt stating the truth, such as *“I don’t like it. Answer: negative. I like it. Answer:”*. The role of the extracted high‑level task vector is to enable the query to **execute the function**, for the ease of latter aligning with the correct label unembedding in subsequent layers — i.e., once the correct high‑level task information is identified, the transformer can then perform the task for the given query with the task vector.
> >
> > Your rebuttal does not compare against your concurrent work \[17] Jiang et al. (2025), as I had asked. That paper’s interpretation of the task vector is more accurate and aligns better with mainstream understanding. Its use of compressed sensing tools is also quite reasonable, since the task/function vector reflects the outcome of a high‑level “grasp” of the task, rather than a low‑level execution mechanism. In need, the "back box" I mentioned mainly refer to the low‑level execution mechanism, where the function $f(x\_{query}, \mathbf{h})$'s formula ($\mathbf{h}$ as task vector, $x\_{query}$ as representation of the query information) is unknown, and the executation would finally yield the alignment to the label unembedding by utilizing both $\mathbf{h}$ and $x\_{query}$.
> >
> > Overall, while the literature on task vectors varies, they are in essence **critical vectors that determine whether a model can perform a given task/function**. Aligning the latent unembedding is merely a consequential side effect when comparing with- and without- task vector case (zero‑shot case) — and this consequence is easy to imagine, since the representation of the high-level task and its label should have relationship due to semantic relevance.

---

> > > ### Author Response · Authors · 2025-08-04
> > >
> > > **Dear Reviewer kMpH,**
> > >
> > > **Thank you for your feedback to our rebuttal. Below, we respond to the issues you raised.**
> > >
> > > **First, regarding your concern about our summary of the task vector literature:**
> > >
> > > We reiterate that in-weight task vectors from the task arithmetic literature ([2–6, 9–13, 19, 20, 29]) and the linear task vectors that operate by shifting hidden states ([1, 7, 8, 14–18]) differ substantially in definition, nature, and form. Claiming they are similar simply because they both encode task functionality is a **crude oversimplification of the critical differences between them**. Since our work focuses on ICL, where **no** model weight updates occur, extensive discussion of in-weight task vectors would be off-topic and dilute the core message of our geometry-inspired, process-oriented analysis of ICL. Even [17], which you view as an example of studying task vectors in the setting of in-context learning, **includes none of the in-weight task vector papers in the Task Representation subsection of its Related Works section**. Instead, it mainly discusses on [31–33], which are also our main focus in literature review. Moreover, in the same subsection, [17] states that **"In-context vectors enhance ICL through latent space steering"**, directly supporting our framing and analysis of task vectors in ICL.
> > >
> > > **Second, regarding your comment that the alignment process we observe is merely a side effect of executing the task represented by the task vector:**
> > > While we agree that for task with simple one-to-one input-output mapping (i.e. country-capital), the mechanism of task vector follow the two-stage mechanism you outlined with task execution in subsequent layers following by summary of task information into vector format in initial layers, we refer you to **W3 Part 1: Direct Evidence for the Geometric Significance of IHs and PTHs** in our response to **Reviewer LiD7**, where we present a different picture. We use IH outputs as task vectors following [32], injecting them into layer 16 of the 32-layer LLaMA3-8B. As shown in the lower part of **Table 2**, output alignment (logit lens accuracy) increases immediately after injection and rises all the way to 54.76%—well above the 36.11% random baseline—indicating the model has learned the respective tasks. **This suggests that in our classification setting, the two stages collapse into one:** understanding the task, executing it, and improving hidden state alignment occur as a unified process. This is consistent with Fig. 2 in the main text. Since the model can well separate queries of positive and negative sentiments in the zero-shot setting where it is unaware of the task, the task vector inherent in the ICL input which informs the model of the task outlined by the demonstrations has to be one that rotates and steers the already separated hidden states with the label unembeddings, which causes the increase in alignment measures iconic to ICL. These findings justify incorporating task vectors into our geometric analysis and offer new insights. **Based on your feedback, we will add these results to Section 5.3, along with a discussion comparing them to the two-stage mechanism described in prior work.**
> > >
> > > **Regarding your concern about differences in task types and datasets used in prior task vector studies, and about contrasts with [17]:**
> > > As noted in our rebuttal, classification tasks have already been studied in [15], so we are not the first to explore task vectors in this setting.
> > >
> > > **Moreover, the lack of prior work on classification tasks highlights the need for such analysis.** These tasks are more authentic (vs. synthetic tasks like lower case->upper case), more complex (with many-to-many input-output mappings), and better reflect ICL's core nature—learning input-output relationships. Thus, we believe analyzing task vectors in classification yields fresh insights.
> > >
> > > **Finally, while NIPS policy does not encourage comparisons among contemporary works, regarding the difference you noted between our findings and those of [17],** we want to pinpoint that there are also important similarities. In Section 5.3 of the main text, we inject task vectors from [32] into layer 30 of LLaMA2-70B (an early layer, as the model has 80 layers), achieving non-trivial accuracy across datasets. By contrast in the rebuttal, when using task vectors from [31], we observe greater effectiveness in later layers—unlike [17]. **We thus view these intriguing similarities and differences suggest the various interesting insights as new insights that analysis of task vectors on classification tasks can offer**, rather than as weaknesses of our work.
> > >
> > > **References**
> > >
> > > [31] Hendel, Roee, Mor Geva, and Amir Globerson. In-context learning creates task vectors.
> > >
> > > [32] Todd, Eric, et al. Function vectors in large language models.
> > >
> > > [33] Dongfang Li, Xinshuo Hu, Zetian Sun, Baotian Hu, Min Zhang, et al. In-context learning state vector with inner and momentum optimization.

---

> ### Comment · Reviewer_kMpH · 2025-08-08
>
> Dear Authors,
>
> Thank you for your response.
>
> **1. Task Naming**
> First, I would like to clarify that the task your paper focuses on should more accurately be termed **“single-token bi-classification”** rather than simply “classification.” This distinction is crucial to inform the reader properly. Other tasks such as *country-capital* are also regarded as a classification over a set of classes, and calling one task “classification” while excluding the other introduces confusion.
>
> **2. Interpretation of Phases**
> I respectfully disagree with your interpretation. My intention was to point out that your so-called *“separability phase”* (or clustering phase) corresponds to a **high-level information extraction** step—specifically, the extraction of low-dimensional task-relevant information. From an information-theoretic perspective, this aligns with the task vector formulation phase in [31, 15] or **“Compression”** phase described in [17].
>
> Similarly, what you refer to as the *“alignment phase”*—which I continue to interpret as the **task execution** phase (involving the query and the task representation)—aligns with the task execution phase in [31, 15] or the **“Expansion”** phase in [17], where the system executes the task after compressing relevant information.
>
> **3. On Task Vectors and Experimental Observations**
> The example I previously gave—“*I don’t like it. Answer: positive. I like it. Answer:*”—was intended to highlight a key distinction: in **single-token bi-classification**, the *task vector* defined in \[31] (*In-context learning creates task vectors*) does **not** appear to exist. In \[31], the task vector is such that, when attending to any query token in the latent space, it can produce the correct output with high probability in later layers.
>
> However, based on my extensive experimental experience, I could not find such a vector in the bi-classification setting. That is precisely why bi-classification tasks are **absent** in \[15] and \[31]. (I wonder why you claimed \[15] includes it.)
>
> That said, the **initial clustering phase does exist**, even though it does not result in the formation of a task vector in the sense of \[31]. This is not surprising—transformers are not inherently designed to form such task vectors; it occurs only in specific cases as studied in \[31] etc.
>
> **4. Missing Literature: Wang et al. \[34] and Beyond**
> Importantly, the information aggregation process, which corresponding to your first phases during the bi-classification task—*without* forming a task vector—is studied in detail in **Wang et al. \[34] (EMNLP Best Paper 2023)**. This paper is currently missing from your manuscript, but it is highly relevant. Wang et al. show that during the clustering phase, label-word anchors encode the task information, and that the subsequent “execution phase” performs the classification based on these anchors. This supports the interpretation that your “alignment phase” corresponds to their “execution phase” (label prediction).
>
> Notably, Wang et al. \[34], as well as \[31], \[15], and [17], all observe that the **first phase ends around similar layers**—even though bi-classification in Wang et al. does not result in a task vector as defined in [31]. Therefore, your argument that these phases collapse into one in your setting is, in my view, not persuasive and lack of understanding.
>
> **5. Gaps in Literature Review and ICL Understanding**
> Beyond the task vector and bi-classification literature, your manuscript also lacks adequate discussion of core **In-Context Learning (ICL)** works, especially \[35] and \[36]. This is particularly important given your attempt of developing a simple theory of ICL. In fact, the latent variable in the BMA perspective corresponds precisely to the first phase—recognition of the task identity—as emphasized in [15], [17], [31], and [34].
>
>
> To summarize, my concerns regarding the paper remain. The manuscript currently exhibits an **insufficient and sometimes incorrect understanding of task vectors and ICL**, and the necessary citations and conceptual distinctions are largely missing. I maintain my position that **the task vector should reflect high-level task information extraction**, which enables functionality—not low-level execution once that information is already extracted.
>
>
> As for NeurIPS policy, I only regarded it as an arXiv paper, also with flaws, imperfection and missing discussions. The heavy issues in [17] do not influence my judgment of this manuscript.
>
> Apologies for the delay in my response due to crazy schedule. Also wishing you a pleasant day!
>
> Warm regards,
>
> Reviewer kMpH
>
> ---
>
> **References**
>
> [34] Wang et al. *Label Words are Anchors: An Information Flow Perspective for Understanding In-Context Learning*. EMNLP Best Paper 2023.
>
> [35] Xie et al. *An Explanation of In-Context Learning as Implicit Bayesian Inference*
>
> [36] Zhang et al. *What and How Does In-Context Learning Learn? Bayesian Model Averaging, Parameterization, and Generalization*

---

> > ### Author Response · Authors · 2025-08-09
> >
> > **Dear Reviewer kMpH,**
> >
> > **Thank you for your feedback to our rebuttal. Below, we respond to the issues you raised.**
> >
> > **First, we would like to clarify some fundamental misunderstandings you have about our task setting**
> >
> > The datasets we use in our experiments are by no means restricted to binary classification. Out of the seven datasets we use in the experiments, 4 have classes $\geq 3$, 2 have classes $=6$. Moreover, we never claimed that tasks like *country-capital* with simple one-to-one input-output mapping are classification tasks. As we have stated very clearly, the value of classification tasks in the study of task vector in that it has complex many-to-many or many-to-one input-output mapping. Hence, your statement that our settings in [15] is also not correct. We would like to kindly refer you to **Appendix G, page 18** of [15] where they study *animal-category*, *name-nationality*, and *country-language*. These are typical multiway classification tasks with many-to-many or many-to-one mappings, with the mapping from inputs to outputs are **not injective** (as in *country-capital*), but surjective.
> >
> > **Second, regarding your interpretation of our two-phase hypothesis**
> >
> > First, we want to make it very clear that we **do not** see our separability-alignment characterization of ICL as the negation of the **task encoding-task execution** characterization to which you are committed. Instead, the highlight of our framework is that it leverages the process-oriented viewpoint and geometry-inspired approaches to provide mechanistic insights concerning some phenomena unexplained by the **task encoding-task execution** characterization primarily for classification tasks, for instance the black-box mechanism through which task vector takes effect. After all, we believe that the geometric characterization of ICL is ontologically as accurate as your characterization based on the abstract concept of the task, because in the end, learning the task inevitably boils down to the geometric process of the hidden states aligning the the unembedding vectors to the task-related labels. we do not see our characterization as being fundamentally opposed to the **task encoding-task execution** one, nor to the **compression-expansion** which you seem to have welcomed despite the fact that it still has a clear geometric feature. Moreover, we would like to clarify a misunderstanding: we do not claim that in classification tasks, the two stages collapse into one, that is against our main thesis and would be most whimsical. Instead, we are talking about the **mechanism of the task vector**. As we have demonstrated in our rebuttal to **Reviewer LiD7**, injecting the task vector imnmediately lead to the increase in the logit lens accuracy at the injected layer. However, if you view that the task vector first facilitates the task encoding, and then in subsequent layers it activates the task execution which promotes the alignment is correct, then the rise in the alignment would not be so swift.
> >
> >
> > **Third, regarding your concern about the task vector experiments in classification tasks**
> >
> > We regret to hear your belief in the ineffectiveness of task vectors and your opinion that the lack of experiments in previous works should be attributed to this factor, despite the fact that we have repeated indicated its effectiveness to you in Section 5.3, in our response to your review, and in our rebuttal to **Reviewer LiD7** who also confirmed the effectiveness. We sincerely hope that you may update your view about the compatibility of task vector and classification tasks based on **experimental results, but not based on personal convictions.**
> >
> > **Finally, regarding your requests for us to cite three more papers**
> >
> > First. we are glad that we seem to have pursuaded you into stop insisting that we should cite the task arithmetic papers ([2–6, 9–13, 19, 20, 29]) which are not remotely related to task vector in ICL settings. Regarding your referenced paper, we would like to first point out that the Bayesian accounts of ICL provided in [35] and [36] differ significantly from the two-stage explanation of ICL featuring a bipartition of early layers and late layers which you claim. A close reading of them reveal that in their theory, the task recognition or task encoding happens **with the increments of demonstrations, not with the progression of layers**, which make them irrelevant to the layer-wise analysis and characterization of ICL in [34], [17] and our work. Regarding [34], we would like to pinpoint that since we are not writing a survey paper, the our main focus is to demonstrate a clear and comprehensive picture of our findings, instead of presenting a painstakingly exhaustive literature search. Indeed, [18] which shares our topic does not cite [34] as well, but that does not prevent it from becoming an ICML spotlight paper.

---

### Official Review · Reviewer_4WZT · 2025-06-30

**Clarity:** 2
**Significance:** 3
**Originality:** 4
**Rating:** 5
**Confidence:** 4

**Summary:**

The main goal of the paper is to unify the task vector and the induction-head accounts for in-context learning, with a focus on classification problems. The authors show that, during in-context learning, the model representations go through two phases: First, the representations of the different classes are separated, then they are aligned with their corresponding unembedding vectors. The authors show, both through observational and causal studies, that predominantly the former process is carried out by previous token heads, and the latter through induction heads.

**Questions:**

- Why does the mean-based alignment results suggest PTH are more crucial than IH? Would we not expect IH ablations to create more dramatic effects.
- There is this recent paper named [Which Attention Heads Matter for In-Context Learning?](https://arxiv.org/abs/2502.14010), where the authors show that, while function vector and induction heads are connected early in training, function vector heads specialise to be functionally different types of heads, and that those are the parts of the model that drive in-context learning. However, you provide some findings where you can create task vectors simply through the outputs of induction heads. I think this is a fascinating difference of findings, and would be curious to hear your opinion on why this could be. Is it a matter of task, how you identify the induction heads, or something else?

**Ethical Concerns:**

["NO or VERY MINOR ethics concerns only"]

**Final Justification:**

Thanks for the very thorough response. I'm now further convinced of the reported findings, as they seem to hold through across several different methods of ablation and analysis. I also thank the authors for not only verbally addressing my last question but even running further experiments to give better intuition. I will update my score accordingly.

**Limitations:**

Yes, limitations are discussed.

**Paper Formatting Concerns:**

No formatting concerns.

**Quality:**

3

**Strengths And Weaknesses:**

## Strengths

The paper is very clear in its goals and how it attempts (and succeeds) to unify two distinct but very related approaches to in-context learning. There is extensive empirical evidence using various methods and models to support the claims. The findings are quite clear and easy to interpret. The figures clearly communicate the results (especially Figure 1). Lastly, I appreciate the authors share their code, which seems easy to understand.

## Weaknesses

The main weaknesses come from the ablation studies in section 5.3. The authors do not clarify how the ablations are done, but from previous sections (line 127), I'm inferring that they did a 0 ablation, where they set the output of a given head to 0. I'm not sure if this is the best type of ablation, as it might shift the activations out of distribution. I'm not majorly concerned about this, as your random head ablations using this method do not seem to have a negative effect. However, I think a mean ablation would be a more principled approach. See [this paper](https://arxiv.org/abs/2404.15255) for a discussion.

Of greater interest is the number of heads ablated and where these heads are. As we know, there exists a bunch of induction heads across multiple layers in large models. Do you ablate them all? Where are they located? Do you identify them simply using the prefix matching score on a random text? These are all important questions regarding the interpretation of the results. Apologies if I missed these details, but I could only find a description of how you construct the task vectors in the Appendix, not of these ablation decisions.

---

I believe this is already a strong paper. I'd be further willing to increase my score if the questions regarding the ablation studies above (especially regarding the details), and the clarification questions below are addressed adequately.

---

> ### Author Rebuttal · Authors · 2025-07-31
>
> **Dear Reviewer 4WZT,**
>
> Thank you for your constructive feedback on our manuscript, which we greatly appreciate. Below, we respond to the weaknesses and questions you raised.
>
> ---
>
> ## **W1 Part 1 and W2: Clarifications about the attention head ablation approach**
>
> We appreciate your concerns regarding the details of our attention head ablation procedure. As described in **lines 314-319** in Section 5, we use a **zero-ablation** approach. Regarding your question about the number of heads ablated and how they are identified: we ablate the top 10% of heads ranked by IH and PTH scores, as well as a control group consisting of 10% randomly selected heads that exclude the top 10% IHs and PTHs. Further details on the computation of IH and PTH scores can be found in **Appendix G**.
>
> In **Appendix G**, we explain that we use the first 50 queries from each dataset to compute these scores. For PTHs, each head's score is calculated by summing the attention weights it assigns to the immediately preceding token at all token positions across a query, and then summing over all 50 queries. For IHs, we first append 8 demonstrations to each query. Then, we compute each head's IH score by summing the attention weights it assigns to the 8 demonstration labels at the final ":" position of the query, and then summing over all queries.
>
> In response to your question about the **layerwise distribution of ablated top IHs**, we analyzed the top 10% IHs in Llama3-8B. Specifically, we divided the model's 32 layers into 8 equal intervals (1-4, 5-8, ...) and computed the proportion of IHs found in each interval:
>
> |Layer Interval|1-4|5-8|9-12|13-16|17-20|21-24|25-28|29-32|
> |--|---|---|----|-----|-----|-----|-----|-----|
> |Percentage|2.4510|6.0458|16.1765|20.2614|23.5294|18.4641|7.8431|5.2288|
> *Table 1: Distribution of ablated top 10 IHs across layer intervals*
>
> The results show that IHs are most concentrated in the middle layers—precisely where we observe phase transitions in ICL—supporting our claim about a two-stage ICL mechanism.
>
> ---
>
> ## **W1 Part 2: Using mean ablation as an alternative approach**
>
> We agree with your suggestion that supplementing our findings with **mean ablation** results strengthens their robustness. Accordingly, we repeated the experiments using mean ablation on Llama3-8B across all 6 datasets. We report the dataset-averaged layerwise dynamics of **separability** and **alignment** under different ablation conditions. We report **separability score** and **singular alignment** as a representative of all alignment measures.
>
> |Separability Score|...|14|15|16|17|18|19|20|21|22|23|24|25|26|27|28|29|30|31|32|
> |--|---|--|--|--|--|--|--|--|--|--|--|--|--|--|--|--|--|--|--|--|
> |ICL|...|0.8920|0.8992|0.8991|0.9073|0.9012|0.9106|0.9089|0.9081|0.9027|0.9010|0.8921|0.8856|0.8847|0.8848|0.8829|0.8808|0.8783|0.8764|0.8547|
> |PTH ablated|...|0.5584|0.5515|0.5434|0.5516|0.5562|0.5582|0.5574|0.5579|0.5478|0.5503|0.5433|0.5502|0.5556|0.5496|0.5516|0.5331|0.5368|0.5473|0.5225|
> |IH ablated|...|0.6473|0.6576|0.6838|0.6897|0.6990|0.6948|0.6956|0.6841|0.6712|0.6600|0.6584|0.6536|0.6504|0.6505|0.6352|0.6360|0.6340|0.6330|0.6176|
> |Random heads ablated|...|0.7105|0.7515|0.7659|0.7609|0.7738|0.7854|0.7852|0.7873|0.7885|0.7841|0.7824|0.7951|0.7902|0.7891|0.7830|0.7838|0.7829|0.7698|0.7248|
>
> |Singular Alignment|...|14|15|16|17|18|19|20|21|22|23|24|25|26|27|28|29|30|31|32|
> |--|---|--|--|--|--|--|--|--|--|--|--|--|--|--|--|--|--|--|--|--|
> |ICL|...|0.0291|0.0407|0.0634|0.0787|0.0948|0.1351|0.1375|0.1440|0.1649|0.1862|0.2016|0.1996|0.2090|0.2095|0.2041|0.2033|0.1917|0.1533|0.1015|
> |PTH ablated|...|0.0137|0.0150|0.0217|0.0221|0.0293|0.0373|0.0380|0.0428|0.0624|0.0816|0.0945|0.1262|0.1268|0.1278|0.1317|0.1456|0.1290|0.0883|0.0287|
> |IH ablated|...|0.0238|0.0234|0.0226|0.0201|0.0318|0.0375|0.0269|0.0332|0.0395|0.0314|0.0324|0.0313|0.0353|0.0342|0.0270|0.0285|0.0234|0.0216|0.0156|
> |Random heads ablated|...|0.0170|0.0260|0.0485|0.0604|0.0890|0.1369|0.1362|0.1417|0.1595|0.1722|0.1914|0.1923|0.2007|0.2065|0.1987|0.1945|0.1819|0.1345|0.0597|
>
> *Table 2: Impacts on separability score and singular alignment when IH, PTH, or random heads are ablated*
>
>
> These results confirm our hypothesis: **PTHs primarily affect separability**, while **IHs primarily affect alignment**.
>
> ---
>
> ## **Q1: PTHs and mean-based alignment**
>
> As explained in **lines 324–326**, ablating PTHs can substantially impact mean-based alignment because this metric depends on the norm of the difference between the hidden state cluster means projected onto the label unembedding direction. If the label-conditioned clusters are not separable (as happens when PTHs are ablated), this norm becomes small, reducing alignment. In contrast, ablating IHs affects the alignment between this difference and the label unembedding differnce, but does not collapse the cluster separation that much—hence could cause a smaller impact on separability.
>
> ---
>
> ## **Q2: Regarding the referenced paper**
>
> Thank you for bringing up this relevant work. We believe the differences in findings can be attributed to two main factors:
>
> **First**, the two studies use different methods to identify IHs. The paper in question computes prefix matching scores on random symbol sequences (e.g., `abc...abc...`), whereas we evaluate attention to demonstration labels in ICL-formatted prompts, following the approach in [1] (the original paper proposing function vector heads). This method ensures that the identified IHs are more task-relevant. Indeed, [1] also observed that 3 of their top 10 function vector heads exhibited IH-like attention.
>
> To validate this, we conducted an experiment comparing the impact of ablating IHs identified via the method in the referenced paper versus ours. We applied mean ablation on SST-2 and measured output alignment:
>
> |Output Alignment|...|14|15|16|17|18|19|20|21|22|23|24|25|26|27|28|29|30|31|32|
> |---------------|---|--|--|--|--|--|--|--|--|--|--|--|--|--|--|--|--|--|--|--|
> |IHs ablated (approach in paper)|...|0|0|0|0|0|0|0.0023|0.0023|0.0218|0.0310|0.0677|0.0482|0.0745|0.0975|0.0206|0.0057|0|0|0.0218|
> |IHs ablated (ours)|...|0|0|0|0|0|0|0|0|0|0|0|0|0|0|0|0|0|0|0|
> *Table 3: Ouput Alignment with ablation of IHs identified using two different criteria*
>
> The results confirm that **the outputs of our IHs encode more task-specific semantics** and serve as stronger task vectors.
>
> **Second**, while top-2% IHs and function vector heads do not perfectly overlap, the paper also reports **strong correlation** between their induction and FV scores across all heads. For example, top function vector heads often fall in the 90–95% percentile of induction scores and vice versa. This supports our use of top IHs as **effective, albeit not optimal, FV heads**.
>
> ---
>
> ## References
>
> [1] Todd, Eric, et al. "Function vectors in large language models."

---

> > ### Comment · Reviewer_4WZT · 2025-08-03
> >
> > Thanks for the very thorough response. I'm now further convinced of the reported findings, as they seem to hold through across several different methods of ablation and analysis. I also thank the authors for not only verbally addressing my last question but even running further experiments to give better intuition. I will update my score accordingly.

---

> > > ### Author Response · Authors · 2025-08-03
> > >
> > > Thank you for your thoughtful follow-up and for taking the time to consider our additional clarifications and experiments. We truly appreciate your careful evaluation and are grateful for your updated assessment of our work.

---

### Official Review · Reviewer_Ua9F · 2025-07-01

**Clarity:** 3
**Significance:** 3
**Originality:** 3
**Rating:** 5
**Confidence:** 4

**Summary:**

This paper investigates the unified mechanism of attention heads and task vectors on the classification performance of ICL. The authors start by theoretically deriving two conditions: output alignment and directional alignment for the hidden states of the queries to achieve maximum separability, which is the upper bound of the classification accuracy. They further propose several tractable metrics to measure the separability, output alignment, and directional alignment for various real-world LLMs. The experiments reveal the dynamics of the separability and alignment across the layers of LLMs, and how the previous token heads (PTH) and induction heads (IH) affect these features.

**Questions:**

1. In lines 309-310, based on my understanding, shouldn't the PTHs attend to "I like it." instead of "Answer"?
2. Could you further elaborate on, at least intuitively, how PTHs enhance separability and how IHs improve alignment? I find the explanations from line 329 to line 338 a bit vague.
3. Could you discuss how the findings in this work could potentially benefit the design of ICL algorithms?

**Ethical Concerns:**

["NO or VERY MINOR ethics concerns only"]

**Final Justification:**

The authors' rebuttal has addressed most of my concerns. The inclusion of the generation task strengthens the generality of the results. Moreover, the proposed denoising and classifier-based approaches offer practical methods for enhancing the classification performance of ICL. Overall, this work provides a comprehensive and in-depth analysis of how ICL works from the perspective of hidden representations. That said, I still find the contribution somewhat limited in scope, as it remains confined to classification tasks. As such, I will maintain my current rating.

**Limitations:**

Yes.

**Quality:**

3

**Strengths And Weaknesses:**

**Strengths**

1. This work depicts a clear view of the evolution of the hidden features induced by ICL, offering a key view for explaining how ICL improves the classification performance: ICL mainly improves the alignment between the hidden states of the queries and the label-related unembedding vectors, rather than improving separability.

2. The proposed separability and alignment metrics are theoretically motivated. Moreover, these two metrics can serve as two fine-grained indicators of the performance in classification tasks of ICL.

3. The empirical results of the layer-wise trends of separability and alignment (Fig. 2 and Fig. 3) are convincing due to the wide range of the tested model architectures, model sizes, datasets, and experimental settings (number of demonstrations, how the demonstrations are selected, etc.).

4. The findings of the roles in shaping the hidden representations of different types of attention heads are novel and interesting.

**Weaknesses**

1. The theory and experiments are limited to classification tasks. The proposed metrics cannot be applied to more diverse non-classification tasks, such as general QA or instruction following tasks, which are more common in practice.

2. The mechanism of why PTHs improve separability and why IHs improve alignment is not fully revealed in this work.

3. How the empirical findings could inspire potential improvements of ICL designs is not discussed.

---

> ### Author Rebuttal · Authors · 2025-07-25
>
> **Dear Reviewer Ua9F,**
>
> Thank you for your thoughtful review of our manuscript, which we greatly appreciate. Below, we respond to the weaknesses and questions you raised.
>
> ---
>
> ## **W1: Limitations of solely focusing on classification tasks**
>
> We appreciate your view that structuring our analysis solely around classification tasks seems limiting. Per your suggestion, and to address your concern that our geometric framework of ICL may not generalize to generation tasks, **we repeat our experiments on a text generation dataset** with the following setup. The demonstrations and queries take the format "Write a favourable/unfavourable review of {subject}:" where {subject} is the name of a kind of food (e.g., tapas). The labels are GPT-generated favourable or unfavourable reviews about the {subject}. Eight demonstrations and a query are concatenated to form each ICL-style prompt, and we compute our separability and alignment measures, as well as the effective dimension, on the dataset of prompts. The query labels needed for training the classifier to compute the separability score are determined based on whether a favourable or unfavourable review is requested in the query. We also compute the alignment measures with respect to the unembedding vectors of the "positive" and "negative" tokens to address the issue that labels for generation tasks comprise multiple tokens and take indefinite forms. Below in Table 1, we report the results concerning the separability score and singular alignment (as a representative alignment measure) across all layers of Llama3-8B in the zero-shot and ICL settings. (All experimental results presented in the rebuttal are conducted on Llama3-8B.)
>
> | Singular Alignment      | ...    | 14     | 15     | 16     | 17     | 18     | 19     | 20     | 21     | 22     | 23     | 24     | 25     | 26     | 27     | 28     | 29     | 30     | 31     | 32     |
> |-------------------------|--------|--------|--------|--------|--------|--------|--------|--------|--------|--------|--------|--------|--------|--------|--------|--------|--------|--------|--------|--------|
> | zero-shot               | ...    | 0.0304 | 0.0026 | 0.0163 | 0.0132 | 0.0200 | 0.0158 | 0.0147 | 0.0076 | 0.0057 | 0.0084 | 0.0088 | 0.0054 | 0.0064 | 0.0135 | 0.0155 | 0.0278 | 0.0278 | 0.0310 | 0.0108 |
> | ICL                     | ...    | 0.0270 | 0.0360 | 0.0410 | 0.0361 | 0.0545 | 0.0899 | 0.0762 | 0.0836 | 0.0918 | 0.0802 | 0.0731 | 0.0717 | 0.0766 | 0.0781 | 0.0547 | 0.0498 | 0.0613 | 0.0626 | 0.0810 |
>
> | Separability Score          | ...    | 14     | 15     | 16     | 17     | 18     | 19     | 20     | 21     | 22     | 23     | 24     | 25     | 26     | 27     | 28     | 29     | 30     | 31     | 32     |
> |-------------------------|--------|--------|--------|--------|--------|--------|--------|--------|--------|--------|--------|--------|--------|--------|--------|--------|--------|--------|--------|--------|
> | zero-shot               | ...    | 0.992  | 0.992  | 0.985  | 0.985  | 0.977  | 0.969  | 0.969  | 0.961  | 0.961  | 0.969  | 0.969  | 0.969  | 0.969  | 0.969  | 0.977  | 0.969  | 0.977  | 0.992  | 1      |
> | ICL                     | ...    | 1      | 1      | 1      | 1      | 1      | 1      | 1      | 1      | 1      | 1      | 1      | 1      | 1      | 1      | 1      | 1      | 1      | 1      | 1      |
> *Table 1: Layer-wise trend of singular alignment and separability score for the generation task: ICL (8-shot) and zero-shot*
>
> These results demonstrate the increase in alignment measures in the middle layers unique to the ICL setting and show that both ZSL and ICL hidden states exhibit high separability, thereby demonstrating the **generalizability of our framework to generation tasks**.
>
> ---
>
> ## **W2 & Q2: Explanation for why PTHs improve separability, IHs improve alignment**
>
> Below is a more detailed exposition building on our explanation in Section 5.3. First, it is widely documented that the hidden states of texts with different semantic values (e.g., positive/negative) are linearly separable in LLMs ([1][2][3]). PTHs mainly attend to the query texts with positive or negative sentiment and thus can effectively encode such separability into the final token's hidden state through their outputs, formed by the query texts' hidden states via their embedding matrices. We have also conducted an experiment showing that ablating PTHs affects separability in the zero-shot case (where only the query texts exist), thus validating this explanation.
>
> | Separability Score (SST-2)          | ...    | 23     | 24     | 25     | 26     | 27     | 28     | 29     | 30     | 31     | 32     |
> |-------------------------------------|--------|--------|--------|--------|--------|--------|--------|--------|--------|--------|--------|
> | zero-shot                           | ...    | 0.927  | 0.917  | 0.927  | 0.929  | 0.927  | 0.927  | 0.922  | 0.922  | 0.911  | 0.917  |
> | zero-shot (PTH ablated)             | ...    | 0.819  | 0.817  | 0.819  | 0.814  | 0.821  | 0.823  | 0.817  | 0.810  | 0.826  | 0.833  |
> *Table 2: Layer-wise trend of separability score for the SST-2: zero-shot & PTH ablation*
>
> Second, in the middle to late layers, IHs emerge and attend to the label tokens. Via the same mechanism discussed above, they encode the information of the semantically rich label tokens into the final token's hidden state, thus promoting alignment. The alignment between the query hidden states and their respective labels is achieved through IHs distributing attention strengths among the demonstration labels in context (discussed in detail in [4][5]). Our experiment in Appendix H.2 (showing that ablating IHs has less effect when the query label is not in context) also supports this explanation.
>
> ---
>
> ## **W3: Implications for ICL performance**
>
> We agree with you that connecting our analysis to practical implications regarding ICL performance is important. We would like to first draw your attention to Section 5.2 and Fig. 4 in particular, where we present a considerable increase in logits lens accuracy achieved through low-rank denoising, which exploits the alignment structure of the hidden states with label unembedding vectors. We apply rank-5 denoising to Llama3-8B final layer hidden states. As a result, we can increase the final layer output accuracy averaged across six datasets as reported in Table 3.
>
> Furthermore, the high separability score of ICL hidden states presented in Section 5.2 and Fig. 2 implies that we can also improve ICL accuracy by training a classifier on several ICL hidden states and using its prediction on the hidden states of the remaining prompts to get the label, resulting in an ICL calibration label akin to the one proposed by [6]. Using this method, we also manage to significantly increase the average ICL accuracy as shown in Table 3.
>
> |Method|ICL|ICL+Denoising|ICL+Classifier|
> |------|-----|----------------|----------------|
> |Accuracy|76.20|78.59|85.47|
> *Table 3: Improving ICL accuracy using denoising or classifier as calibration methods*
>
> In summary, **our findings lead to two methods to improve ICL performance that are lightweight, effective (the low-rank denoising method is also supervision-free), and can be seamlessly integrated into any ICL scenario involving text classification.**
>
> ---
>
> ## **Q1: Where PTHs attend**
>
> Yes, you are correct—PTHs should attend to the query text at the final position, as per our response to W2 & Q2. The reason we wrote that PTHs attend to the answer here is that, in the identification of PTHs, due to complex tokenization issues, it can be difficult to accurately delineate the token range of the query text and calculate attention scores. That is why we identify PTHs as the heads that attend to the immediately preceding token at each position, which is why we wrote that PTHs attend to "Answer" at the final position. But this is still valid at the final position because the hidden state of "Answer" will encode information from the query text due to the autoregressive structure of Transformers. **We have made the change here according to your suggestion.**
>
> ---
>
> # References
> [1] Marks, Samuel, and Max Tegmark. "The geometry of truth: Emergent linear structure in large language model representations of true/false datasets."
>
>
> [2] Park, Kiho, Yo Joong Choe, and Victor Veitch. "The linear representation hypothesis and the geometry of large language models."
>
> [3] Tigges, Curt, et al. "Linear representations of sentiment in large language models."
>
> [4] Yu, Zeping, and Sophia Ananiadou. "How do large language models learn in-context? query and key matrices of in-context heads are two towers for metric learning."
>
> [5] Halawi, Danny, Jean-Stanislas Denain, and Jacob Steinhardt. "Overthinking the truth: Understanding how language models process false demonstrations."
>
> [6] Cho, Hakaze, et al. "Token-based decision criteria are suboptimal in in-context learning."

---

### Note · Authors · 2025-08-13

**Dear Reviewers, AC, SAC, and PC,**

We sincerely thank you for your time, constructive engagement, and valuable feedback. We appreciate the opportunity to clarify our contributions and address the comments raised during the review process.

**Acknowledged Strengths**

We are grateful that reviewers recognized key strengths of our work:

All reviewers agreed our study offers clear mechanistic insights by illustrating a two-stage geometric framework for the evolution of hidden states enabling ICL, and that this framework naturally unifies explanations for the efficacy of special attention heads and task vectors, two leading mechanistic accounts of ICL.

All reviewers also found our geometric analysis well-motivated and strongly supported by extensive quantitative experiments across diverse architectures and datasets. Ua9F highlighted the theoretical rigor, while LiD7 valued the qualitative experiments on layer-wise semantic dynamics, which strengthen the quantitative results.

**Summary of Rebuttal Outcomes and Promised Revisions**

We addressed these major concerns raised by the reviewer:

We conducted our geometric analysis on a task generation task and confirmed outcomes consistent with classification tasks, addressing generalizability concerns of Ua9F and LiD7. Results are added to the appendix.

We demonstrated in 5.1, 5.2 that our findings naturally yield methods improving ICL performance in classification tasks, answering questions from Ua9F and LiD7 on downstream benefits. We will explicitly note these practical impications in 5.1 and 5.2.

We clarified, via Section 5.3 and Appendix G, the details of attention head ablation procedures, resolving queries from 4WZT and LiD7. We have included more details of the ablation procedures in 5.3.

**Addressing Remaining Concerns**

We emphasize that studying task vectors and ICL in classification is both valuable and novel despite the opinion that it is less common in prior literature (Reviewer kMpH). We have demonstrated task vector effectiveness from multiple angles (Section 5.3, rebuttal to LiD7, and response to kMpH), which should address kMpH’s doubts. Per kMpH’s suggestion, we added a discussion linking our work to **relevant** literature about ICL task vectors and clarified why some suggested papers are not appropriate.

We believe the clarifications, additional evidence, and strengths acknowledged by the reviewers demonstrate the novelty, rigor, and significance of our contributions.

---

### Decision · Program_Chairs · 2025-09-17

**Decision:**

Accept (poster)

**Comment:**

This paper studies the mechanisms underlying the in-context learning capabilities of transformers.
Focusing on in-context classification tasks, the authors characterize the layer-wise dynamics by revealing the two evolution stages of the hidden states, corresponding to separability and alignment of query hidden states.
In particular, it is shown that separability happens in early layers and alignment develops in later layers, and certain attention heads with specific functioning contribute to such phenomena.
This provides a unified view of the role of attention heads and task vectors in the context of in-context learning.

The paper is well-written and the idea is interesting and clearly illustrated.
The authors have derived novel insights into the cross-layer evolution of the geometric properties of the hidden states, by providing both empirical evidence and theoretical justification.
The findings provide explanations of ICL from a geometric perspective, which are valuable for understanding the inner workings of transformers.

The authors are encouraged to incorporate the additional results obtained during rebuttal into the final version of the paper.
Also, as noted by the reviewers, some missing references on related works about understanding cross-layer dynamics of transformers for ICL should be included to provide a more comprehensive background.